# Loss of FCoV-23 spike domain 0 enhances fusogenicity and entry kinetics

M. Alejandra Tortorici[1], Annette Choi[2], Cecily A. Gibson[1,3], Jimin Lee[1], Jack T. Brown[1], Cameron Stewart[1], Anshu Joshi[1], Sheri Harari[4], Isabelle Willoughby[1,5], Catherine Treichel[1,5], Elizabeth M. Leaf[1,5], Jesse D. Bloom[3,4], Neil P. King[1,5], Christine Tait-Burkard[6], Gary R. Whittaker[2,7,8] & David Veesler[1,3 ✉]

The ability of coronaviruses to recombine and cross species barriers affects human and animal health globally and is a pandemic threat[1,2]. FCoV-23 is a recently emerged, highly pathogenic recombinant coronavirus responsible for a widespread outbreak of feline infectious peritonitis. Here we report cryogenic electron microscopy structures of two FCoV-23 spike isoforms that correspond to the in-host loss of domain 0 observed in clinical samples. The loss of domain 0 markedly enhances the fusogenicity and kinetics of entry into cells and possibly enables biotype switching and lethality. We show that FCoV-23 can use several aminopeptidase N orthologues as receptors and reveal the molecular determinants of receptor species tropism, including a glycan that modulates human receptor engagement. We define antigenic relationships among alphacoronaviruses that infect humans and other mammalian species and identify a cross-reactive alphacoronavirus monoclonal antibody that inhibits FCoV-23 entry. Our results pave the way for the development of vaccines and therapeutics that target this highly pathogenic virus.

Coronaviruses circulating in wildlife pose a considerable public health threat owing to their zoonotic potential. Moreover, cross-species transmission between animals can lead to the establishment of new disease reservoirs, which may affect both wildlife populations and agricultural infrastructures. Companion animals, such as cats and dogs, can play a part in these spillover events by serving as intermediate hosts not only for coronaviruses[1,2] but also for viruses such as influenza[3] and rabies[4]. The current expansion of highly pathogenic avian influenza H5N1 reservoirs in wild birds, poultry, marine mammals and dairy cows, with occasional spillover to humans, highlights the ongoing risk of cross-species transmission[5,6].

Alphacoronaviruses infect a wide range of animal species and include porcine respiratory virus (PRCV), transmissible gastroenteritis virus (TGEV), feline coronaviruses (FCoVs), canine coronaviruses (CCoVs) and the human pathogens HCoV-229E[7], HCoV-NL63 (ref. 8) and CCoV-HuPn-2018 (which is closely related to HuCCoV_Z19Haiti)[9–11]. Extensive recombination occurs among alphacoronaviruses taxonomically classified as a single virus species (*Alphacoronavirus-1*), which infect several animal species, including cats, dogs, pigs and potentially rabbits[12–14]. Two serotypes of FCoV, designated FCoV-1 and FCoV-2, are major pathogens of wild felids and domestic cats. They are capable of an in-host biotype switch from the relatively mild feline enteric coronavirus (FECV) to the highly pathogenic, macrophage-tropic feline infectious peritonitis (FIP) virus[14,15]. Although the use of antiviral drugs in cats[16] has helped to alleviate the mortality burden of FIP, this treatment remains a prohibitively expensive option for many cat owners.

All coronaviruses are decorated with several copies of a homotrimeric spike (S) glycoprotein that is the target of neutralizing antibodies, drives viral entry through receptor binding and membrane fusion and contributes to pathogenesis[17–20]. Although many alphacoronaviruses use aminopeptidase N (APN) as a primary receptor[21], the identification of ACE2 as the HCoV-NL63 receptor[22] revealed a diversity of receptor use in this genus. Furthermore, the alphacoronavirus-specific S glycoprotein domain 0 (D0) mediates attachment to host cell-surface carbohydrates, which may facilitate the entry process[9,23–27]. Despite the diversity and medical and veterinary importance of these pathogens, cross-species transmission of animal alphacoronaviruses to humans and associated immunity remain poorly understood[28].

In 2023, a FCoV-2 virus (named FCoV-23) was newly identified as the cause of a large outbreak of FIP in cats on the Mediterranean island of Cyprus, and this virus led to the documented spread of an import-related case to the UK[13,16]. FCoV-23 is highly virulent, with most cats showing signs consistent with effusive FIP, profound neurological signs and elevated viral loads in cells with macrophage-like morphology in the colon[13]. Genome sequencing showed that FCoV-23 recombined its *S* gene and a small region of *Orf1b* with a CB/05-like virus, which has an *S* gene that is most closely related to the NA/09 alphacoronavirus. CB/05 and NA/09 were first identified in Italy and Greece, respectively, and are considered pantropic hypervirulent CCoVs, with such viruses now circulating widely in Europe and possibly other parts of the world[29,30]. Although the FCoV-23 genome is highly conserved among sequenced isolates, in-frame deletions of S D0 occurred in an almost cat-specific

[1]Department of Biochemistry, University of Washington, Seattle, WA, USA. [2]Department of Microbiology and Immunology, College of Veterinary Medicine, Cornell University, Ithaca, NY, USA. [3]Howard Hughes Medical Institute, Seattle, WA, USA. [4]Basic Sciences Division and Computational Biology Program, Fred Hutchinson Cancer Center, Seattle, WA, USA. [5]Institute for Protein Design, University of Washington, Seattle, WA, USA. [6]The Roslin Institute, Royal (Dick) School of Veterinary Studies, University of Edinburgh, Edinburgh, UK. [7]Public and Ecosystem Health, Cornell University, Ithaca, NY, USA. [8]Feline Health Center, Cornell University, Ithaca, NY, USA. ✉e-mail: dveesler@uw.edu

manner in >90% of studied cases, which suggested the occurrence of in-host evolution[13].

## The FCoV-23 S glycoprotein architecture

To reveal the 3D organization of the infection machinery of FCoV-23, we determined cryogenic electron microscopy (cryo-EM) structures of a prefusion-stabilized S glycoprotein ectodomain trimer construct comprising D0, which we designated 'S-long' (Fig. 1a,b, Extended Data Figs. 1 and 2 and Supplementary Table 1). FCoV-23 S-long comprises an amino-terminal $S_1$ subunit, folding as five domains designated 0 and A–D, and a C-terminal $S_2$ subunit (Fig. 1a,b). 3D classification of the cryo-EM data revealed the presence of two distinct S protein conformations: one with a population of trimers with three D0 regions swung out, and the other with a population of trimers with two D0 regions swung out and one D0 region in a proximal conformation. These structures were resolved at 2.3 Å and 2.7 Å, respectively (Fig. 1a,b and Extended Data Fig. 2). In both conformations, domain B (the receptor-binding domain (RBD)) adopts a closed state (Fig. 1a,b). We also determined a 2.5-Å resolution cryo-EM structure of a prefusion-stabilized FCoV-23 S protein ectodomain trimer lacking D0. This structure corresponds to the C10-DA isolate, which has a 211-amino-acid residue deletion spanning positions 53–264 of the S-long construct (Extended Data Figs. 1 and 3 and Supplementary Table 1). We designated this construct 'S-short', which is representative of the in-host deletion that occurs in >90% of sequenced viruses[13]. For this dataset, we identified a single conformational state that closely resembles the S-long structure except for the absence of D0, which indicated that no major structural changes occurred with the lack of D0. The overall FCoV-23 $S_1$ subunit architecture is reminiscent of that of the CCoV-HuPn-2018 $S_1$ subunit, with which it can be superimposed with a root mean square deviation (r.m.s.d.) of 0.89 Å over 496 aligned Cα pairs (excluding D0). The FCoV-23 $S_1$ subunit folds with a square-shaped tertiary structure, typical of alphacoronavirus and deltacoronavirus S glycoproteins, which sets them apart from the V-shaped organization observed in these proteins in betacoronaviruses and gammacoronaviruses[9,17,18,26,31] (Fig. 1c and Extended Data Fig. 4). The FCoV-23 $S_2$ subunit adopts a spring-loaded prefusion conformation and also closely resembles that of CCoV-HuPn-2018, with which it can be superimposed with a r.m.s.d. of 1.3 Å over 453 aligned Cα pairs (Fig. 1d). The absence of a polybasic cleavage site at the $S_1$–$S_2$ subunit junction concurs with the classification of FCoV-23 as part of FCoV-2 (ref. 32) and with the lack of proteolytic processing observed during S protein biogenesis, similar to CCoV-HuPn-2018 S[9] (Fig. 1g,h).

Structure-based phylogenetic analysis clustered the FCoV-23 RBD with the RBDs of CCoV-HuPn-2018, TGEV, PRCV and FCoV-2, which underscored their close evolutionary relationships (Fig. 1e). The similarity further extended to the conservation of key APN-interacting residues, including FCoV-23 S Y549/Q551 and W592 (corresponding to CCoV-HuPn-2018 residues Y543/Q545 and W586, respectively)[9,33,34], and to the conformational masking and glycan shielding of the receptor-binding loops by the N567 oligosaccharide (topological equivalent to the CCoV-HuPn-2018 N561 glycan) from a neighbouring RBD in the closed S protein trimer (Fig. 1f). FCoV-23 RBD mutations relative to other FCoV-2 sequences[13] map at the periphery or outside the predicted APN-binding motif; therefore, they are not expected to substantially affect host receptor tropism. These findings indicate that FCoV-23 uses APN as an entry receptor through a similar binding mode to that observed for CCoV-HuPn-2018 (ref. 9), PRCV and TGEV[35]. Notably, the FCoV-1 UU4 RBD is more distantly related to the FCoV-23 RBD than the alphacoronavirus PEDV and HCoV-229E RBDs or porcine deltacoronavirus RBD despite the classification of all FCoVs as a single viral species (Fig. 1e). This result is reminiscent of phylogenetic incongruence observed for merbecoviruses[36–38]. This extensive genetic distance further agrees with the lack of APN utilization for FCoV-1, for

which the receptor is unknown[33,39], and for HCoV-NL63, which uses the ACE2 receptor[22].

## FCoV-23 S sensitivity to host proteases

Host proteases are crucial determinants of cell tropism, host range and pathogenesis by activating coronavirus S glycoproteins for membrane fusion through cleavage[20,40,41]. To investigate the range of host cell proteases that can activate FCoV-23 S, we quantified the velocity of cleavage of a fluorogenic peptide comprising residues $S_{958}$KRKYR|SAIE$_{967}$ (where the vertical line indicates the position of the predicted scissile bond in the $S_2'$ cleavage site) in the presence of a panel of proteases previously shown to be involved in coronavirus S activation[42–45]. Trypsin, cathepsin L, plasmin, PC1 and factor Xa efficiently cleaved the FCoV-23 peptide, whereas minimal processing was observed with furin (Extended Data Fig. 5a–f). Furthermore, we did not observe major differences in proteolytic sensitivity at the $S_1$–$S_2$ junction or at the $S_2'$ cleavage site of FCoV-23 S-long and S-short anchored in the membrane of vesicular stomatitis virus (VSV) pseudoviruses after incubation with receptor, trypsin and proteinase K. This result suggests that D0 does not extensively affect S processing (Extended Data Fig. 5g–i).

Trypsin is present in the digestive tract and can support enteric infections[31], whereas plasmin and factor Xa are systemically secreted proteases that are involved in the blood coagulation cascade and may promote the broad tissue tropism of FCoV-23 in cats. Plasmin cleavage and the presence of a tyrosine residue at position P2 of the influenza A/WSN/33 haemagglutinin cleavage site is associated with neurotropism of this virus[46]. This finding may explain the neurotropism of FCoV-23 in cats given that it also has a P2 tyrosine residue.

## Lack of FCoV-23 S-mediated haemagglutination

We previously showed that the CCoV-HuPn-2018 S D0 haemagglutinates turkey, dog and human erythrocytes in a sialic-acid-dependent manner, which suggests that cell-surface sialosides may have a role in viral entry[9]. To evaluate the ability of the FCoV-23 S to engage cell-surface sialoside attachment factors, we incubated D0, domain A or domain B multivalently displayed on the surface of I53-50 nanoparticles (NPs) with cat, chicken, cow, turkey and human erythrocytes. CCoV-HuPn-2018 D0 NPs (positive control) haemagglutinated human, turkey and chicken erythrocytes, whereas we did not detect FCoV-23 S-mediated haemagglutination of any of the red blood cells tested (Extended Data Fig. 6a–f). Although the FCoV-23 D0 and CCoV-HuPn-2018 D0 share only 27% amino acid sequence identity, their structures superimposed with a r.m.s.d. of 5.8 Å over 197 aligned Cα pairs (out of 246 residues), which underscores their structural similarity. However, structure-based sequence alignment of these two domains highlighted numerous diverging regions that could explain the distinct haemagglutination phenotypes mediated by the most divergent domain among these two S glycoproteins (Extended Data Fig. 6g,h). These findings indicate that FCoV-23 S has a different specificity or lower binding affinity for cell-surface sialosides than CCoV-HuPn-2018 or that it does not engage sialoside attachment factors for entry.

## APN is a receptor for FCoV-23

To test whether FCoV-23 uses APN for entry, we used biolayer interferometry (BLI) to assess the ability of the FCoV-23 RBD to bind several APN orthologues. We detected binding of dimeric *Felis catus* APN (*Fc*APN) and *Canis lupus familiaris* (*Cf*APN) ectodomain dimers fused to a human crystallizable fragment (Fc) fragment, but not human or *Gallus gallus* APN-Fc (*Gg*APN-Fc), to the immobilized FCoV-23 RBD, as was the case for CCoV-HuPn-2018 (Fig. 2a and Extended Data Fig. 7a,b). The apparent binding avidities ($K_{d,app}$) of *Fc*APN and *Cf*APN were 11 and 1.3 nM, respectively (Fig. 2b,c and Supplementary

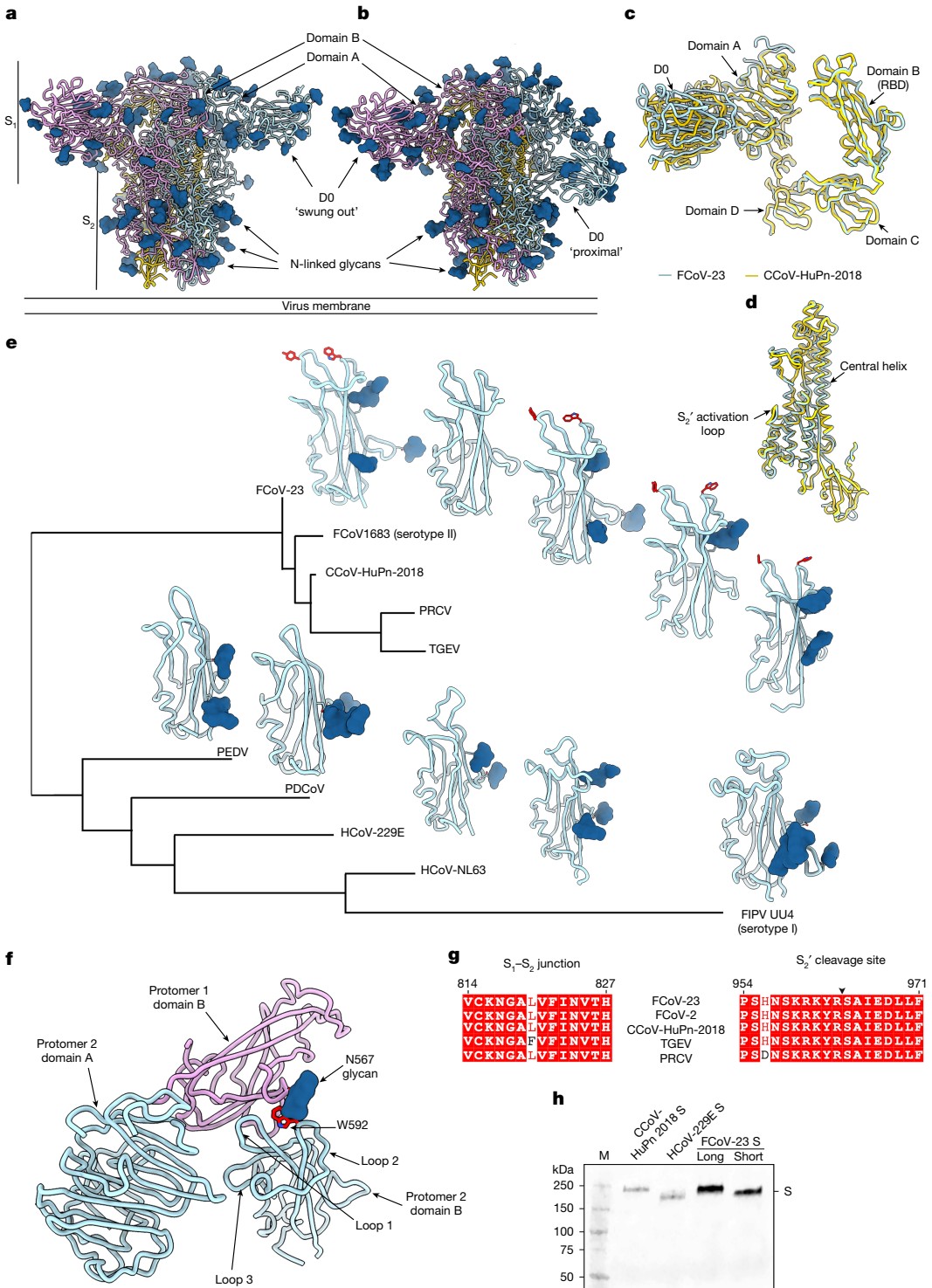

**Fig. 1 | Architecture and evolution of the FCoV-23 S glycoprotein. a**,**b**, Ribbon diagrams of the cryo-EM structures of prefusion FCoV-23 S-long in two distinct conformations defined by the positioning of D0: the 'swung out' conformation (**a**) and the 'mixed' structure (**b**). The structure with mixed D0 conformations (**b**) is rendered from a composite model of the global S and local D0 refinements. The positions of the $S_1$ and $S_2$ subunits are indicated. **c**, Superimposition of FCoV-23 (blue) and CCoV-HuPn-2018 (yellow) $S_1$ subunits with D0 swung out. **d**, Superposition of the FCoV-23 and CCoV-HuPn-2018 $S_2$ subunits. **e**, Structure-guided phylogenetic classification of alphacoronavirus and deltacoronavirus RBDs. Conserved APN-interacting residues are shown in red (except for the FCoV1683 RBD, which is an AlphaFold3-predicted structure[62]). **f**, Zoomed-in view of FCoV-23 S domain A and RBDs (domain B) showing the conformational masking and glycan shielding of the receptor-binding loops (labelled loop 1 and loop 2) in the context of the S trimer (only part of two protomers are shown for clarity). **g**, Amino acid sequence conservation of the residues spanning the $S_1$–$S_2$ subunit junction and the $S_2'$ cleavage site for FCoV-23, FCoV-2 (GenBank AFH58021.1), CCoV-HuPn-2018 (UniProt A0A8E6CMP0), TGEV (UniProt Q0PKZ5) and PRCV (UniProt Q84852). The residue numbering corresponds to FCoV-23 S-long (Extended Data Fig. 1). The arrowhead indicates the presumed $S_2'$ cleavage site. **h**, Western blot analysis of VSV particles pseudotyped with FCoV-23 S-long and S-short, HCoV-229E (P100E isolate, 2001, AAK32191.1) S and CCoV-HuPn-2018 S. The 76E1 monoclonal antibody[63] and the Alexa Fluor 680-conjugated goat anti-human antibody were used as primary and secondary antibodies, respectively. The western blot was carried out once. M, molecular mass marker.

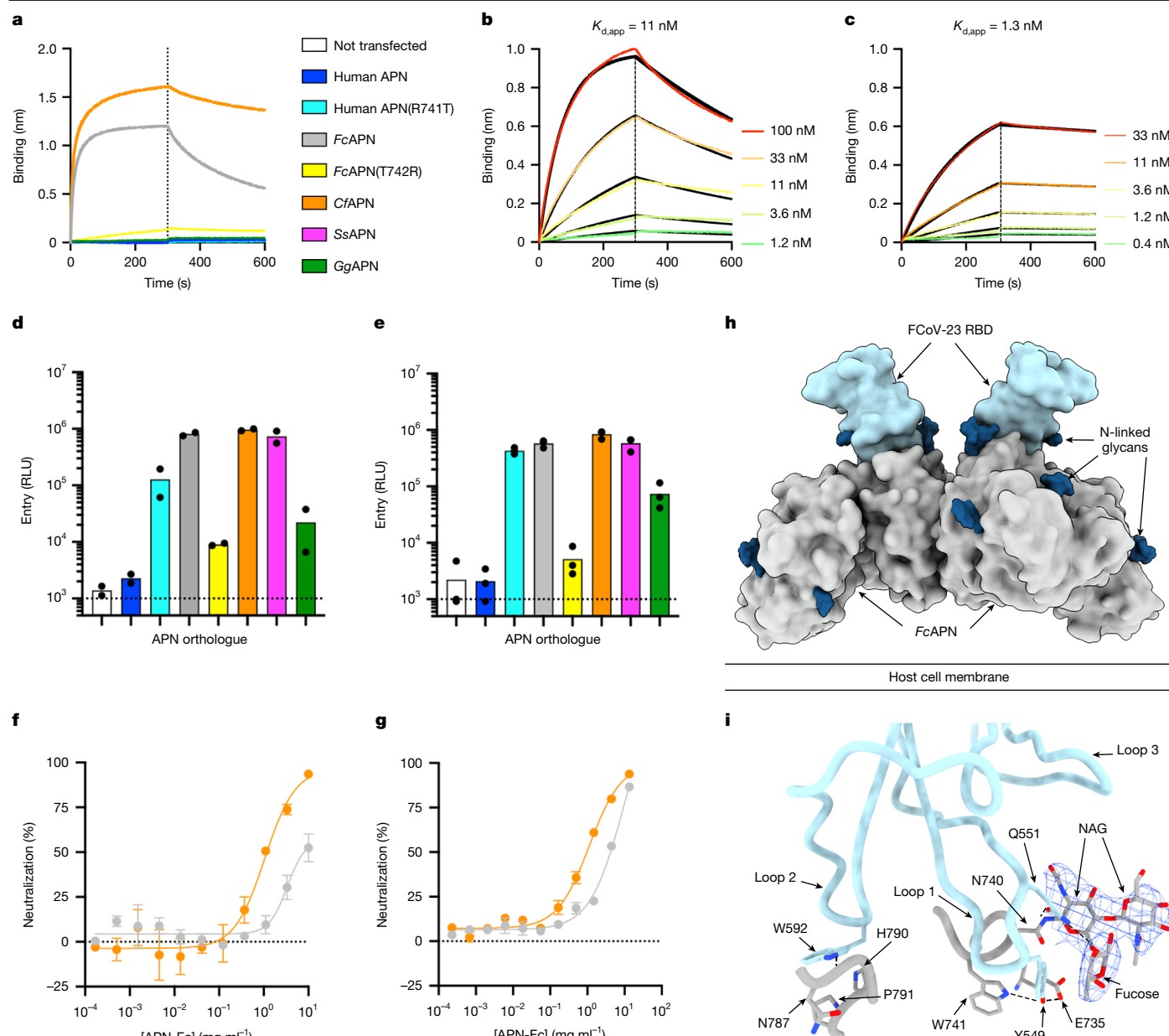

**Fig. 2 | APN is an entry receptor for FCoV-23. a**, BLI binding analysis of dimeric human APN-Fc, human APN-Fc(R741T) (glycan knock-in), *Fc*APN-Fc, *Fc*APN-Fc (T742R) (glycan knockout), *Cf*APN-Fc and *Gg*APN-Fc ectodomains at a concentration of 1 μM to immobilized biotinylated FCoV-23 RBD. **b,c**, BLI binding kinetic analysis of *Fc*APN-Fc (**b**) and *Cf*APN-Fc (**c**) to immobilized biotinylated FCoV-23 RBD. Global fit (1:1 model) to the data is shown in black and affinities are reported as $K_{d,app}$ owing to avidity resulting from the dimeric nature of APN. A representative experiment is shown out of two biological replicates. **d,e**, Entry of VSV particles pseudotyped with FCoV-23 S-long (**d**) or FCoV-23 S-short (**e**) into HEK293T cells transiently transfected with membrane-anchored *Fc*APN, *Fc*APN(T742R), *Cf*APN, *Ss*APN, *Gg*APN, human APN or human APN(R741T) orthologues. Each data point represents a biological replicate performed with technical duplicates. Bars represent the mean of 2–3 biological replicates.

The colour key is shown in **a**. RLU, relative luciferase units. **f,g**, Concentration-dependent inhibition of FCoV-23 S-long (**f**) and FCoV-23 S-short (**g**) VSV pseudovirus entry into HEK293T cells transiently transfected with membrane-anchored *Fc*APN mediated by dimeric *Fc*APN-Fc (grey) and *Cf*APN-Fc (orange) ectodomains. Each curve represents a biological replicate performed with technical duplicates. Error bars represent the s.e.m. of the technical duplicates. **h**, Surface representation of the cryo-EM structure of FCoV-23 RBD (light blue) in complex with the *Fc*APN ectodomain dimer (grey). N-linked glycans are rendered as dark blue surfaces. **i**, Close-up view showing key interactions formed between the FCoV-23 RBD (light blue) and *Fc*APN (grey). Dashed lines show select salt bridges and hydrogen bonds. The N740 glycan cryo-EM density is shown as a blue mesh. NAG, *N*-acetyl glucosamine.

Table 2). To evaluate the impact of the loss of D0 on receptor engagement, we carried out APN pull-down assays using prefusion FCoV-23 S-long and S-short ectodomain trimers immobilized on NTA cobalt-based beads. Both types of S trimers interacted with comparable efficiency with the *Cf*APN-Fc ectodomain (Extended Data Fig. 7c,d). This result rules out the idea that D0 has a role in inducing large conformational changes in the RBD, which aligns with the

structural data (except for the exclusive detection of closed trimers by cryo-EM).

We subsequently assessed the ability of HEK293T cells transiently transfected with a panel of membrane-anchored APN orthologues to enable the entry of VSV particles pseudotyped with FCoV-23 S-long, FCoV-23 S-short, CCoV-HuPn-2018 S or HCoV-229E S (Fig. 2d,e and Extended Data Fig. 7e,f). In agreement with the binding data, *Fc*APN

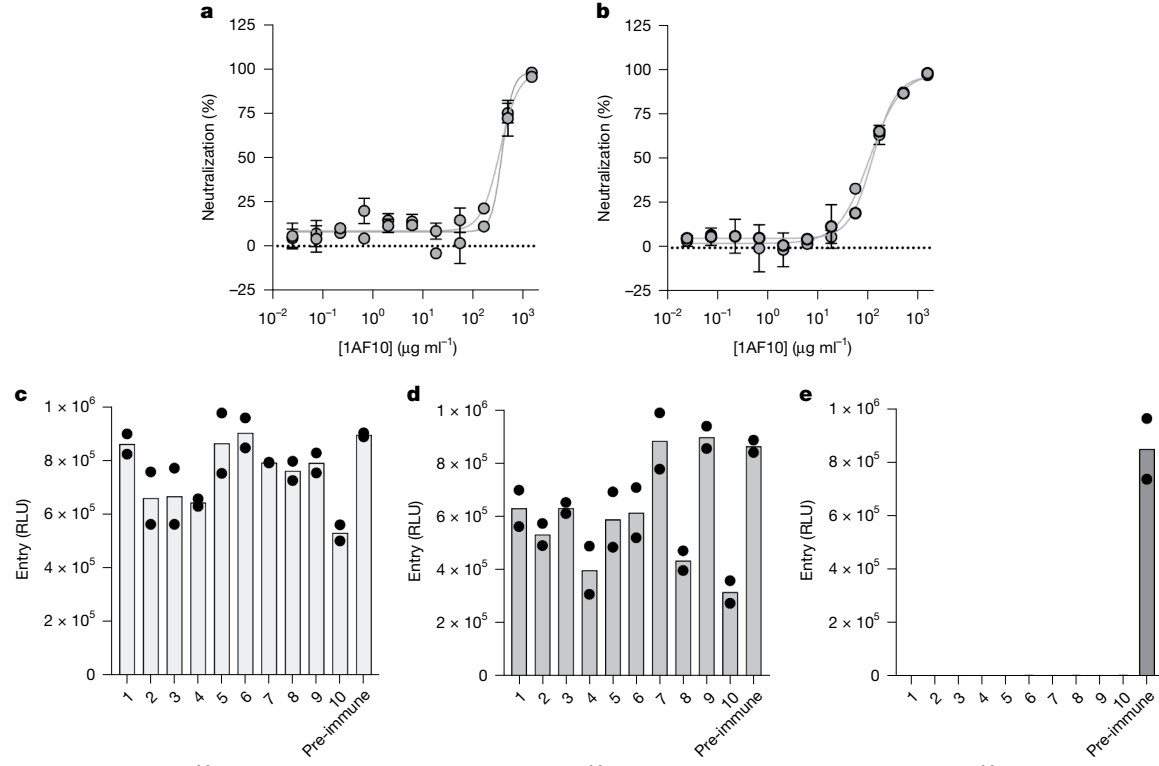

**Fig. 3 | Evaluation of cross-neutralization of FCoV-23 by alphacoronavirus-elicited antibodies. a,b,** Dose-dependent neutralization of FCoV-23 S-short (**a**) and FCoV-23 S-long (**b**) pseudotyped VSV particles in the presence of various concentrations of the 1AF10 neutralizing monoclonal Fab fragment using HEK293T cells transiently transfected with membrane-anchored *Fc*APN. Each curve represents a biological replicate (*n* = 2, using distinct pseudovirus batches) each performed with 2 technical replicates. Error bars represent the s.e.m. of the technical duplicates. **c**–**e,** Inhibition of FCoV-23 S-short (**c**)

CCoV-HuPn-2018 S (**d**) and HCoV-229E S (P100E isolate) (**e**) VSV pseudovirus entry into HEK293T cells transiently transfected with membrane-anchored *Fc*APN (FCoV-23, **c**), *Cf*APN (CCoV-HuPn-2018, **d**) or human APN (HCoV-229E, **e**) in the presence of a 1:10 dilution of HCoV-229E (P100E isolate) S-elicited mouse sera. Each data point represents a biological replicate (*n* = 2, using distinct pseudovirus batches) performed with technical duplicates. Bars represent the mean of two independent biological replicates.

and *Cf*APN promoted efficient entry of FCoV-23 S VSV particles into cells, as was the case for APN from *Sus scrofa* (*Ss*APN) (Fig. 2d,e). APN from *G. gallus* (*Gg*APN) enabled reduced but detectable entry, whereas wild-type human APN did not (Fig. 2d,e). The observation of pseudovirus entry mediated by *Gg*APN, for which we did not detect RBD binding, probably resulted from avidity due to the multivalent presentation of S trimers, as previously observed for other coronaviruses[37,38]. These data show that several APN orthologues render cells susceptible to FCoV-23 S-mediated entry, irrespective of the presence or absence of D0, a result that is in agreement with the pull-down data (Extended Data Fig. 7c,d). Finally, we observed concentration-dependent inhibition of FCoV-23 S-long, FCoV-23 S-short and CCoV-HuPn-2018 S VSV entry into cells mediated by dimeric *Fc*APN-Fc and *Cf*APN-Fc ectodomains, which underscores the APN-specific entry pathway of these viruses (Fig. 2f,g and Extended Data Fig. 7g–i). Collectively, these findings identify APN as a bona fide entry receptor for FCoV-23, which seems to have broad receptor species tropism, mirroring that of CCoV-HuPn-2018.

## Molecular basis of FCoV-23 engagement of APN

To understand how FCoV-23 recognizes APN, we determined a cryo-EM structure of the complex between the FCoV-23 RBD and *Fc*APN at 2.5 Å resolution (Fig. 2h,i, Extended Data Fig. 8 and Supplementary Table 1). One FCoV-23 RBD binds to each of the two protomers of the *Fc*APN dimer through interactions largely similar to those observed for the *Cf*APN-bound CCoV-HuPn-2018 RBD[9] and the *Ss*APN-bound PRCV RBD[35] structures. In receptor-binding loop 1, the $Y549_{FCoV-23}$ side chain

interacts with the $N740_{APN}$ glycan core fucose through CH–π interactions[47] and is hydrogen-bonded to the E735 and W741 side chains. By contrast, the $Q551_{FCoV-23}$ side chain is hydrogen-bonded to the $N740_{APN}$ side chain and interacts with the proximal N-acetyl-glucosamine and core fucose (Fig. 2i). In receptor-binding loop 2, $W592_{FCoV-23}$ is packed against $H790_{APN}$ and $P791_{APN}$, and its imino group is hydrogen-bonded to the main-chain carbonyl of $N787_{APN}$. We note that the substitution of $R540_{CCoV-HuPn-2018}$ with $L546_{FCoV-23}$ abrogates a salt bridge formed with *Cf*APN(E786) (corresponding to *Fc*APN(Q779)).

Given the similarities of the FCoV-23 and CCoV-HuPn-2018 RBDs and of their host receptor tropism, we speculated that the lack of an oligosaccharide at position N739 of human APN, which is present in *Cf*APN, *Fc*APN, *Ss*APN and *Gg*APN, explained the inability of FCoV-23 to engage the human APN orthologue, as is the case for CCoV-HuPn-2018 (ref. 9) and TGEV[35,48]. Transient transfection of a membrane-anchored human APN(N739) oligosaccharide knock-in mutant (R741T substitution) rendered cells permissive to FCoV-23 S VSV entry, which confirmed the importance of this glycan-mediated interaction for receptor engagement (Fig. 2d,e). Conversely, removal of this oligosaccharide from *Fc*APN (T742R substitution) reduced FCoV-23 S-mediated and CCoV-HuPn-2018 S-mediated entry into cells by two orders of magnitude (Fig. 2d,e and Extended Data Fig. 7e), which further demonstrated its key role for binding. Efficient HCoV-229E S-mediated entry occurred with human APN, *Fc*APN and *Gg*APN and did not depend on the presence of a glycan at positions N739 (human APN) or N740 (*Fc*APN) (Extended Data Fig. 7f). This result aligns with the fact that the HCoV-229E RBD recognizes a different APN site relative to the other viruses evaluated here[49].

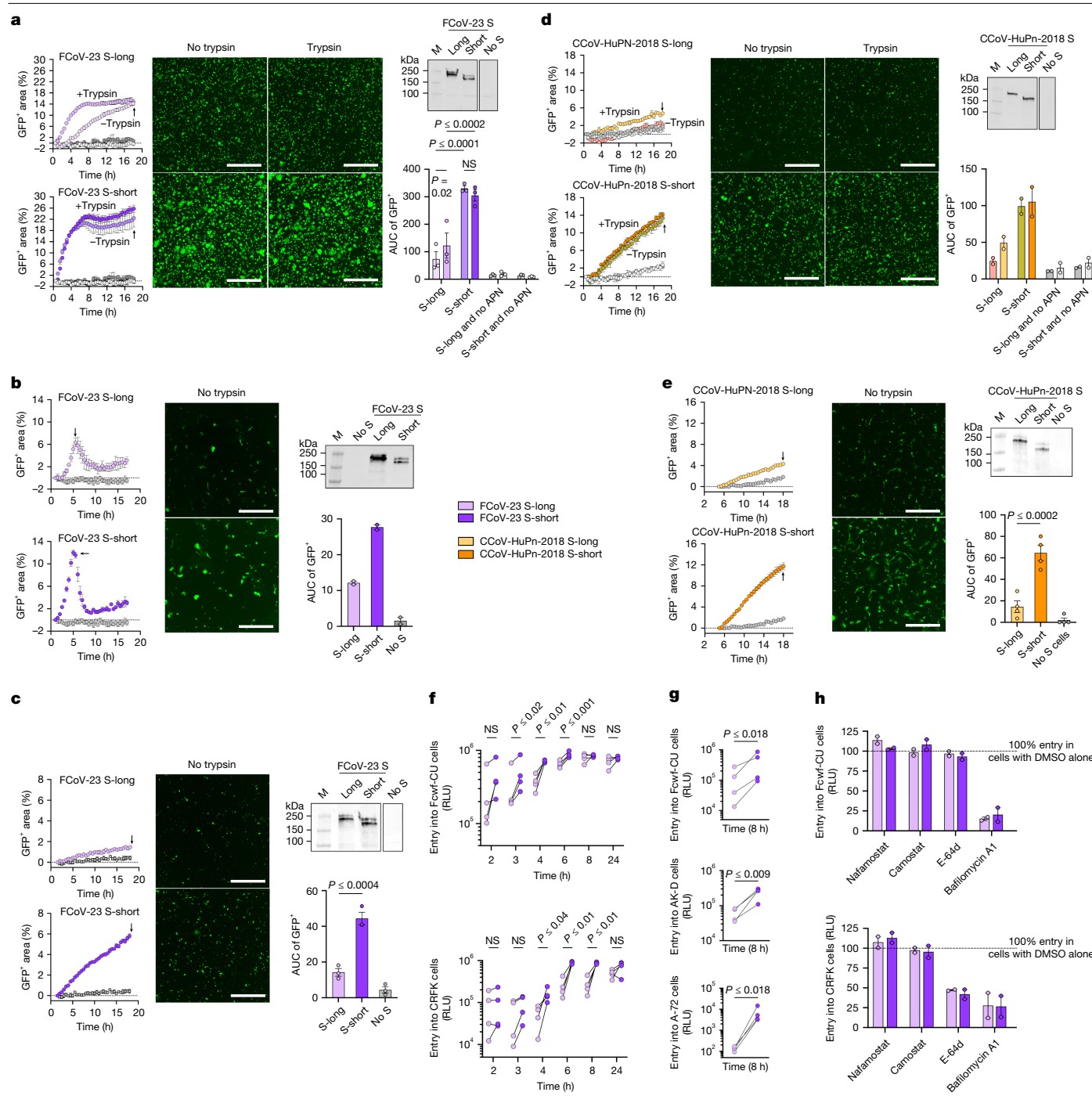

**Fig. 4 | See next page for caption.**

## FCoV-23 S targets feline and canine cells

To gain insights into the cell tropism of FCoV-23, we pseudotyped murine leukaemia virus (MLV) particles with FCoV-23 S-short or S-long constructs and assessed their ability to enter a panel of cell lines representing multiple mammalian species and tissues. FCoV-23 S-short and S-long MLV pseudoviruses effectively entered feline macrophage (Fcwf-CU) cells, feline kidney epithelial (CRFK) cells and feline epithelial lung (AK-D) cells. Notably, there was a higher entry of S-short MLV than S-long MLV (ratio of short to long: 2.8, 1.7 and 2.5, respectively, for each cell line) at 72 h after transduction (Extended Data Fig. 9a,b). Furthermore, we detected markedly more effective pseudovirus entry into canine A-72 fibroblast cells with FCoV-23 S-short MLV particles

compared with FCoV-23 S-long MLV particles (ratio of short to long: 780) at 72 h after infection (Extended Data Fig. 9a,b). Collectively, these data suggest that FCoV-23 has broad feline cell and tissue tropism. The results also agree with the pathology observed in infected cats, in which FCoV-23 is observed in multiple tissues[13].

## Antigenicity of the FCoV-23 S glycoprotein

FCoV-23 S shares around 82% and 48% amino acid sequence identity with CCoV-HuPn-2018 S and HCoV-229E S (P100E isolate), respectively, with the S1 subunits being more divergent than the S2 subunits. Furthermore, the FCoV-23 RBD shares about 94% and 28% amino acid sequence identity with the CCoV-HuPN-2018 and HCoV-229E RBDs, respectively.

**Fig. 4 | Loss of S D0 increases fusogenicity and entry kinetics. a–e**, Left, kinetics of cell–cell fusion (GFP⁺ area) between BHK-21–GFP1–10 cells transiently transfected with FCoV-23 S-long or S-short (**a**–**c**) or CCoV-HuPn-2018 S-long or S-short (**d**,**e**) with BHK-21–GFP11 target cells transiently transfected with membrane-anchored *Fc*APN (**a**) or *Cf*APN (**d**), cat macrophage-like Fcwf-CU cells (**b**), cat epithelial kidney CRFK cells (**c**) or canine fibroblast A-72 cells (**e**) endogenously expressing *Fc*APN or *Cf*APN and transiently transfected with GFP11. Middle, representative fields of view obtained 18 h (**a**,**c**,**d**,**e**) or 5.5 h (**b**) after S-expressing cells (untreated or pretreated with trypsin) were added to target cells are shown (time points are indicated by arrows on the corresponding left panels). Scale bars, 1,000 μm. Right, western blot quantification of membrane-anchored S expression in BHK-21–GFP1–10 cells detected using the 76E1 monoclonal antibody[63]. Data shown are one representative of 2–4 biological replicates. Each dot in the bar graphs represents a biological replicate (*n* = 3 with 2–4 technical replicates each for **a**, *n* = 2 with 5 technical replicates each for **b**, *n* = 3 with 5–10 technical replicates each for **c**, *n* = 2 with 2–3 technical replicates each for **d**, and *n* = 4 with 5–10 technical replicates each for **e**). Data are mean and s.e.m. Statistical analysis was performed using area under the curve (AUC) of the GFP⁺ area between the biological replicates with two-way analysis of variance (ANOVA) (**a**) or one-way ANOVA (**c**,**e**) with Tukey's multiple comparison test using data at *t* = 18 h or 5.5 h only for Fcwf-CU cells (**b**). NS, not significant. **f**, Entry kinetics of VSV particles pseudotyped with FCoV-23 S-long or S-short in Fcwf-CU cells (top) and CRFK cells (bottom). Each dot represents a biological replicate (*n* = 4, using distinct pseudovirus batches), each comprising 10–12 technical replicates. Comparison of entry values between S-long and S-short for each time point was performed using multiple ratio-paired *t*-tests. **g**, Entry kinetics of MLV particles pseudotyped with FCoV-23 S-long or S-short in Fcwf-CU cells (top), AK-D cells (middle) and A72 cells (bottom). Each dot represents a biological replicate (*n* = 4, using distinct pseudovirus batches) each comprising 4 technical replicates. Comparison of entry values between S-long and S-short for each time point was performed using a ratio-paired *t*-test. **h**, Relative entry of VSV pseudotyped with the indicated FCoV-23 S in Fcwf-CU cells (top) or CRFK cells (bottom) cells treated with 25 μM nafamostat, camostat, E-64d or 1 μM bafilomycin A1. Normalization was based on the entry values obtained from cells treated with DMSO alone. Each dot represents one biological replicate (*n* = 2, using distinct pseudovirus batches) each comprising 10–12 technical replicates. Means and s.e.m. of the two biological replicates are shown with bars and lines, respectively.

To investigate the antigenic relationships among alphacoronaviruses, we first evaluated the ability of the 1AF10 neutralizing monoclonal antibody, which targets the RBDs of PRCV and TGEV, to inhibit FCoV-23 S-mediated entry. The antigen-binding fragment (Fab) of 1AF10 neutralized FCoV-23 S-short and S-long VSV pseudoviruses with similar potency to that observed for CCoV-HuPn-2018 S VSV[9] (Fig. 3a,b and Extended Data Fig. 7h,i). These findings support the close relatedness between FCoV-23 S, CCoV-HuPn-2018, PRCV and TGEV S (Fig. 1e), which all cross-react with this monoclonal antibody and leads to blockade of receptor engagement. This result identifies 1AF10 as a possible countermeasure against these viruses.

We investigated the neutralizing activity of mouse hyperimmune sera obtained after vaccination with prefusion HCoV-229E S 2P (P100E isolate, 2001). Although all mouse sera completely neutralized HCoV-229E S VSV pseudovirus, they exhibited weak to no inhibition of FCoV-23 S-short VSV and CCoV-HuPn-2018 S VSV (Fig. 3c,e). These data suggest that previous HCoV-229E infection or vaccination would elicit (at best) weak cross-neutralization of FCoV-23 and CCoV-HuPn-2018. This result is in line with the more distant genetic relationships between the S glycoproteins, particularly the RBDs, of HCoV-229E and these viruses.

## D0 dampens fusogenicity and entry kinetics

To understand the functional impact of the loss of D0 on S-mediated syncytium formation, we used a membrane-fusion assay with a split GFP system to quantify membrane fusion after GFP reconstitution. We used BHK-21 cells stably expressing GFP1–10 (BHK-21–GFP1–10) and transiently transfected with FCoV-23 S-short or S-long to evaluate cell–cell fusion with GFP11-expressing cells, including BHK-21 transiently expressing membrane-anchored *Fc*APN as well as Fcwf-CU and CRFK cells, which endogenously express *Fc*APN. We observed markedly increased fusogenicity mediated by FCoV-23 S-short, relative to S-long, in all cell lines tested and at most time points in the presence of equivalent amounts of each of the two S isoforms (Fig. 4a–c and Extended Data Fig. 10a–e). Although the addition of exogenous trypsin to S-expressing cells enhanced S-long-mediated membrane fusion at early time points (Fig. 4a), which may be due to proteolytic removal of D0 and direct activation of the fusion machinery (Extended Data Fig. 5h), S-long remained less fusogenic than S-short.

Given the close evolutionary relationships between the FCoV-23 and CCoV-HuPn-2018 S glycoproteins, we evaluated whether the presence of D0 similarly affected S-mediated membrane fusion for this human virus. CCoV-HuPn-2018 D0 deletion led to a marked enhancement of syncytium formation between S-expressing BHK cells and BHK and canine fibroblast A-72 target cells (transiently transfected with or endogenously expressing *Cf*APN, respectively; Fig. 4d,e). Pretreatment of S-expressing cells with trypsin modestly promoted fusion mediated by S-long, but not by S-short (Fig. 4d). These results therefore point to a similar phenotype for loss of D0 for FCoV-23 and CCoV-HuPn-2018.

We next investigated whether the increase in fusogenicity promoted elevated kinetics of S-mediated entry into cells by comparing VSV pseudoviruses with FCoV-23 S-short and S-long at various time points after transduction. Compared with S-long VSV particles, S-short VSV particles entered 1.7-fold, 2-fold and 1.35-fold more efficiently in Fcwf-CU cells at 3, 4 and 6 h after infection, respectively, when comparing the geometric mean entry of each group (Fig. 4f, top, and Extended Data Fig. 10f). In CRFK cells, S-short VSV particles entered 3.5-fold, 3.9-fold and 3.5-fold more efficiently at 4, 6 and 8 h after infection, respectively, than S-long VSV particles (Fig. 4f, bottom, and Extended Data Fig. 10f). These differences were reduced at the end of the experiment (24 h time point), which may explain the small differences in MLV pseudovirus entry detected for the two isoforms 72 h after transduction (Extended Data Fig. 9a). Accordingly, we observed greater entry of MLV pseudotyped with FCoV-23 S-short than S-long at 8 h after transduction in Fcwf-CU (4.1-fold), AK-D (4-fold) and A-72 (41-fold) cells when comparing the geometric mean entry of each group (Fig. 4g and Extended Data Fig. 9b,c). These results show that loss of D0 promotes more rapid entry into target cells that endogenously express *Fc*APN or *Cf*APN, a result in agreement with the increased syncytium formation for FCoV-23 S-short relative to S-long.

Pseudovirus entry into Fcwf-CU cells was reduced by bafilomycin A1 (an inhibitor of endosomal acidification[37]) (Fig. 4h, top), and entry into CRFK cells was dampened by bafilomycin A1 and E-64d (a cysteine protease inhibitor) for both FCoV-23 S-short and S-long (Fig. 4h, bottom). These results, and the lack of effect of camostat and nafamostat (serine protease inhibitors), suggest that FCoV-23 favours an endosomal entry route in the cell lines tested. Given the comparable impact of host protease inhibitors on the entry of FCoV-23 S-short and S-long pseudoviruses along with the enhanced entry kinetics (VSV and MLV pseudoviruses) and fusogenicity (cell–cell fusion) mediated by FCoV-23 S-short relative to S-long, these properties probably reflect intrinsic functional differences of these two S isoforms instead of distinct entry routes.

## Discussion

Most alphacoronavirus S glycoproteins have a D0, not present in other coronavirus genera, that most probably arose through the duplication of domain A (the latter domain is often referred to as the

N-terminal domain for betacoronaviruses)[26]. Both domains adopt a similar galectin-like β-sandwich fold and have been shown to interact with host cell sialoside-attachment factors for alphacoronavirus CCoV-HuPn-2018 (ref. 9), TGEV[50] and PEDV[24,25] D0 and for OC43/HKU1 (refs. 51,52), MERS-CoV[53] and SARS-CoV-2 (ref. 54) A domains. The FCoV-23 S structures presented here revealed distinct D0 conformations, similar to those observed for CCoV-HuPn-2018 (ref. 9) and PEDV[55,56], which may correspond to snapshots of the conformational changes that lead to viral entry into cells. Notably, in-frame deletions of FCoV-23 S D0 occurred in an almost cat-specific manner in >90% of studied Cypriot and import FIP cases, which implicated in-host evolution[13]. These findings are reminiscent of the evolutionary process that led from TGEV to PRCV after deletion of D0 in the TGEV *S* gene, which resulted in distinct tissue tropism between the two viruses[50]. TGEV replicates mainly in small intestine epithelial cells and causes severe and often fatal diarrhoea in young piglets, whereas PRCV replicates mostly in respiratory epithelial cells and causes mild respiratory symptoms without diarrhoea. Therefore, the loss of D0 may have produced opposite outcomes in terms of severity for TGEV/PRCV and FCoV-23 and might alter FCoV-23 tissue tropism (as is the case for TGEV and PRCV). The high pathogenicity of FCoV-23 and its detection in macrophage-like cells by immunohistochemistry are characteristics of the virus biotype switch from FECV to FIP. Furthermore, the markedly greater fusogenicity and entry kinetics of FCoV-23 S-short in various cell lines than S-long suggest that deletion of D0 may promote increased syncytium formation, replication kinetics and pathology, as described for the SARS-CoV-2 Delta variant as a result of the P681R substitution[41]. These findings offer a possible explanation for the repeated loss of D0 among FCoV-23-infected cats, and this loss can occur genomically (as observed in vivo) and putatively proteolytically (as observed with FCoV-23 S-long VSV pseudovirus).

Coronavirus RBDs are main targets of neutralizing antibodies and typically account for most plasma-neutralizing activity induced by infection or vaccination against matched and mismatched viruses[57–59]. Structure-guided phylogenetic classification clustered the FCoV-23, CCoV-HuPn-2018 (ref. 9), PRCV, TGEV[35] and FCoV-2 RBDs, which engage the APN entry receptor with a similar binding mode. The observation that the FCoV-1 RBD architecture[60] is more divergent from the FCoV-2 RBD than from the HCoV-229E[49] and PEDV[56] RBDs underscores the distinct receptor engagement and evolutionary pathways of these two types of feline pathogens, which will probably require the development of serotype-specific vaccines[33,39]. The identification of APN as the FCoV-23 entry receptor, of multiple feline cell lines supporting FCoV-23 S-mediated entry and of a broad spectrum of proteases cleaving isolated S2′ peptides illuminate key factors modulating host and cell tropism. The rare detection of weak cross-neutralization observed with HCoV-229E S vaccine-elicited mouse sera underscores the antigenic distance between this human endemic coronavirus and FCoV-23, although both pathogens belong to the alphacoronavirus genus. However, some degree of broad humoral immune protection may result from alphacoronavirus exposure, either through weak viral cross-neutralization of distinct alphacoronaviruses or antibody-mediated Fc effector functions. Indeed, the latter mechanism was observed with S-elicited fusion machinery-directed polyclonal antibodies after mismatched sarbecovirus challenge[61].

Cross-species transmission of pathogens is one of the greatest threats to human and animal health worldwide and is exacerbated by travel, urbanization, deforestation and intensive farming. This threat is currently illustrated by the spillover of highly pathogenic avian influenza H5N1 virus to birds and mammals, including dairy cattle, cats and humans[5,6]. The *Alphacoronavirus-1* species comprise a set of highly recombinogenic viruses circulating in dogs, cats and pigs. This species has been proposed to be divided into two clades with functionally distinct S glycoproteins[12,14]. Furthermore, several *Alphacoronavirus-1* recombinant viruses have spilled over from dogs or cats to humans[2]. The wide circulation of *Alphacoronavirus-1*

combined with our poor understanding of their genetic diversity and the accelerating rate of viral spillovers to the human population emphasize the importance of preparing for the emergence of highly pathogenic coronaviruses.

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

## Methods

### Cell lines

The following cell lines used in this study were obtained from the American Type Culture Collection (ATCC): human epithelial embryo cells (HEK293T, CRL-3216), human lung carcinoma epithelial cells (A549, CRM-CCL-185), human lung adenocarcinoma epithelial cells (Calu3, HTB-55), African green monkey kidney cells (VeroE6, CRL-1586), rhesus monkey kidney cells (LLC-MK2, CCL-7), Syrian golden hamster kidney fibroblast cells (BHK-21, CCL-10), rat lung epithelial cells (L2, CCL-149), chicken embryo fibroblast cells (DF-1, CRL-3586), feline epithelial kidney cells (CRFK, CCL-94), feline airway epithelial cells (AK-D, CCL-150), canine tumour fibroblast cells (A-72, CRL-1542), canine epithelial kidney cells (MDCK, CCL-34), pig epithelial kidney cells (LLC-PK1, CL-101) and bovine epithelial kidney cells (MDBK, CCL-22). Human bronchial epithelial cells (HBEC cells) were provided by R. Cerione's laboratory at Cornell University. Ovine kidney cells (OV KID) and ovine fetal turbinate (OFT) cells were provided by D. Diel at Cornell University Animal Health Diagnostic Center. Feline macrophage-like cells, Fcwf-4, were initially obtained from the ATCC (CRL-2787), and the progeny (Fcwf-CU) that was significantly better at propagating feline coronavirus was selected by E. Dubovi and G.R.W. at Cornell University to be significantly better at propagating feline coronavirus[64]. VeroE6 cells expressing transmembrane serine protease TMPRSS2 (VeroE6-TMPRSS2, JCRB1819) were obtained from the Japanese Collection of Research Bioresources (JCRB Cell Bank). The cell lines ExpiCHO and Expi293F were obtained from Thermo Fisher. Cells were cultured at 37 °C in the medium recommended by the manufacturer in an atmosphere of 5% $CO_2$. Cell lines were not routinely tested for mycoplasma contamination nor were they authenticated.

### Plasmids

Genes used in this study were synthesized by GenScript, codon optimized for expression in mammalian cells and cloned into pcDNA3.1(+) between KpnI and XhoI, in-frame with a Kozak's sequence to direct translation. Ectodomains used the signal peptide derived from μ-phosphatase (MGILPSPGMPALLSLVSLLSVLLMGCVAETGT) and a C-terminal Avi-tag and octa-histidine tag. FCoV-23 S-long and S-short sequences were obtained from a previous study[13] and are described in Extended Data Fig. 1. To stabilize the prefusion FCoV-23 S ectodomain constructs, residues E1146 and L1147 were mutated[65] to proline, and a foldon trimerization domain was added. The HCoV-229E S ectodomain used for mouse immunizations was synthetized by GeneArt (Thermo Fisher) and comprised residues 1–1128, with a C-terminal truncation of 60 residues, and was stabilized by mutations of residues T871 and I872 to proline and by the addition of a foldon trimerization domain.

To pseudotype VSV(ΔG-luc)[66], membrane-anchored wild-type S glycoproteins with their native signal peptide encoding genes from HCoV-229E (P100E isolate, 2001, AAK32191.1) residues 1–1,155, TGEV (ABG8935.1) residues 1–1,425, CCoV-HuPn-2018 (QVL91811.1) residues 1–1,425, FCoV-23 S-long and S-short, CCoV-HuPn-2018 lacking D0 (residues 259–1,425) (QVL91811.1), all with C-terminal truncations of 23 residues, except for HCoV-229E with a 18 C-terminal truncation, were codon optimized for expression in mammalian cells and cloned into pcDNA3.1(+) as indicated above. FCoV-23 S-long, FCoV-23 S-short and CCoV-HuPn-2018 S lacking D0 also had a C-terminal GGS**YPYDVPDYA** sequence (HA-tag in bold).

FCoV-23 S and CCoV-HuPn-2018 S B domains (also referred as RBDs) matching with the full-length sequences indicated above included residues 529–677 and 523–671, respectively, and were fused to a C-terminal Avi-tag and His-tag for biotinylation and affinity purification.

Full-length APN constructs have been previously described[9]. In brief, APN from human (NCBI GenBank identifier NP_001141.2), *F. catus* (NP_001009252.2), *C. familiaris* (NP_001139506.1), *S. scrofa* (AGX93258.1) and *G. gallus* (ACZ95799.1) comprise residues 1–967, 1–967, 1–975, 1–963 and 1–967, respectively. APN ectodomains from human wild-type and R741T mutant (both comprising residues 66–967), *F. catus* (residues 64–967), *C. familiaris* (residues 71–975), *S. scrofa* (residues 62–963) and *G. gallus* (residues 69–967) from the same sequence codes shown above were fused to a thrombin-cleavage site followed by a human Fc fragment at the C-terminal end for affinity purification.

The 1AF10 Fab constructs for expression in mammalian cells of the heavy and light chains have been previously described[9]. For NP assembly, FCoV-23 D0 (residues 19–264), domain A (residues 265–504) and domain B (residues 529–677) were fused to the N terminus of the trimeric I53-50A NP component[67] using a 16-residue-long glycine and serine linker. CCoV-HuPn-2018 D0 (residues 17–259) fused to the N terminus of the trimeric I53-50A NP component has been previously described[9]. Heavy chain and light chain genes encoding 76E1 full IgG were inserted each into a different pcDNA3.1(+) plasmid, fused to the N-terminal CD5 signal peptide MPMGSLQPLATLYLLGMLVASVLA.

The plasmid pQCXIP-BSR-GFP11 to express GFP11 for fusion assays was obtained from Addgene (https://www.addgene.org/68716/).

### Mutagenesis

The membrane-anchored wild-type *Fc*APN-encoding plasmid was used as a template to knockout the glycan at position N740 (NWT to NWR) using the following nonoverlapping primers: 5′-aagaatt gg**a**gggaccacccc-3′ (the codon change T to R is indicated in bold) and 5′-tgtcacgcgctcaaagtggttg-3′ using fusion DNA polymerase (Thermo Fisher). After treating the PCR products with DpnI (New England Biolabs) for 1 h at 37 °C, amplified plasmids were purified using a PCR & DNA cleanup kit (Monarch) treated with T4 polynucleotide kinase (New England Biolabs) for 1 h at 37 °C and ligated using T4 DNA ligase (New England Biolabs) at 25 °C overnight before being used for the transformation of One Shot MAX Efficiency DH10B chemically competent cells (Invitrogen). Introduction of the desired mutations was verified by sequencing purified plasmids by Plasmidsaurus. Plasmids with the desired mutation were amplified and purified using an EndoFree mega kit (Qiagen) so that they were suitable for transfection into mammalian cells.

### Protein expression and purification

To produce stabilized FCoV-23 S-long and S-short, 200 ml Expi293F cells grown to a density of $3 × 10^6$ cells per ml and at 37 °C were transfected with 640 μl Expifectamine reagent (Thermo Fisher) and 200 μg of the corresponding plasmids following the manufacturer's recommendations. The day after transfection, feed and enhancer were added to the cells. At 4–5 days after transfection, supernatants were clarified by centrifugation at 400*g* for 15 min and resuspended in 20 mM Tris-HCl pH 8.0, 20 mM imidazole and 300 mM NaCl. Supernatants were further centrifuged at 14,000*g* for 30 min and passed through a 1 ml Histrap Excel column (Cytiva) previously equilibrated with binding buffer (25 mM Tris-HCl pH 7.4 and 350 mM NaCl). FCoV-23 S was eluted using a linear gradient of 500 mM imidazole. To produce stabilized HCoV-229E S 2P, 500 ml of Expi293F cells at $3 × 10^6$ cells per ml were transiently transfected using polyethylenimine linear (PEI). In brief, 500 μg DNA (1 μg ml$^{-1}$ final concentration) was mixed with Opti-MEM and 1.5 mg PEI (3 μg ml$^{-1}$ final concentration) was mixed with Opti-MEM. The PEI mixture was added to the DNA mixture and gently mixed. After 15 min of incubation at room temperature, the PEI–DNA mixture (25 ml) was added dropwise to cells. On day 4 after transfection, stabilized HCoV-229E S 2P was purified using the protocol described above.

To express B domains from FCoV-23 S and CCoV-HuPn-2018 S C-terminally fused to an Avi-tag and a histidine tag, 100 ml of Expi293F cells at $3 × 10^6$ cells per ml were transiently transfected with 320 μl Expifectamine and 100 μg of the respective plasmids following the manufacturer's instructions. Four days after transfection, supernatants were clarified by centrifugation at 800*g* for 10 min, supplemented with 20 mM imidazole, 300 mM NaCl and 25 mM Tris-HCl pH 8.0,

further centrifuged at 14,000$g$ for 30 min and passed through a 1 ml His trap HP column (Cytiva) previously equilibrated with binding buffer (25 mM Tris-HCl pH 7.4 and 150 mM NaCl). B domains were eluted using 25 mM Tris-HCl pH 7.4, 150 mM NaCl and a linear gradient of 500 mM imidazole. A similar protocol was used to express and purify the Fab domain of 1AF10 fused to an 8-residue histidine tag, except that 50 ml of Expi293F cells were transfected with a mixture containing 50 µg of each individual plasmid encoding the Fab light and heavy chain and 160 µl Expifectamine reagent (Thermo Fisher).

To express the human APN, FcAPN, CfAPN, SsAPN and GgAPN ectodomain orthologues fused to the Fc fragment of human IgG, Expi293F cells were transiently transfected with the respective plasmids following the manufacturer's protocol. In brief, 50 ml of Expi293F cells at $3 \times 10^6$ cells per ml were transfected using 160 µl Expifectamine and 50 µg APN plasmid. Four days after transfection, supernatants were clarified by centrifugation at 800$g$ for 10 min, supplemented with 300 mM NaCl and 25 mM Tris-HCl pH 8.0, further centrifuged at 14,000$g$ for 30 min and passed through a 1 ml HiTrap Protein A HP column (Cytiva). Proteins were eluted using 0.1 M citric acid pH 3.0 in individual tubes containing 200 µl of 1 M Tris-HCl pH 9.0 to immediately neutralize the low pH needed for elution. APN samples were further purified by size-exclusion chromatography (SEC) on a Superdex 200 column 10/300 GL (GE Life Sciences) previously equilibrated in 25 mM Tris-HCl pH 8.0 and 150 mM NaCl. Fractions containing the proteins were pooled and buffer exchanged to 25 mM Tris-HCl pH 8.0 and 150 mM NaCl.

To produce FcAPN and CfAPN without the Fc fusion for cryo-EM and for protease digestion experiments, respectively, the Fc fragments were removed using thrombin (Millipore Sigma) in a reaction mixture containing 3 µg thrombin per mg of FcAPN-Fc, 25 mM Tris-HCl pH 8.0, 150 mM NaCl and 2.5 mM $CaCl_2$ and incubated overnight at room temperature. The reaction mixture was loaded to a protein A column to remove uncleaved FcAPN-Fc and the Fc tag, and purified FcAPN was buffer exchanged to 25 mM Tris-HCl pH 8.0 and 150 mM NaCl.

To express the 76E1 full IgG, 200 ml of Expi293F cells at $3 \times 10^6$ cells per ml were transiently transfected using Expifectamine and a mix containing 200 µg each of plasmids encoding the IgG heavy chain and light chain following the manufacturer's protocol. Four days after transfection, Expi293 cell supernatant was clarified by centrifugation at 4,121$g$ for 30 min, supplemented with 20 mM phosphate buffer pH 8.0 and 0.5 mM phenylmethylsulfonyl fluoride. Supernatant was passed through a protein A affinity column (Cytiva) twice before being washed with 20 mM phosphate buffer pH 8.0. Antibody 76E1 was eluted with 100 mM citric acid pH 3.0 into tubes containing 1 M Tris-HCl pH 9.0 for immediate neutralization of the low pH used for elution. Eluted 76E1 antibody was buffer exchanged into 20 mM phosphate buffer pH 8.0 and 100 mM NaCl, and purity was evaluated by SDS–PAGE.

### Protein biotinylation
B domains of FCoV-23 S and CCoV-HuPn-2018 S were biotinylated using a BirA biotin-protein ligase standard reaction kit (Avidity) following the manufacturer's protocol. In a typical reaction, 40 µM B domains was incubated overnight at 4 °C with 2.5 µg BirA enzyme in reaction mixtures containing 1× BiomixB, 1× BiomixA and 40 µM BIO200. The B domains were further separated from BirA by SEC using Superdex 75 increase 10/300 GL (GE LifeSciences) and concentrated using 10 kDa filters (Amicon).

### BLI binding assays
Biotinylated B domains from FCoV-23 S and CCoV-HuPn-2018 S were immobilized at 2 µg ml$^{-1}$ in 10X kinetics buffer (Sartorius) to 0.5 nm shift to streptavidin (SA) biosensors (Sartorius) previously hydrated in 10X kinetics buffer for at least 10 min. Loaded tips were dipped into a solution containing 1 µM APN orthologues from *F. catus*, *C. familiaris*, *S. scrofa*, *G. gallus*, human, and the human APN(R741T) mutant in 10X kinetics buffer. For $K_{d,app}$ determination, loaded tips were dipped into

various concentrations of FcAPN-Fc or CfAPN-Fc for 300 s followed by a 300-s dissociation phase in 10X kinetics buffer. Data were baseline subtracted, and the plots were fitted using Sartorius analysis software (v.11.1). Data were plotted in Graphpad Prism 10. These experiments were done side-by-side with two different batches of B domain and APN orthologue preparations.

### Pull-down assays
To further characterize the interaction between CfAPN-Fc and FCoV-23 S-short and S-long, we performed pull-down assays. In brief, 100 µl (4 mg) magnetic beads, Dynabeads His-tag (Thermo Fisher) were washed twice with 200 µl 1× TBS (20 mM Tris-HCl pH 8.0 and 150 mM NaCl) using a magnetic stand (Thermo Fisher). Before the last wash, beads were divided into eight aliquots, after which the buffer was removed. Beads were coupled or not with 10 µg FCoV-23 S-short or S-long in TBS buffer. The bead–protein mixture was incubated at room temperature with gentle rotation. After 30 min, unbound proteins were removed and beads were washed three times with 30 µl TBS each time before resuspension in a solution containing 15 µg purified CfAPN-FC. After incubation for 1 h at room temperature with gentle rotation, beads were washed three times with TBS. The following samples were collected and characterized by SDS–PAGE: unbound proteins after coupling the spike proteins (UB$_1$) and after incubation with CfAPN (UB$_2$); the last washes after coupling the spike proteins (W$_1$) and after incubation with CfAPN (W$_2$); and bound proteins. Negative controls consisting of beads not coupled to FCoV-23 S-short or S-long were also analysed by SDS–PAGE.

### Bacterial protein expression and purification of NP components
The I53-50A and I53-50B proteins were expressed as previously described[67]. In brief, transformed Lemo21(DE3) (NEB) in LB broth (10 g tryptone, 5 g yeast extract and 10 g NaCl) were grown at 37 °C to an $OD_{600}$ of 0.8 with constant agitation. Protein expression was induced by adding 1 mM isopropil-β-D-1-thiogalactopyranoside and the temperature was reduced to 18 °C. Cells were collected after 16 h and lysed by microfluidization using a Microfluidics M110P at 18,000 p.s.i. in 50 mM Tris-HCl pH 8.0, 500 mM NaCl, 30 mM imidazole, 1 mM phenylmethylsulfonyl fluoride and 0.75% 3-[(3-cholamidopropyl) dimethylammonio]-1-propanesulfonate (CHAPS). Lysates were clarified by centrifugation at 24,000$g$ for 30 min and applied to a Ni Sepharose 6 fast flow column (Cytiva) for purification by immobilized metal affinity chromatography on an AKTA Avant150 FPLC system (Cytiva). Proteins were eluted with a linear gradient of 30 mM to 500 mM imidazole in 50 mM Tris-HCl pH 8.0, 500 mM NaCl and 0.75% CHAPS buffer. Peak fractions were pooled, concentrated in 10 K MWCO centrifugal filters (Millipore), sterile filtered (0.22 mm) and applied to either a Superdex 200 Increase 10/300 or HiLoad S200 pg GL SEC column (Cytiva) previously equilibrated in 50 mM Tris-HCl pH 8.0, 500 mM NaCl and 0.75% CHAPS buffer.

### In vitro NP assembly
Concentration of purified individual NP components was determined by measuring the absorbance at 280 nm and the corresponding calculated extinction coefficients. NPs were prepared by incubating D0-A-I53-50A, domain A-I53-50A or domain B-I53-50A trimers with pentameric I53-50B at molar ratios of 1.1:1, respectively, in 50 mM Tris-HCl pH 8.0 and 500 mM NaCl. Formation of CCoV-HuPn-2018 D0 NPs required mixing D0-I53-50A with I53-50A at a molar ratio of 1:6 with pentameric I53-50B. All in vitro assemblies were incubated at room temperature with gentle rocking for at least 20 min before subsequent purification by SEC on a Superose 6 column to remove residual unassembled components. Fractions were analysed by negative-stain electron microscopy and by SDS–PAGE. Assembled NPs eluted in the void volume of a Superose 6 column and were pooled and stored at 4 °C until use in haemagglutination assays.

### Negative-stain electron microscopy
NPs diluted to 0.01 mg ml⁻¹ in 50 mM Tris-HCl pH 8.0 and 150 mM NaCl were adsorbed to glow-discharged home-made carbon-coated copper grids for 30 s. The excess liquid was blotted away with filter paper (Whatman 1) and 3 µl of 2% w/v uranyl formate was applied to the grids. Finally, the stain was blotted away, and the grids were allowed to air dry for 1 min. Grids were imaged on a 120 kV FEI Tecnai G2 Spirit with a Gatan Ultrascan 4000 4k × 4k CCD camera at ×67,000 nominal magnification using a defocus ranging between 1.0 and 2.0 mm and a pixel size of 1.6 Å.

### Haemagglutination assay
The haemagglutination assay was done according to standard procedures. In brief, 25 µl CCoV-HuPn-2018 S D0, FCoV-23 S D0, S domain A and S domain B NPs at 125 µg ml⁻¹ were incubated with 25 µl 1% cat (Cornell University), chicken (Lampire), human (Rockland Immunochemicals), cow (Lampire) and turkey (Lampire) erythrocytes diluted in DPBS (Gibco) in V-bottom, 96-well plates (Greiner Bio-One) for 30 min at room temperature. Plates were then photographed and haemagglutination was analysed. CCoV-HuPn-2018 D0 NPs were used as a positive control.

### VSV pseudotyped virus production
FCoV-23, CCoV-HuPn, HCoV-229E and TGEV S pseudotyped VSV particles were generated as previously described[9]. In brief, HEK293T cells in DMEM supplemented with 10% FBS and 1% penicillin–streptomycin and seeded in poly-D-lysine-coated 10-cm dishes were transfected with a mixture of 24 µg of the corresponding plasmid encoding CCoV-HuPn S, TGEV S or HCoV-229 S using 60 µl Lipofectamine 2000 (Life Technologies) in 3 ml Opti-MEM following the manufacturer's instructions. After 5 h of incubation at 37 °C, DMEM supplemented with 20% FBS and 2% penicillin–streptomycin was added. The next day, cells were washed three times with DMEM and were transduced with VSVΔG-luc[66]. After 2 h, the virus inoculum was removed and cells were washed five times with DMEM before the addition of DMEM supplemented with anti-VSV-G antibody (I1-mouse hybridoma supernatant diluted 1:25 (v/v), from CRL-2700, ATCC) to minimize parental background. After 18–24 h, supernatants containing pseudotyped VSV were collected, centrifuged at 2,000g for 5 min to remove cellular debris, filtered through a 0.45 µm membrane, concentrated ten times using a 30 kDa cut-off membrane (Amicon), aliquoted and frozen at −80 °C.

### VSV pseudotyped virus infection and neutralization
For pseudotyped VSV infection and neutralization, HEK293T cells were transfected with plasmids encoding the different full-length APN orthologues following a previously described protocol[9]. In brief, HEK293T cells at 90% confluency and seeded in poly-D-lysine-coated 10-cm dishes were transfected with a mixture of Opti-MEM containing 8–10 µg of the corresponding plasmid encoding full-length APN orthologues using 30 µl Lipofectamine 2000 (Life Technologies) according to the manufacturer's instructions. After 5 h of incubation at 37 °C, cells were trypsinized, seeded into poly-D-lysine-coated clear-bottom white-walled 96-well plates at 50,000 cells per well and cultured overnight at 37 °C. For infections, 20 µl of the corresponding pseudotyped VSV diluted 1:20 was mixed with 20 µl DMEM, and the mixture was added to cells that were previously washed twice with DMEM. After 2 h of incubation at 37 °C, 40 µl DMEM was added and cells were further incubated overnight at 37 °C.

For neutralization assays, a single dilution of 1:10 for mouse serum samples or 11 3-fold serial dilutions of Fab 1AF10 or APN ectodomains were prepared in DMEM. Next, 20 µl FCoV-23 S, CCoV-HuPn-2018 S, HCoV-229E S or TGEV S pseudotyped VSV was added 1:1 (v/v) to each Fab 1AF10 or APN ectodomains, and mixtures were incubated for 45–60 min at 37 °C. After removing medium, transfected HEK293T cells were washed three times with DMEM, and 40 µl of the mixture containing pseudotyped VSV and Fab or APN ectodomains and sera were added. One hour later, 40 µl DMEM was added to the cells. After 17–24 h, 60 µl One-Glo-EX substrate (Promega) was added to each well and plates were placed in a shaker in the dark. After 5–15 min of incubation, plates were read on a Biotek plate reader. Relative luciferase units were plotted and normalized in Graphpad Prism 10. Cells alone without pseudotyped virus were defined as 0% infection, and cells with virus only (no sera) were defined as 100% infection.

### VSV pseudotyped virus entry assay with protease inhibitors
Camostat mesylate (100 mM), nafamostat mesylate (100 mM), E-64d (at 10 mM) and bafilomycin A1 (100 µM) (Sigma) inhibitors were dissolved in DMSO and frozen in aliquots. Fcwf-CU cells and CRFK were seeded into white-walled, clear-bottom 96-well plates (Corning) at a density of 40,000 cells and grown overnight at 37 °C and 5% CO₂. The next day, the growth medium was removed and cells were washed twice with DMEM (for CRFK cells) or EMEM−1% HEPES (for Fcwf-CU cells) before adding 50 µl of the respective medium containing 25 µM camostat mesylate, nafamostat mesylate or E-64d, 1 µM bafilomycin A1 or DMSO. After 2 h of incubation at 37 °C, protease inhibitors were removed, except for bafilomycin, which was kept at 1 µM throughout the whole experiment, and 25 µl VSV-pseudotyped viruses with FCoV-23 S-long or S-short diluted 1:20 were added to the cells. After 2 h, an equal volume of DMEM supplemented with 20% FBS (for CRFK cells) or EMEM−1% HEPES supplemented with 40% FBS (for Fcwf-CU cells) was added. After 20–24 h, ONE-Glo EX (Promega) was added to each well, and the cells were incubated for 5 min at 37 °C. Luminescence values were measured using a BioTek Synergy Neo2 plate reader. Luminescence readings were analysed using (GraphPad Prism 10). The relative luminescence units values recorded from cells infected in the presence of DMSO and values from cells infected in the presence of protease inhibitors were plotted and compared. Two biological replicates ($n = 2$) were each done with a different batch of VSV-pseudotyped virus and 10–12 technical replicates were done for each inhibitor.

### VSV pseudotyped virus entry kinetics
For pseudotyped VSV virus entry kinetics, FCoV-23 S-long and S-short were quantified in VSV pseudotyped viruses by western blot analysis to ensure infection with the same amount of S from both pseudotypes. Fcwf-CU and CRFK cells were seeded into white-walled, clear-bottom 96-well plates (Corning) at a density of 40,000 cells and grown overnight at 37 °C and 5% CO₂. The next day, the growth medium was removed and cells were washed twice with DMEM (for CRFK cells) or EMEM−1% HEPES (for Fcwf-CU cells) before adding 50 µl VSV pseudotyped viruses with FCoV-23 S-long or S-short. Luminescence values were measured at the indicated times using a BioTek Synergy Neo2 plate reader, and the readings were analysed using GraphPad Prism 10. The relative luminescence values recorded from cells infected with VSV pseudotyped with FCoV-23 S-long and from cells infected with VSV pseudotyped with FCoV-23 S-short were plotted and compared. Four biological independent experiments ($n = 4$) were each done with a different batch of VSV-pseudotyped virus and 12 technical replicates. Comparison of entry values between S-long and S-short for each time point was performed using multiple ratio paired $t$-tests.

### MLV pseudotyped virus production
MLV pseudotyped particles were produced as previously described[68]. In brief, HEK293T cells were seeded at $5 \times 10^5$ cells per ml in 6-well plates (Costar) a day before transfection. At about 60% confluency, cells were transfected with 800 ng pCMV-MLV gag-pol, 600 ng pTG-Luciferase and 600 ng pCDNA3.1(+) FCoV-23 S-short or pCDNA3.1(+) FCoV-23 S-long, pCAGGS empty vector as a negative control or pCAGGS-VSV-G as a positive control and incubated at 34 °C. Supernatants were collected 48 h after transfection and cell debris was removed by centrifugation

at 1,000*g*. The supernatant-containing particles were filtered through a 0.45 µm membrane, aliquoted and frozen at −80 °C.

## Pseudotyped MLV infection and entry kinetics

Fcwf-CU, CRFK, AK-D, A-72, MDCK, LLC-MK2, VeroE6, VeroE6-TMPRSS2, A549, Calu3, HBEC, LLC-PK1, MDBK, OFT, OVKID, L2, BHK 21 and DF-1 cells were seeded at $3-5 \times 10^5$ cells per ml in a 24-well plate (Costar) and cultured at 37 °C. At about 90–100% confluency, cells were infected with 200 µl MLV particles and incubated on a rocker at 37 °C. After 1.5 h, 300 µl of the corresponding complete medium was added to cells. After 72 h (Extended Data Fig. 8a) or 8 h (Extended Data Fig. 8c), cells were lysed with 100 µl of 1× luciferase cell culture lysis reagent (25 mM Tris-phosphate pH 7.8, 2 mM dithiothreitol (DTT), 2 mM EDTA, 10% glycerol and 1% Triton X-100, Promega), and luminescence was measured using a Luciferase Assay system (Promega). Luciferase activity was measured by adding 20 µl luciferin substrate to 10 µl cell lysate and read using a GloMax 20/20 luminometer (Promega). MLV infection assays were done in duplicate or triplicate for each of the $n = 2$ biological replicates. For the entry kinetics experiments, FCWF-CU, AK-D and A72 cells were seeded at $3 \times 10^5$ cells per ml in a 48-well plate (Costar) and cultured at 37 °C. At 100% confluency, cells were infected with 120 µl MLV particles and incubated on a rocker at 37 °C. Cells were lysed with 100 µl of 1× luciferase cell culture lysis reagent, and luminescence was measured by adding 20 µl luciferase substrate to 10 µl cell lysate at 8 h and 72 h after infection. The kinetic assay was performed with four biological independent experiments, each with three to four technical replicates. Comparison of entry values between S-long and S-short for each time point was performed using ratio paired *t*-tests. We note that for MLV pseudoviruses, expression of the luciferase reporter gene entails reverse transcription, genomic integration and forward transcription and is therefore more complex than for VSV pseudoviruses.

## Fluorogenic peptide cleavage assay

The FCoV-23 S$_2$′ fluorogenic peptide, $_{958}$SKRKYRSAIE$_{967}$, site was synthesized by Biomatik, with a 7-methoxycoumarin-4-yl acetyl group covalently bound at the N terminus end and a 2,4-dinitrophenyl group at the C-terminal end used as a substrate in a fluorogenic peptide cleavage assay performed as previously described[69]. In brief, FCoV-23 S$_2$′ peptide was resuspended in water to 1 mM and cleavage was performed in a black, flat-bottom 96-well plate (Costar) in a total volume of 100 µl per well as follows: 94.5 µl enzyme-specific buffer and 5 µl S$_2$′ fluorogenic peptide was added to each well (50 µM per well) with 0.5 µl of protease. TPCK trypsin (Thermo Fisher) was used at 4.3 nM per well in 67 mM NaH$_2$PO$_4$ pH 7.6. Furin (NEB) was used at 1 unit per well in 20 mM HEPES–NaOH pH 7.5, 0.2 mM CaCl$_2$ and 1 mM 2-mercaptoethanol. PC1 (NEB) was used at 1 U per well in 100 mM HEPES–NaOH pH 6.0, 1 mM CaCl$_2$ and 1 mM 2-mercaptoethanol. Xa (NEB) was used at 1 U per well in 20 mM Tris-HCl pH 8.0, 100 mM NaCl and 2 mM CaCl$_2$. Plasmin (Innovative Research) was used at 0.5 µg ml$^{-1}$ per well in 100 mM Tris-HCl pH 7.4, 0.1% Tween 20 and 0.1 mM EDTA. Cathepsin L (R&D) was used at 0.5 µg ml$^{-1}$ per well in 50 mM 2-(*N*-morpholino) ethanesulfonic acid and NaOH pH 6.0, 5 mM dithiothreitol, 1 mM EDTA and 0.005% Brij-35. After buffer, protease and peptide were mixed, fluorescence was measured every 60 s for 2 h at 37 °C using a SpectraMax Gemini XPS (Molecular Devices) at an excitation wavelength of 330 nm and an emission wavelength of 390 nm. Data were plotted as relative fluorescent units on the *y* axis and time on the *x* axis.

## Protease digestion experiments of VSV pseudotyped viruses

VSV pseudotyped with FCoV-23 S-short or S-long were thawed and with or without trypsin at 10 mg ml$^{-1}$ for 15 min at 37 °C. To analyse the influence of APN on trypsin digestion patterns, VSV pseudotyped with FCoV-23 S-short or S-long were incubated with either 6.2 µM recombinant *Cf*APN ectodomain or membrane-anchored *Cf*APN for 30 min at room temperature before the addition or not of trypsin at 10 mg ml$^{-1}$ for 15 min at 37 °C followed by treatment (or not) with proteinase K at 0.6 mg ml$^{-1}$ for 15 min at room temperature. 4× SDS–PAGE loading buffer was then added to all samples before boiling at 95 °C for 10 min. Samples were run on a 4–15% gradient Tris-glycine gel (Bio-Rad) and transferred to PVDF membranes. HA-tag polyclonal antibodies at 1:1,000 dilution (Proteintech) and Alexa Fluor 680-conjugated goat anti-rabbit at 1:50,000 dilution (Jackson Immuno Research) were used as primary and secondary antibodies, respectively. A LI-COR processor was used to develop the western blots.

## S-mediated cell–cell fusion assay

Membrane fusion kinetics experiments were conducted at 37 °C in an atmosphere of 5% CO$_2$, as previously described[70,71] but with some modifications. On day 1, BHK-21–GFP1–10 cells were seeded into 6-well plates at a density of $4 \times 10^5$ cells per well and BHK-21–GFP11 cells were seeded into poly-D-lysine-coated 10-cm dishes at $4-5 \times 10^6$ cells. On day 2, BHK-21–GFP1–10 cells were washed once with DMEM, placed in DMEM containing 10% FBS and 1% penicillin–streptomycin and transfected using Lipofectamine 2000 (Life Technologies) with 4 µg per well of plasmid encoding membrane-anchored FCoV-23 S-long or CCoV-HuPn-2018 S-long, or 1 µg per well of plasmid encoding for membrane-anchored FCoV-23 S-short or CCoV-HuPn-2018 S-short (to produce comparable expression levels between S-long and S-short). The same day, BHK-21–GFP11 cells were transfected with plasmid encoding the indicated membrane-anchored APN orthologue using a mixture of 8 µg DNA and 30 µl Lipofectamine 2000 (Life Technologies) per 10-cm dish according to the manufacturer's instructions. Alternatively, if A-72, CRFK or Fcwf-CU cells were used as target cells (instead of BHK-21–GFP11 cells), they were trypsinized before seeding at $5-6 \times 10^6$ cells per poly-D-lysine-coated 10-cm dish. After 2–3 h of incubation, A-72, CRFK and Fcwf-CU cells were transfected with a mixture of 24 µg DNA plasmid encoding pQCXIP-BSR-GFP11 and 60 µl Lipofectamine 2000 (Life Technologies) per 10-cm dish for the transient expression of GFP11. After 4–5 h of incubation, all target cells were trypsinized and seeded into a 96-well-glass bottom, black-walled plates (CellVis) at a density of 36,000 cells per well and cultured overnight. On day 3, S-expressing BHK-21–GFP1–10 cells were washed three times using FluoroBrite DMEM (Thermo Fisher), detached using an enzyme-free cell dissociation buffer (Gibco) and passed through a cell strainer to remove aggregates. BHK-21–GFP1–10 cells were subsequently counted, diluted to 60,000 cells per ml, 40,000 cells per ml or 120,000 cells per ml to seed on top of BHK-21-GFP11, Fcwf-Cu, and A-72 and CRFK cells, respectively, which were previously washed three times with FluoroBrite. Trypsin pretreatment of S-expressing cells was performed when applicable using 5–10 µg trypsin (Sigma) during 15 min of incubation at 37 °C, after which 10–15 µg soybean trypsin inhibitor (Sigma) was added before adding them on target cells. Cells were incubated at 37 °C and 5% CO$_2$ in a Cytation 7 plate Imager (Biotek), and both bright-field and GFP images were collected every 30 min for 18 h. Fusogenicity was assessed by quantifying the fraction of the imaged area with GFP fluorescence for each image using Gen5 Image Prime (v.3.11) software. AUCs were calculated using GraphPad Prism 10, and data were collected for 18 h. For Fcwf-CU cells, comparisons between S-long-mediated and S-short-mediated fusion was done until 5.5 h after adding S-expressing cells (after this length of time, FcwF-CU cells were detaching or clumping, which hindered fluorescence measurements). Statistical analysis was performed using the AUC of the GFP$^+$ area between the biological replicates with two-way ANOVA or one-way ANOVA with Tukey's multiple comparison test, as indicated in the relevant figure legends. S-transfected cells added to BHK-21–GFP11 target cells not expressing APN (no APN) or untransfected BHK-21–GFP1–10 cells (no S) added to target cells expressing GFP11 and APN were used as negative controls.

## Western blotting

For VSV pseudotypes characterization and quantification, 15 µl of each pseudotype was mixed with 4× SDS–PAGE loading buffer, run on a 4–15% gradient Tris-glycine gel (Bio-Rad) and transferred to a PVDF membrane using the protocol mix molecular weight of a Trans-Blot Turbo system (Bio-Rad). The membrane was blocked with 5% milk in TBS-T (20 mM Tris-HCl pH 8.0 and 150 mM NaCl) supplemented with 0.05% Tween-20 at room temperature and with agitation. After 1 h, the fusion-peptide-specific 76E1 monoclonal antibody was added at 2 µg ml$^{-1}$ and incubated overnight at 4 °C with agitation. The next day, the membrane was washed three times with TBS-T and an Alexa Fluor 680-conjugated goat anti-human secondary antibody (1:50,000 dilution, Jackson Immuno Research) was added and incubated for 1 h at room temperature. Alternatively, HA-tag polyclonal antibody at 1:1,000 (Proteintech) and anti-VSV M (23H12) monoclonal antibody (Kerafast) were used as the primary antibodies in combination with Alexa Fluor 680-conjugated goat anti-rabbit and Alexa Fluor 680-conjugated goat anti-mouse (each at 1:50,000 dilution, Jackson Immuno Research) as secondary antibodies, using the same protocol as described above. Membranes were washed three times with TBS-T, after which a LI-COR processor was used to develop the western blots.

For quantification of membrane-anchored S expressed in BHK-21–GFP1–10 for fusion assays, we used 76E1 monoclonal antibody and Alexa Fluor 680-conjugated goat anti-human as the primary and secondary antibodies, respectively.

For MLV pseudotypes quantification, 1.5 ml MLV particles were centrifuged using a SW55Ti rotor (Beckman) with an Optima XPN ultracentrifuge (Beckman) at 110,000$g$ for 1 h and 30 min. Pellets were resuspended in ice-cold PBS (Corning) and centrifuged again at 110,000$g$ for 1 h. Pellets were resuspended in 20 µl PBS (Corning), and NuPage LDS sample buffer (Thermo Fisher) and NuPage sample reducing agent (Thermo Fisher) were added to the resuspended sample. After boiling at 70 °C for 10 min, samples were run on a NuPage 4–12% gradient Bis-Tris gel (Thermo Fisher) and transferred to a PVDF membrane. The membrane was blocked for 30 min with 5% BSA (VWR) in TBS-T (20 mM Tris-HCl, pH 8.0 and 150 mM NaCl) supplemented with 0.05% Tween-20. After 30 min, HA monoclonal antibody (1:1,000 dilution, Thermo Fisher, 26183) and MLV p30 (4B2) monoclonal antibody (1:2,000 dilution, Abcam, ab130757) were added, and the membrane was incubated overnight at 4 °C with agitation. The membrane was washed three times with TBS-T and incubated with Alexa Fluor 488-conjugated goat anti-mouse secondary antibody (1:50,000 dilution, Thermo Fisher, A-11029) for 1 h at room temperature with agitation. The membrane was washed three times and imaged using ChemiDoc Imaging software (Bio-Rad). Spike and MLV p30 band intensity was calculated using Bio-Rad Image Lab software.

## Cryo-EM sample preparation, data collection and data processing for FCoV-23 S-short and S-long

A volume of 3 µl FCoV-23 S-short (without D0) at approximately 0.1–0.3 mg ml$^{-1}$ was loaded three times onto freshly glow-discharged NiTi grids covered with a thin layer of home-made continuous carbon before plunge-freezing using a vitrobot MarkIV (Thermo Fisher) with a blot force of −1 and 4.5 s blot time at 100% humidity and 21 °C. For FCoV-23 S-long (with D0), 3 µl of the sample at 0.15 mg ml$^{-1}$ was loaded onto lacey grids freshly glow-discharged and covered with a thin layer of home-made continuous carbon following the same protocol as for FCoV-23 S-short.

For FCoV-23 S-short and S-long, 11,208 and 13,099 videos were collected, respectively, with a defocus range between −0.8 and −2.0 µm. Both datasets were acquired using a FEI Titan Krios transmission electron microscope operated at 300 kV equipped with a Gatan K3 direct detector and a Gatan Quantum GIF energy filter, operated with a slit width of 20 eV. Automated data collection was carried out using Leginon software (v.3.5)[72] at a nominal magnification of ×105,000 corresponding to a pixel size of 0.843 Å. The dose rate was adjusted to 9 counts per pixel per s, and each video was acquired in counting mode fractionated in 100 frames of 40 ms. Video frame alignment, estimation of the microscope contrast-transfer function parameters, particle picking and extraction were carried out using Warp[73]. One round of reference-free 2D classification was performed using cryoSPARC (v.4.4.1)[74] with binned particles to select well-defined particle images. To further improve particle picking, we trained Topaz picker[75] on the Warp-picked particles on the selected classes after 2D classification. Topaz-picked particles were extracted and 2D classified using cryoSPARC (v.4.4.1). Topaz-duplicated picked particles were removed using 60 Å (for FCoV-23 S-short) or 90 Å (for FCoV-23 S-long) as a minimum distance cut-off. Initial model generation was done using ab initio reconstruction in cryoSPARC (v.4.4.1) and used as a reference for a non-uniform refinement[76] (NUR) (for FCoV-23 S-short) or heterogenous refinement followed by NUR (for FCoV-23 S with D0) in cryoSPARC (v.4.4.1), which enabled the identification of two conformations for the FCoV-23 S-long dataset. Particles were transferred from cryoSPARC (v.4.4.1) to Relion (v.5.0b)[77] using pyem[78] (https://github.com/asarnow/pyem) to be subjected to one round of 3D classification with 50 iterations, using the NUR map as a reference model (angular sampling 7.5° for 25 iterations and 1.8° with local search for 25 iterations) and without imposing symmetry. Selected particles were subjected to a NUR using cryoSPARC (v.4.4.1). Particles were subjected to Bayesian polishing[79] using Relion (v.5.0b), during which they were re-extracted with a box size of 512 pixels at a pixel size of 1 Å. Another round of 2D classification was performed in cryoSPARC (v.4.4.1), which was followed by a NUR that included per-particle defocus refinement.

To improve the density of D0 from the FCoV-23 S-long and S-short maps, particles were symmetry-expanded (for the FCoV-23 S-long map) or not (for the FCoV-23 S-short map) and subjected to Relion (v.5.0b) 3D focus classification without refining angles and shifts using soft masks encompassing D0 in the swung-out conformation from the FCoV-23 S-long map or D0 in the proximal conformation from the FCoV-23 S-short map. Local refinement and local resolution estimation were carried out using cryoSPARC. Reported resolutions are based on the gold-standard Fourier shell correlation using the 0.143 criterion[80], and Fourier shell correlation curves were corrected for the effects of soft masking by high-resolution noise substitution[81].

## Cryo-EM sample preparation, data collection and data processing for FCoV-23 RBD in complex with *Fc*APN

Purified *Fc*APN was incubated overnight at 4 °C with a molar excess of purified FCoV-23 RBD of 1:3 and the complex was purified by SEC. Next, 2.7 µl *Fc*APN–FCoV-23 RBD at 4 or 6 mg ml$^{-1}$ was applied for 15 s onto freshly glow-discharged UltrAuFoil R 2/2 grids with 0.3 µl of 30 or 60 mM of CHAPSO, respectively, before plunge-freezing using a vitrobot MarkIV (Thermo Fisher) with a blot force of −0 and 6 s blot time at 100% humidity and 21 °C. A total of 13,540 micrographs were collected using Leginon (v.3.5)[72], with a defocus range between −0.8 and −2.0 µm on the same microscope set up described for the FCoV-23 S glycoprotein ectodomain. Data processing was similar as described for the FCoV-23 S glycoprotein ectodomain, except that two cycles of Topaz picker[75] on the Warp-picked particles on the selected classes after 2D classification were performed.

## Cryo-EM model building and analysis

Model Angelo[82] was used to generate an initial model, and UCSF Chimera (v.1.8)[83] and Coot (v.0.9.8.8)[84] were used to manually build the model. The model was refined and rebuilt into the maps using Coot (v.0.9.8.8), Phenix (v.1.21)[85] and Rosetta (v.2021.07.61567)[86,87]. Model validation was done using Molprobity[88] and Privateer[89] from the CCP4i2 suite[90]. Figures were generated using UCSF ChimeraX[91]. Palmitoleic

acids resolved in our cryo-EM maps were built in the final models based on previous findings for PEDV[55]. We note that no density was present for palmitoleic acid molecules interacting with domain A and domain D of the protomer with D0 in the proximal conformation.

## Phylogenetic analysis

Structure-guided phylogenetic classification of alphacoronavirus and deltacoronavirus RBDs was carried out using FoldTree[92].

## In vivo immunogenicity study

Female BALB/cAnNHsd mice were purchased from Envigo (order code 047) at 7 weeks of age and were maintained in a specific-pathogen-free facility in the Department of Comparative Medicine at the University of Washington accredited by the Association for Assessment and Accreditation of Laboratory Animal Care. Animal experiments were conducted in accordance with the University of Washington's Institutional Animal Care and Use Committee under approved protocol 4470-01. For each immunization, low-endotoxin HCoV-229E S ectodomain was diluted to 100 μg ml$^{-1}$ in buffer and mixed 1:1 (v/v) with AddaVax adjuvant (InvivoGen vac-adx-10) to obtain a final dose of 5 μg of HCoV-229E S ectodomain per animal per injection. At 8 weeks of age, 10 mice per group were subcutaneously injected in the inguinal region with 100 μl immunogen at weeks 0, 3 and 6. Animals were bled by submental venous puncture at weeks 2 and 5, and terminal blood was collected at week 8. Whole blood was collected in BD microtainer collection tubes and rested at room temperature for 30 min for coagulation. Tubes were then centrifuged for 10 min at 2,000$g$ and serum was collected and stored at −80 °C until use.

## Reporting summary

Further information on research design is available in the Nature Portfolio Reporting Summary linked to this article.

## Data availability

All of the data are presented in the Article. Any additional information required to reanalyse the data reported is available upon request. The following accession codes for the structures presented in this work are available from the Electron Microscopy Databank (EMD) and Protein Databank (PDB): EMD-46714 and PDB 9DB3 for FCoV-23 S-long with swung-out D0 (global refinement); EMD-46739 and PDB 9DBZ for FCoV-23 S-long with mixed D0 conformations (global refinement); EMD-46716 and PDB 9DBE for the local refinement of D0 from the FCoV-23 S-long swung-out dataset; EMD-46710 and PDB 9DB1 for the local refinement of the proximal D0 from the FCoV-23 S-long with mixed D0 conformations; EMD-46709 and PDB 9DB0 for FCoV-23 S-short; and EMD-46708 and PDB 9DAZ for the *Fc*APN-bound FCoV-23 RBD structure. Source data are provided with this paper.

## Code availability

This Article does not report original code.

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

**Acknowledgements** This study was supported by the National Institute of Allergy and Infectious Diseases (P01AI167966 to N.P.K., J.D.B. and D.V.; DP1AI158186 and 75N93022C00036 to D.V.), Investigators in the Pathogenesis of Infectious Disease Awards from the Burroughs Wellcome Fund (to D.V.), the University of Washington Arnold and Mabel Beckman Cryo-EM Center, the Morris Animal Foundation (G.R.W.), EveryCat Health Foundation (to G.R.W.) and the Cornell Feline Health Center (G.R.W.). J.D.B. and D.V. are Investigators of the Howard Hughes Medical Institute and D.V. is the Hans Neurath Endowed Chair in Biochemistry at the University of Washington. C.T.-B. is supported by the BBSRC Institute Strategic Programme grant funding to the Roslin Institute (grant numbers BBS/E/D/20241866, BBS/E/D/20002172 and BBS/E/D/20002174). A.C. is supported by a Liz Hanson Scholarship in Feline Medicine at Cornell University College of Veterinary Medicine.

**Author contributions** M.A.T. and D.V. designed the study and the experiments. M.A.T. designed the constructs used in this study. M.A.T., C.A.G., J.T.B., C.S. and A.J. recombinantly expressed and purified glycoproteins. C.A.G. performed the BLI binding assays. M.A.T. carried out the VSV pseudovirus entry assays, the pseudovirus neutralization assays and entry assays in the presence of protease inhibitors, the fusion experiments and protease cleavage of VSV membrane-anchored spike proteins, and cryo-EM specimen preparation, data collection and processing. M.A.T. and D.V. built and refined atomic models. A.C. performed fluorogenic cleavage assays and MLV pseudovirus entry assays under the supervision of G.R.W. M.A.T. and A.C. performed western blotting. E.M.L., I.W. and C.T. conducted the immunogenicity study under the supervision of N.P.K. S.H., J.D.B. and C.T.-B. contributed unique reagents or data. M.A.T., A.C., C.A.G., J.L., G.R.W. and D.V. analysed the data. M.A.T. and D.V. wrote the manuscript with input from all authors. D.V. supervised the project.

**Competing interests** N.P.K. and D.V. are named as inventors on patents for coronavirus NP vaccines filed by the University of Washington. N.P.K. is a paid consultant of Icosavax and has received unrelated sponsored research agreements from Pfizer and GlaxoSmithKline (GSK). J.D.B. is on the scientific advisory boards of Apriori Bio, Invivyd and the Vaccine Company. J.D.B. consults for GSK.

**Additional information**
**Correspondence and requests for materials** should be addressed to David Veesler.

Extended Data Fig. 1 | Sequence alignment of the FCoV-23 S-long and S-short constructs used in this study. The S-short construct corresponds to the C10-DA isolate[13].

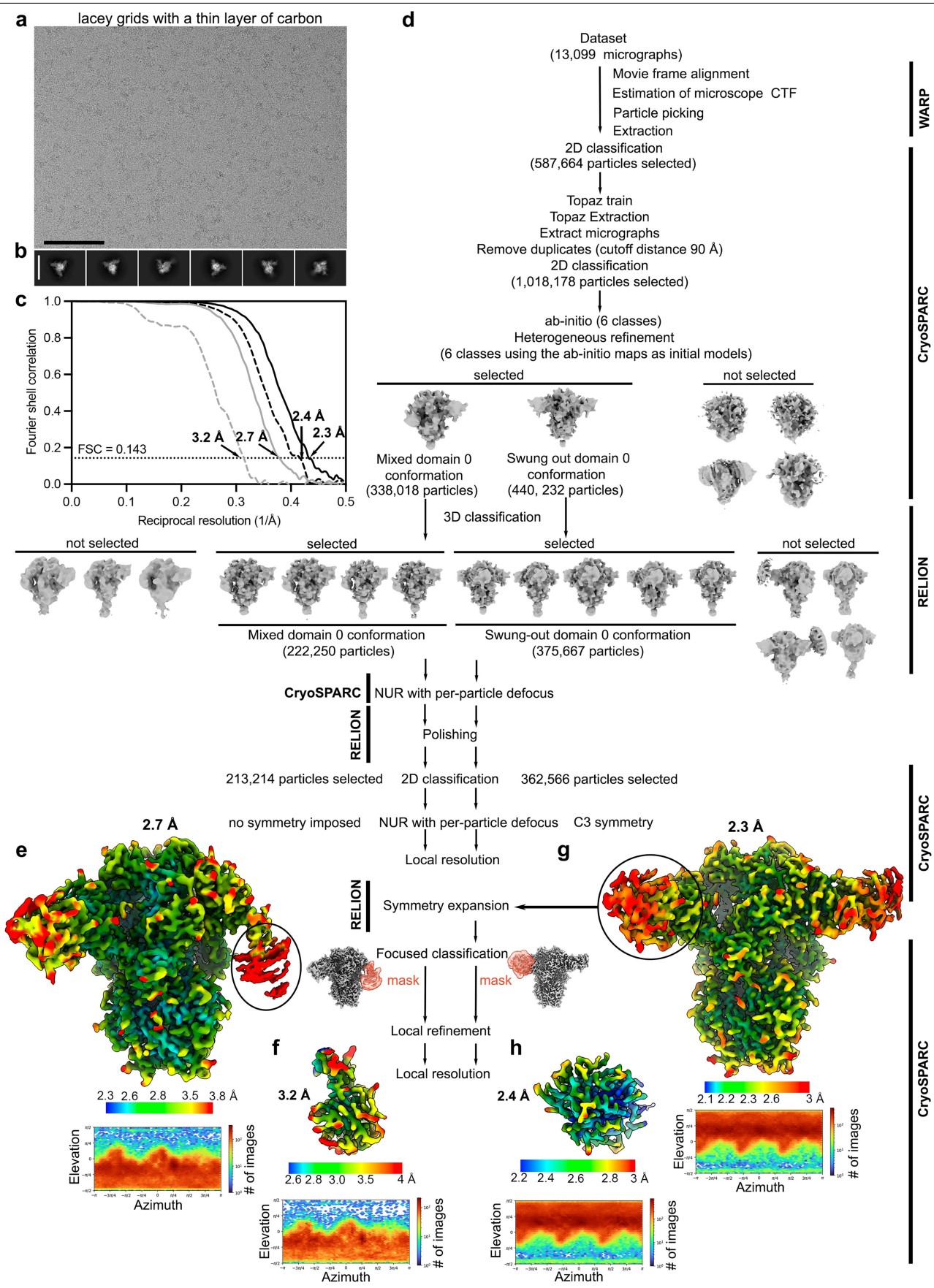

**Extended Data Fig. 2 |** See next page for caption.

**Extended Data Fig. 2 | Data processing and validation of the FCoV-23 S-long glycoprotein (with D0) dataset. a,b**, Representative electron micrograph out of 13,099 (**a**) and 2D class averages (**b**) of the prefusion FCoV-23 S-long ectodomain trimer embedded in vitreous ice. Scale bars of the micrograph and the 2D class averages are 100 nm and 220 Å, respectively. **c**, Gold-standard Fourier shell correlation (FSC) curves for the FCoV-23 S-long trimer with D0 in the "swung out" conformation (black line) or with mixed D0 conformations (gray line), and for the local refinements of D0 in swung out conformation (dashed black line) and in the proximal conformation (dashed gray line). The 0.143 cutoff is indicated by a horizontal dotted line. **d**, Cryo-EM data processing flowchart. **e,f**, Unsharpened maps of the reconstructions of the FCoV-23 S-long trimer with mixed D0 conformations (**e**) and the locally refined proximal D0 (**f**) colored according to local resolution determined using cryoSPARC. **g,h**, Unsharpened maps of the reconstructions of the FCoV-23 S-long trimer with D0 in swung out conformation (**g**) and the locally refined swung out D0 (**h**) colored according to local resolution determined using cryoSPARC. Angular distribution plots with all the particles contributing to the final maps are shown at the bottom of each corresponding map. CTF: contrast transfer function. NUR: non uniform refinement.

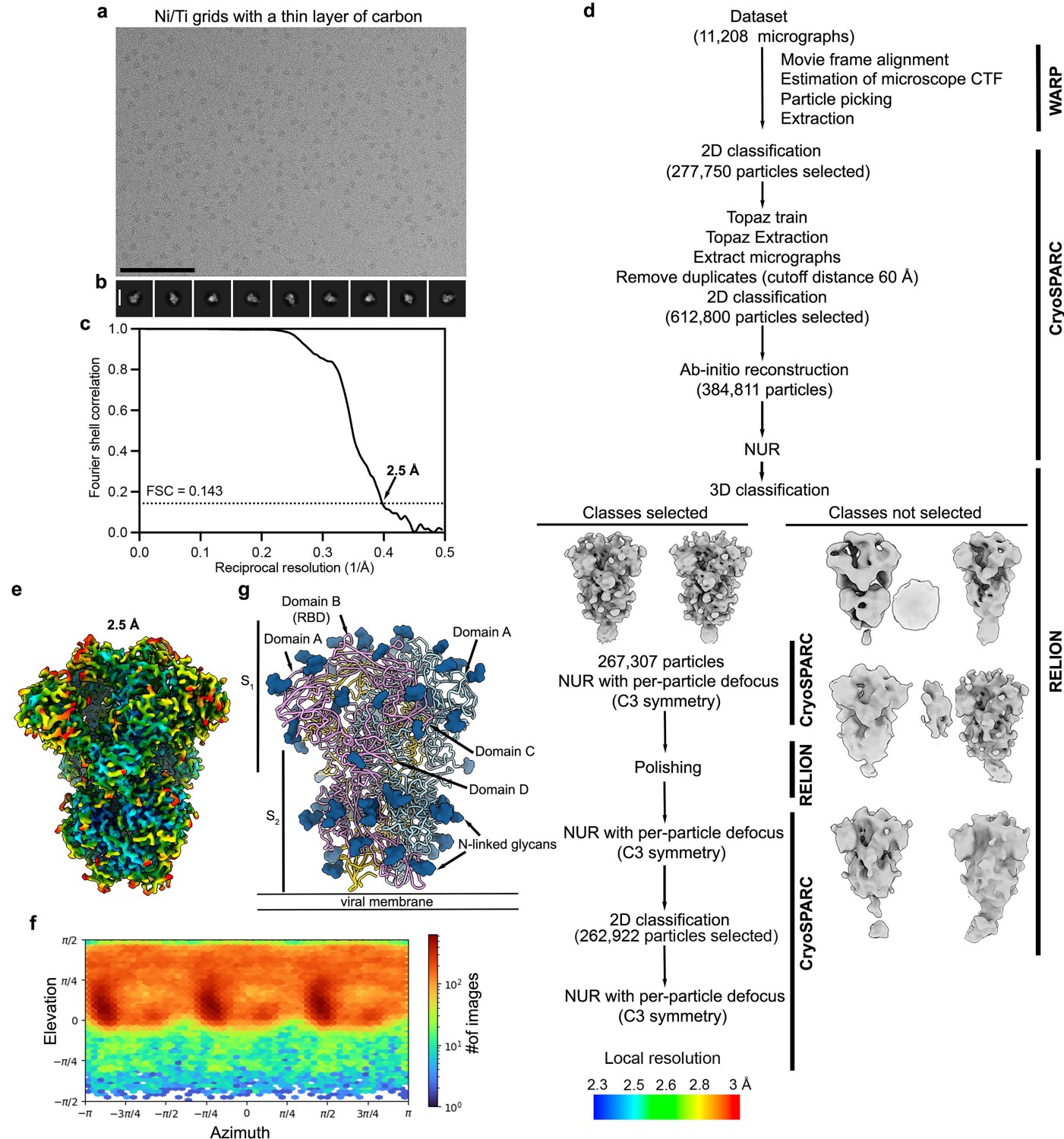

**Extended Data Fig. 3 | Data processing and validation of the FCoV-23 S-short glycoprotein (without D0) dataset. a,b,** Representative electron micrograph out of 11,208 (**a**) and 2D class averages (**b**) of the prefusion FCoV-23 S-short (without D0) ectodomain trimer embedded in vitreous ice. Scale bars of the micrograph and the 2D class averages are 100 nm and 180 Å, respectively. **c,** Gold-standard Fourier shell correlation curve for the FCoV-23 S-short trimer reconstruction. The 0.143 cutoff is indicated by a horizontal dotted line.

**d**, Cryo-EM data processing flowchart. **e**, Unsharpened map of the reconstruction of the FCoV-23 S-short ectodomain trimer colored according to local resolution determined using cryoSPARC. **f**, Angular distribution plot of all the particles that contributed to the map. **g**, Ribbon representation of FCoV-23 S-short highlighting the $S_1$ and $S_2$ subunits. N-linked glycans are shown in surface representation colored dark blue. CTF: contrast transfer function. NUR: non uniform refinement.

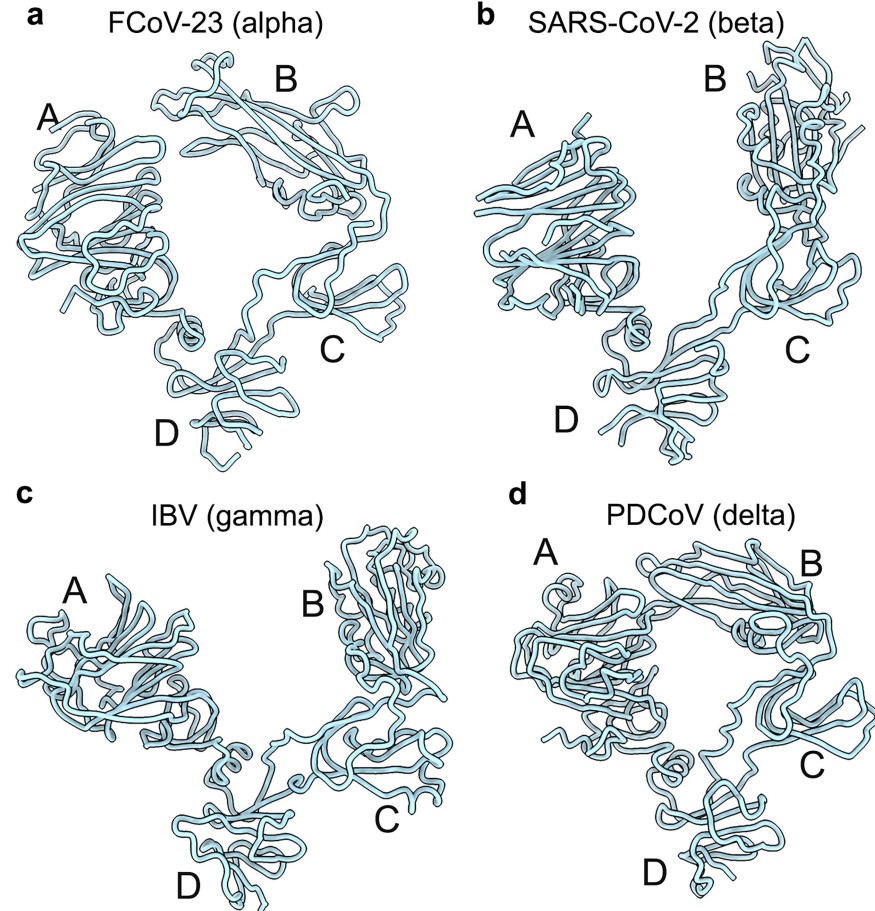

**a** FCoV-23 (alpha)

**b** SARS-CoV-2 (beta)

**c** IBV (gamma)

**d** PDCoV (delta)

**Extended Data Fig. 4 | S₁ subunit architectures of one representative S glycoprotein from the alpha, beta, gamma and delta coronavirus genera. a-d**, Ribbon diagrams of the FCoV-23 S-short (**a**), SARS-CoV-2, PDB 6VXX (**b**), IBV, PDB 6CV0 (**c**) and PDCoV, PDB 6BFU (**d**) S₁ subunits. A, B, C and D domains are labeled in each panel.

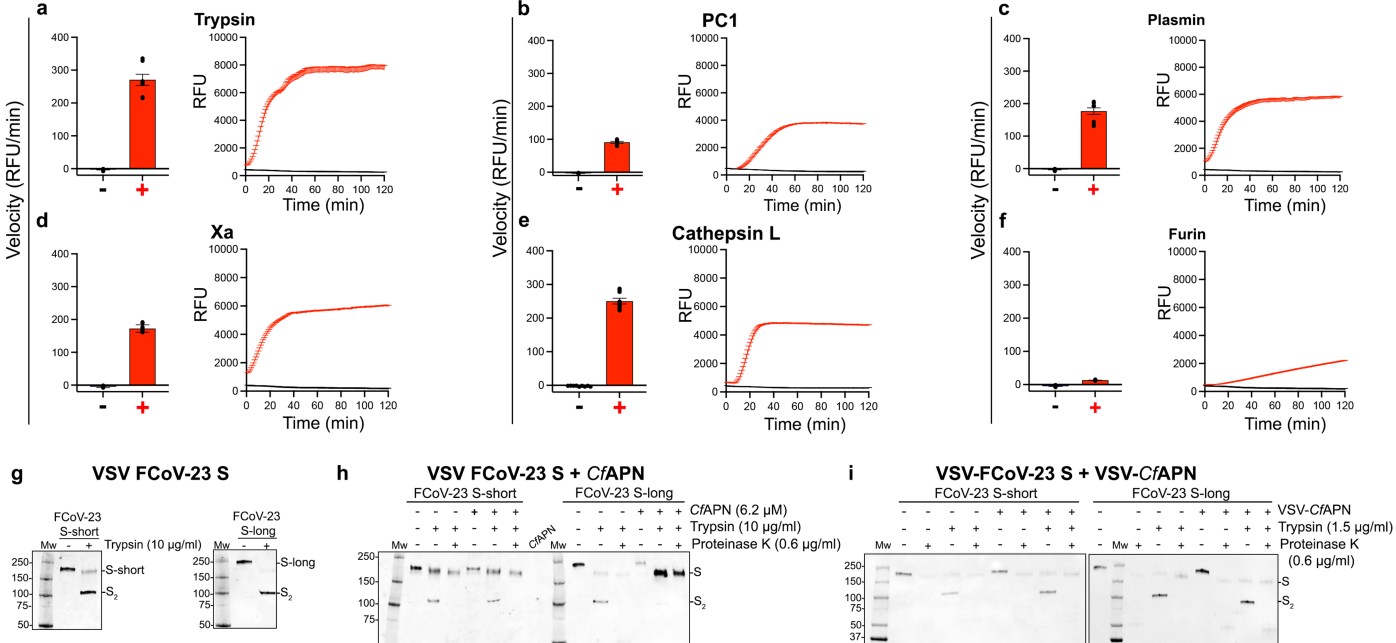

**g**  VSV FCoV-23 S

**h**  VSV FCoV-23 S + *Cf*APN

**i**  VSV-FCoV-23 S + VSV-*Cf*APN

**Extended Data Fig. 5 | Sensitivity of FCoV-23 S to host proteases.**
**a-f**, Quantification of the fluorogenic FCoV-23 S$_2'$ peptide cleavage by the indicated panel of proteases. The S$_2'$ peptide was coupled to a fluorescence resonance energy transfer pair of fluorophores: 7-methoxycoumarin-4-yl acetyl (MCA) at the N-terminus and N-2,4-dinitrophenyl (DNP) at the C-terminus yielding MCA-S$_{958}$KRKYRSAIE$_{967}$-DNP. Data points represent nine technical replicates from n = 1 biological replicate. Means and SEM are shown as bars and lines, respectively. RFU: relative fluorescence units. **g-i**, Western blot evaluation of proteolytic sensitivity of FCoV-23 S-short and S-long anchored in the membrane of VSV pseudoviruses upon incubation with trypsin alone (**g**), with the recombinant CfAPN ectodomain dimer (**h**) or anchored in the membrane of VSV pseudoparticles (**i**) followed by trypsin and proteinase K digestion. Anti-HA tag polyclonal antibody and Alexa Fluor 680-conjugated goat anti-rabbit were used as primary and secondary antibodies, respectively. Each western blot corresponds to one representative experiment out of two biological replicates.

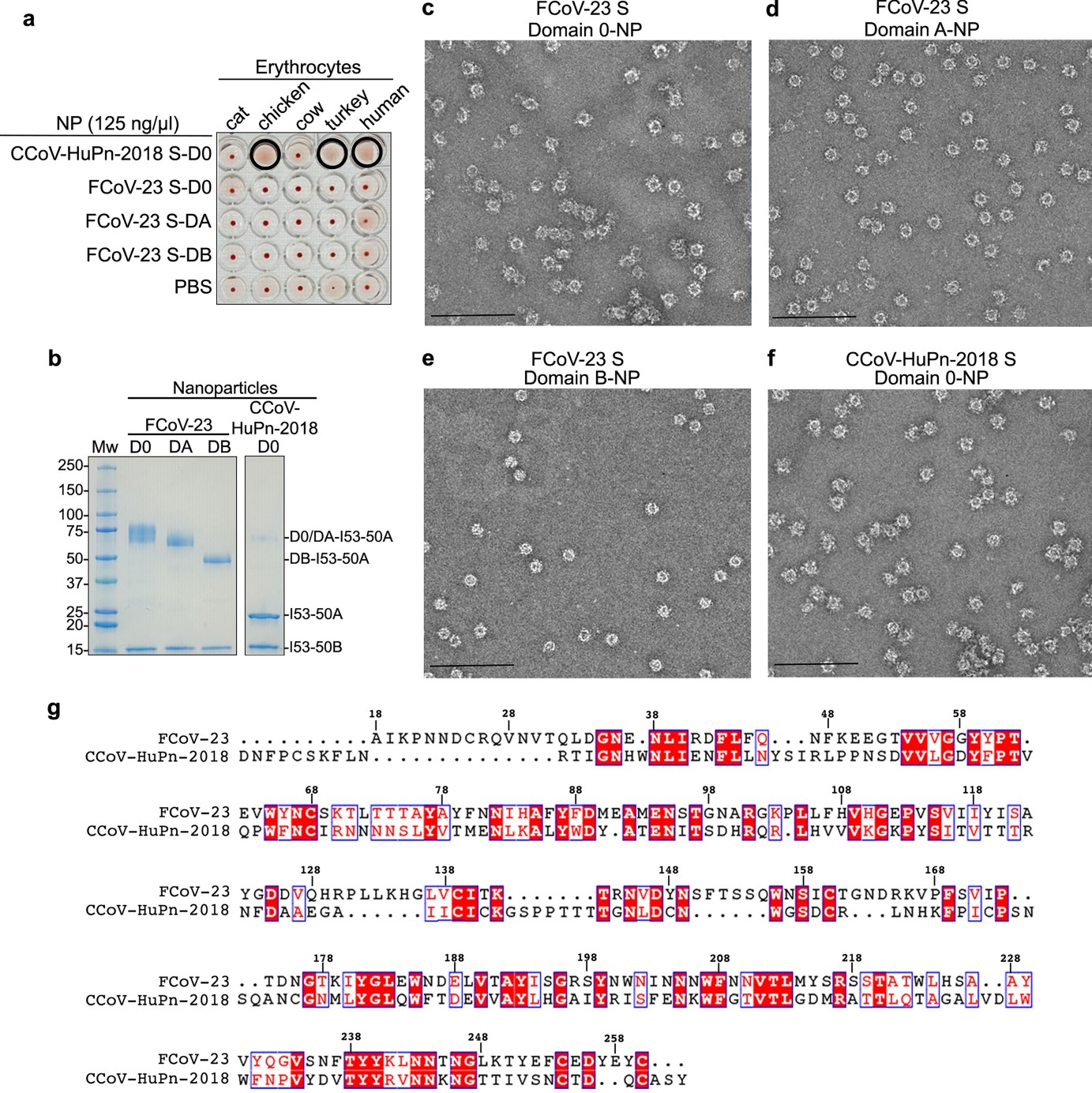

**Extended Data Fig. 6 | FCoV-23 S D0 does not hemagglutinate erythrocytes.** I53-50 nanoparticles (NP) were prepared by incubation of D0-I53-50A (FCoV-23 S-D0), domain A-I53-50A (FCoV-23 S-DA), or domain B-I53-50A (FCoV-23 S-DB) trimers with pentameric I53-50B at a 1:1 molar ratio, for at least 30 min at room temperature. Assembly of the CCoV-HuPn-2018 D0-I53-50 NP (CCoV-HuPn-2018 S-D0) required mixing domain D0-I53-50A and bare I53-50A before adding pentameric I53-50B at 6 (1:5):1 molar ratio. **a**, Hemagglutination assay. Wells positive for hemagglutination are circled. **b**, Representative SDS-PAGE analysis of purified nanoparticles out of 2 biological replicates (defined as distinct batches of purified proteins). **c-f**, Representative electron micrographs of negatively stained FCoV-23 S D0- (**c**), DA- (**d**), DB-I53-50 NPs (**e**), and CCoV-HuPn-2018 S D0-I53-50 NP (**f**). Scale bars: 200 nm. One hemagglutination assay corresponding to one preparation of nanoparticles is shown out of n = 2 biological replicates. **g**, Structure-based sequence alignment between FCoV-23 S D0 and CCoV-HuPn-2018 S D0.

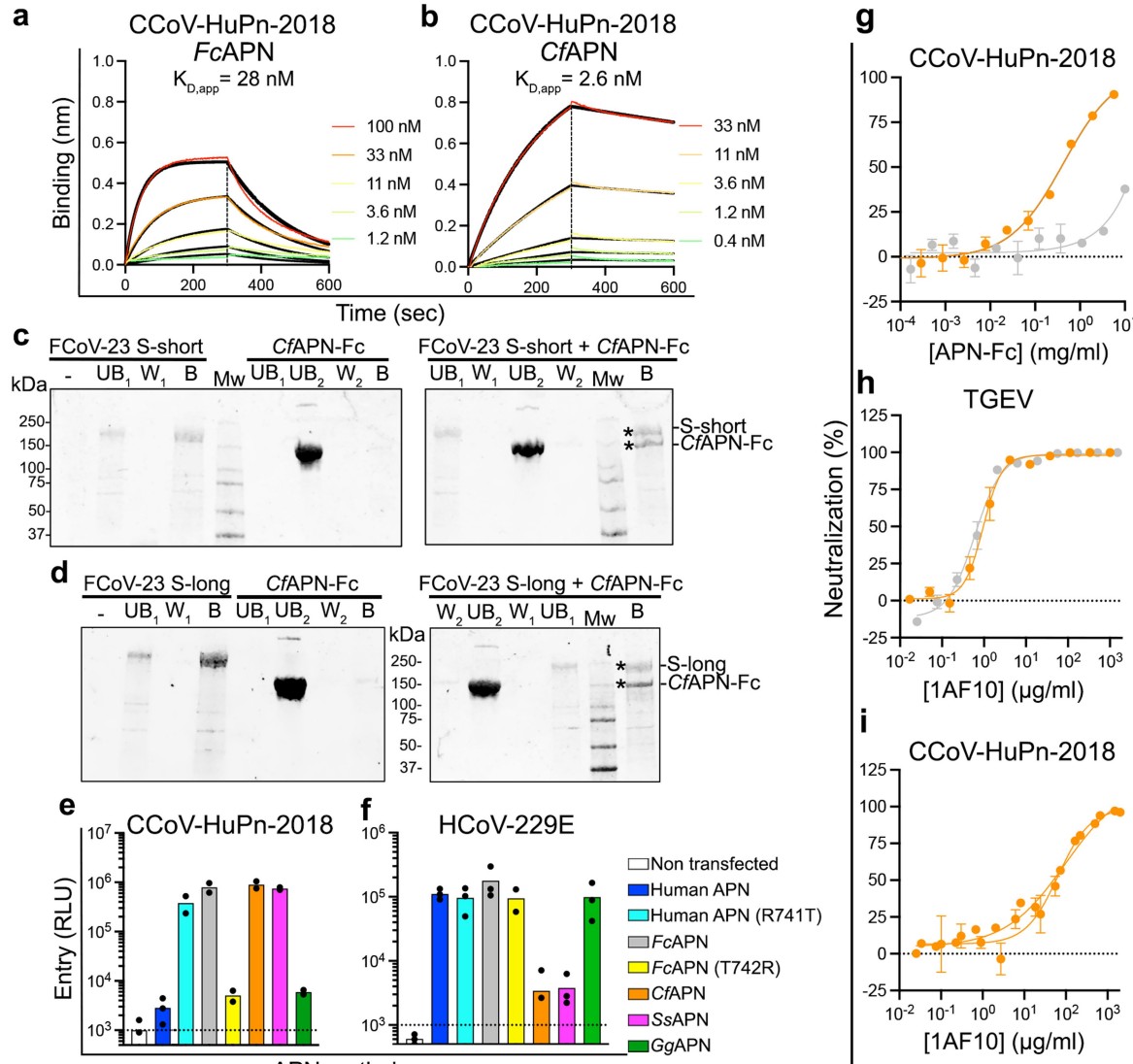

**Extended Data Fig. 7 | Characterization of receptor binding and entry of FCoV-23, CCoV-HuPn-2018 and HCoV-229E. a,b**, BLI binding kinetic analysis of dimeric FcAPN- (**a**) and CfAPN- (**b**) Fc ectodomains to immobilized biotinylated CCoV-HuPn-2018 RBD. Global fit (1:1 model) to the data is shown in black and reported affinities are expressed as apparent $K_D$ ($K_{D, app}$) due to avidity resulting from the dimeric nature of APN. A representative experiment is shown out of two biological replicates. **c-d**, Coomassie-stained SDS-PAGE analysis of CfAPN-Fc pull-down experiments using either FCoV-23 S-short (**c**) or S-long (**d**) immobilized on NTA-cobalt based magnetic beads. Asterisks indicate the positions of the pulled down APN/S complexes. $UB_1$: unbound fraction after incubating with S or buffer, $UB_2$: unbound fraction after incubation with CfAPN-Fc or buffer; $W_1$: final wash after incubating with S or buffer, $W_2$: final wash after incubating with CfAPN-Fc; B, bound. Purified S coupled to beads and not incubated with CfAPN-Fc and CfAPN-Fc incubated with uncoupled beads are shown on the gels at the left-hand side of panels **c** and **d**. One pull down experiment corresponding to one preparation of proteins is shown out of two independent biological replicates. **e-f**, Entry of VSV particles pseudotyped with CCoV-HuPn-2018 S (**e**) or HCoV-229E S (**f**) into HEK293T cells

transiently transfected with membrane-anchored human APN, human APN R741 (glycan knockin), FcAPN, FcAPN T742R (glycan knockout), Cf, Ss and Gg APN orthologues. RLUs, relative luciferase units. Each data point represents a biological replicate performed with technical duplicates. Bars represent the mean of 2-3 biological replicates. **g**, Concentration-dependent inhibition mediated by purified dimeric FcAPN- (gray) and CfAPN- (orange) Fc ectodomains of CCoV-HuPn-2018 S VSV pseudotyped virus entry into HEK293T cells transiently transfected with membrane-anchored CfAPN. Each curve corresponds to a single biological replicate performed with technical duplicates. The error bars represent the SEM of the technical duplicates. **h-i**, Dose-dependent neutralization of TGEV S (**h**) and CCoV-HuPn-2018 S (**i**) VSV pseudoviruses in the presence of various concentrations of the 1AF10 neutralizing monoclonal Fab fragment using HEK293T cells transiently transfected with membrane-anchored CfAPN (orange) or FcAPN (gray). Each curve represents a biological replicate (n = 1 in panel h and n = 2 in panel i, using distinct pseudovirus batches) performed with technical duplicates. The error bars represent the SEM of the technical duplicates.

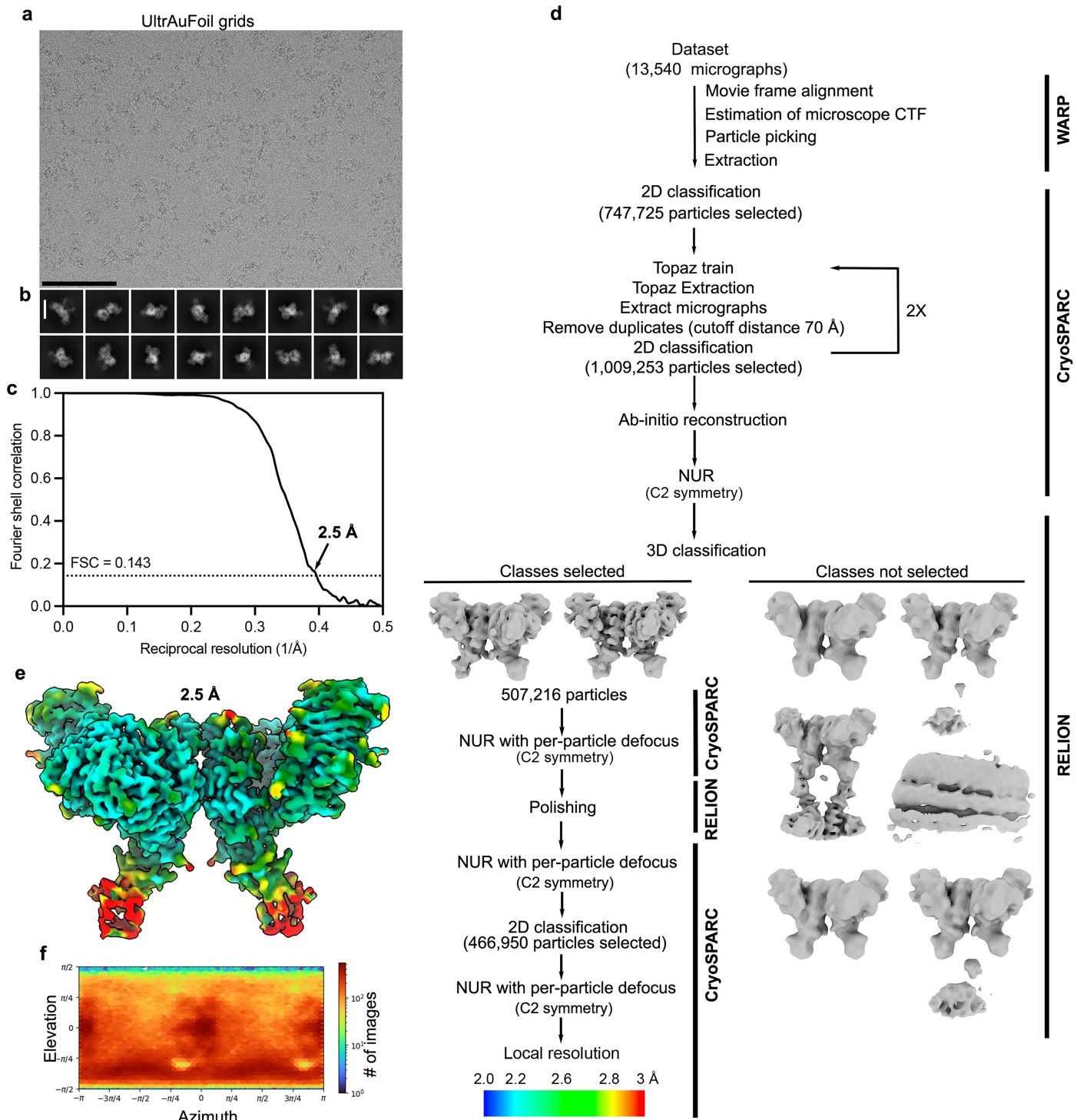

**Extended Data Fig. 8 | Data processing and validation of the FCoV-23 S RBD in complex with F.cat APN. a,b**, Representative electron micrograph out of 13,540 (**a**) and 2D class averages (**b**) of the FCoV-23 S RBD in complex with dimeric FcAPN embedded in vitreous ice. Scale bars of the micrograph and the 2D class averages are 100 nm and 150 Å, respectively. **c**, Gold-standard Fourier shell correlation curve for the FcAPN-bound FCoV-23 RBD (solid black line).

The 0.143 cutoff is indicated by a horizontal dotted line. **d**, Cryo-EM data processing flowchart. **e**, Unsharpened map of the FcAPN-bound FCoV-23 RBD colored according to local resolution determined using cryoSPARC. **f**, Angular distribution plots with all the particles contributing to the final map. CTF: contrast transfer function. NUR: non uniform refinement.

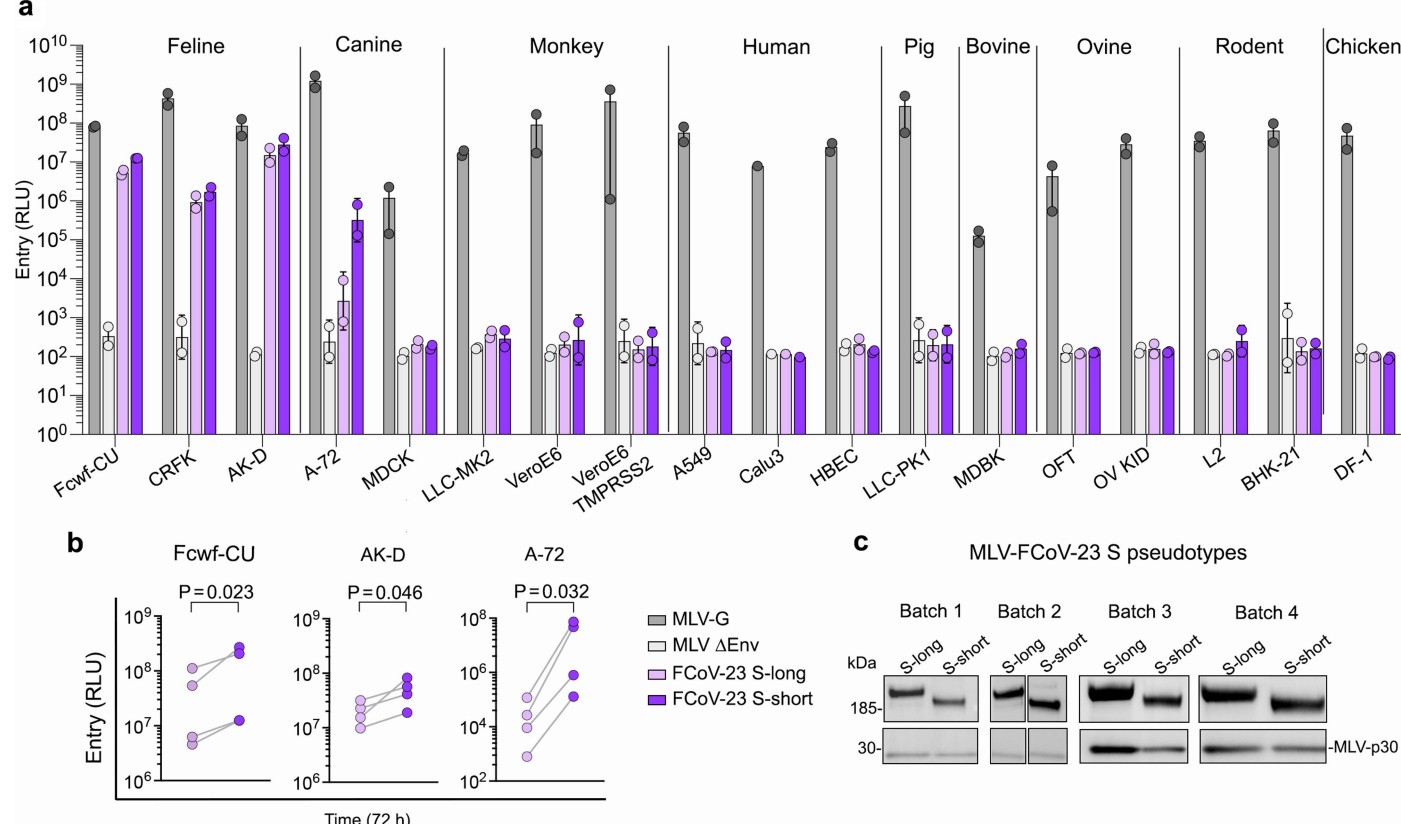

**Extended Data Fig. 9 | FCoV-23 S-long and S-short-mediated entry into cell lines spanning various species and tissues. a,** MLV particles (lacking the envelope glycoprotein) were pseudotyped with FCoV-23 S-long or S-short and entry into cells was assessed 72 h post transduction of Fcwf-CU (feline macrophage-like cells), CRFK (feline epithelial kidney cells), AK-D (feline airway epithelial cells), A-72 (canine tumor fibroblast cells), MDCK (canine epithelial kidney cells), LLC-MK2 (Rhesus monkey kidney cells), Vero-E6 (African green monkey kidney cells), Vero-E6-TMPRSS2 (VeroE6 cells with stable TMPRSS2 expression), A549 (human lung carcinoma epithelial cells), Calu3 (human lung adenocarcinoma epithelial cells), HBEC (human bronchial epithelial cells), LLC-PK1 (pig epithelial kidney cells), MDBK (bovine epithelial kidney cells). OFT (ovine fetal turbinate cells, sheep), OV KID (ovine kidney cells), L2 (rat lung epithelial cells), BHK-21 (Syrian golden hamster kidney cells) and DF-1 (chicken embryo fibroblast cells). MLV-G: MLV particles pseudotyped with the VSV G glycoprotein were used as a positive control. MLV ΔEnv: MLV particles with no envelope were used as negative control. RLUs: Relative luciferase units. Each dot represents a biological replicate (n = 2, using distinct pseudovirus batches) each carried out with 2-3 technical replicates, except for Calu3 cells for which n = 1 biological experiment was performed. Bars represent the mean and SEM of the two biological replicates. **b,** MLV particles (lacking the envelope glycoprotein) were pseudotyped with FCoV-23 S-long or S-short and entry into Fcwf-CU, AK-D and CRFK cells was assessed 72 h post transduction. Each dot represents a biological replicate (n = 4, using distinct pseudovirus batches, including the 2 biological replicates shown in panel a) each comprising 3 to 4 technical replicates. Statistical analysis was performed using ratio paired t-test. **c,** Western blot quantification of FCoV-23 S-short and S-long incorporated in the four batches of MLV-FCoV23 S pseudotypes used in panels a-b and in Fig. 4g. Anti-HA and anti-MLV p30 were used as primary antibodies and Alexa Fluor 488-conjugated goat anti-mouse as a secondary antibody.

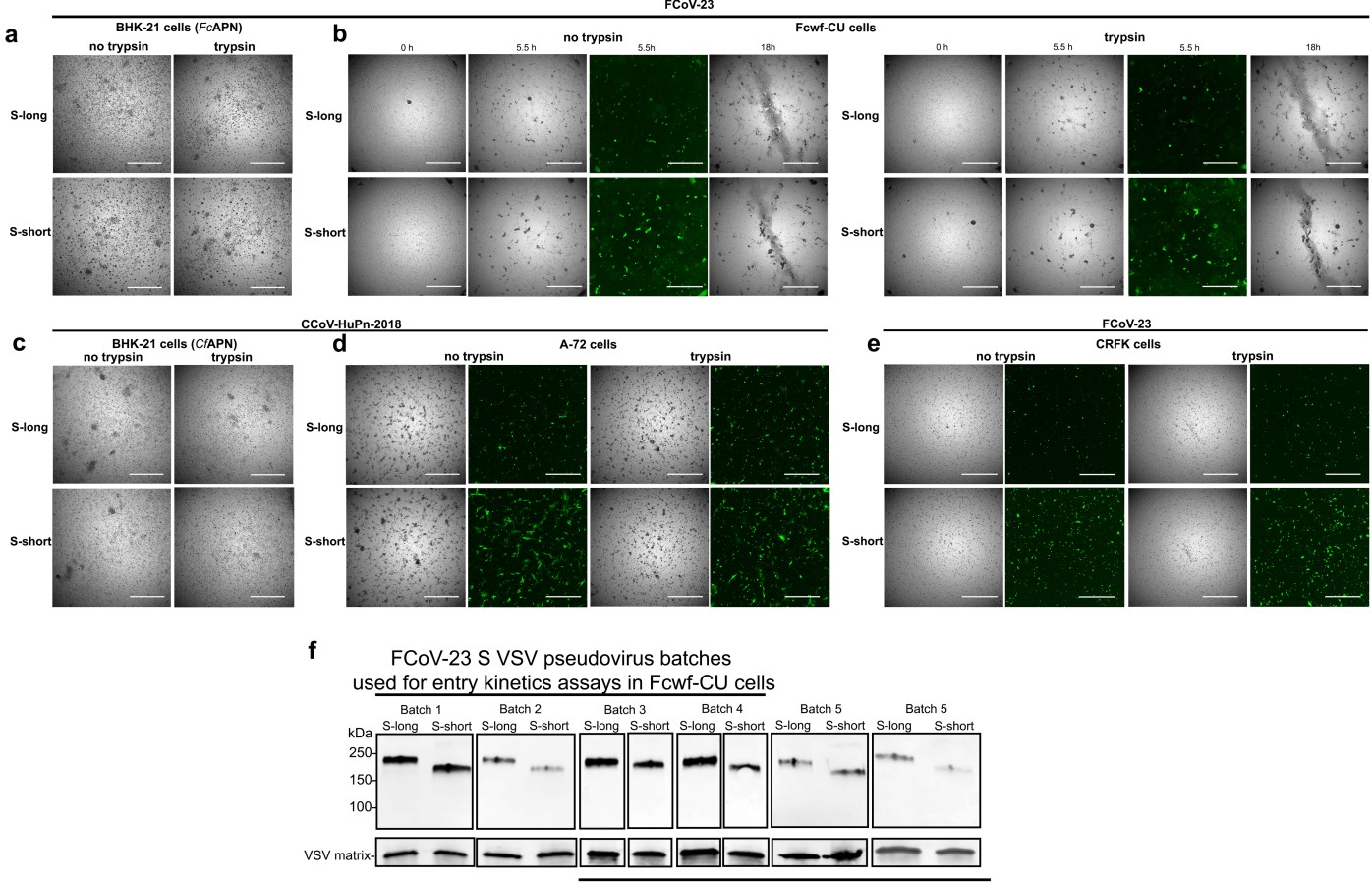

**Extended Data Fig. 10 | Evaluation of cell-cell fusion and pseudovirus entry kinetics. a, c,** Representative bright field images of BHK-21-GFP₁₁ cells transfected with FcAPN (**a**, n = 3 with 2-4 technical replicates each) or with CfAPN (**c**, n = 2 with 2-3 technical replicates each) at 18 h post addition of FCoV-23 S- (**a**) or CCoV-HuPn-2018 S- (**c**) expressing BHK-21-GFP₁₋₁₀ cells pretreated or not with trypsin. **b,** Representative bright field and fluorescent images of Fcwf-CU cells endogenously expressing FcAPN and transiently transfected with GFP₁₁ at various time points post addition of BHK-21-GFP₁₋₁₀ effector cells expressing FCoV-23 S-long or S-short treated with trypsin (n = 1 with 3 technical replicates) or not (n = 2 with 5 replicates each). Images acquired at 0 h, 5.5 h and 18 h post addition of S-expressing cells are shown to indicate the detaching/clumping of the cells. **d,** Representative bright field and fluorescent images of A-72 cells endogenously expressing CfAPN and transiently transfected with GFP₁₁ at 18 h post addition of BHK-21-GFP₁₋₁₀ effector cells expressing CCoV-HuPn-2018 S-long or S-short treated with trypsin (n = 1 with 5 technical replicates) or not (n = 4 with 5-10 technical replicates on each). **e,** Representative bright field and fluorescent images of CRFK cells endogenously expressing FcAPN and transiently transfected with GFP₁₁ at 18 h post addition of BHK-21-GFP₁₋₁₀ effector cells expressing FCoV-23 S-long or S-short treated with trypsin (n = 1 with 5 technical replicates) or not (n = 3 with 5-10 technical replicates each). **f,** Western blot quantification of FCoV-23 S-long and S-short incorporated into each VSV pseudotyped virus batch used in the entry kinetics experiments shown in Fig. 4f. Anti-HA-tag and anti-VSV M (23H12) were used as primary antibodies and Alexa Fluor 680-conjugated goat anti-rabbit and goat anti-mouse were used as secondary antibodies. Lines indicate the pseudoviruses batches used for the indicated cell type kinetics.

----------------

# Reporting Summary

## Statistics

For all statistical analyses, confirm that the following items are present in the figure legend, table legend, main text, or Methods section.

| n/a | Confirmed | |
|---|---|---|
| ☐ | ☒ | The exact sample size (*n*) for each experimental group/condition, given as a discrete number and unit of measurement |
| ☐ | ☒ | A statement on whether measurements were taken from distinct samples or whether the same sample was measured repeatedly |
| ☐ | ☒ | The statistical test(s) used AND whether they are one- or two-sided *Only common tests should be described solely by name; describe more complex techniques in the Methods section.* |
| ☒ | ☐ | A description of all covariates tested |
| ☒ | ☐ | A description of any assumptions or corrections, such as tests of normality and adjustment for multiple comparisons |
| ☐ | ☒ | A full description of the statistical parameters including central tendency (e.g. means) or other basic estimates (e.g. regression coefficient) AND variation (e.g. standard deviation) or associated estimates of uncertainty (e.g. confidence intervals) |
| ☒ | ☐ | For null hypothesis testing, the test statistic (e.g. *F*, *t*, *r*) with confidence intervals, effect sizes, degrees of freedom and *P* value noted *Give P values as exact values whenever suitable.* |
| ☒ | ☐ | For Bayesian analysis, information on the choice of priors and Markov chain Monte Carlo settings |
| ☒ | ☐ | For hierarchical and complex designs, identification of the appropriate level for tests and full reporting of outcomes |
| ☒ | ☐ | Estimates of effect sizes (e.g. Cohen's *d*, Pearson's *r*), indicating how they were calculated |

*Our web collection on statistics for biologists contains articles on many of the points above.*

## Software and code

Policy information about availability of computer code

| Data collection | Datasets were acquired using a FEI Titan Krios transmission electron microscope operated at 300 kV equipped with a Gatan K3 direct detector and a Gatan Quantum GIF energy filter, operated with a slit width of 20eV. Automated data collection was carried out using the Leginon software at a nominal magnification of 105,000x corresponding to a pixel size of 0.843 A. |
|---|---|
| Data analysis | CryoEM model building and analysis<br>Model Angelo114 was used to generate an initial model and UCSF Chimera 1.8 115 and Coot 0.9.8.8116 were used to manually build the model. Model was refined and rebuilt into the maps using Coot 0.9.8.8, Phenix 1.21117 and Rosetta 2021.07.61567118,119. Model validation was done using Molprobity120 and Privateer121 from the CCP4i2 suite. Figures were generated using UCSF ChimeraX 1.8122. Palmitoleic acids resolved in our cryo-EM maps were built in the final models based on previous findings on PEDV.<br><br>VSV pseudotyped virus infections and neutralizations<br>Relative luciferase units were plotted and normalized in Graphpad Prism 10<br><br>Biolayer interferometry (BLI) binding assays<br>Data were baseline subtracted, and the plots were fitted using the Sartorius analysis software (v.11.1). Data were plotted in Graphpad Prism 10. |

For manuscripts utilizing custom algorithms or software that are central to the research but not yet described in published literature, software must be made available to editors and reviewers. We strongly encourage code deposition in a community repository (e.g. GitHub). See the Nature Portfolio guidelines for submitting code & software for further information.

## Data

Policy information about availability of data

All manuscripts must include a data availability statement. This statement should provide the following information, where applicable:

- Accession codes, unique identifiers, or web links for publicly available datasets
- A description of any restrictions on data availability
- For clinical datasets or third party data, please ensure that the statement adheres to our policy

All data supporting the findings of this study are available within the paper and its Supplementary Information.
Accession codes for the structures presented in this work are: EMD-46714/PDB 9DB3 for FCoV-23 S-long with swung-out D0 (global refinement), EMD-46739/PDB 9DBZ for FCoV-23 S-long with mixed D0 conformations (global refinement), EMD-46716/PDB 9DBE for the local refinement of D0 from the FCoV-23 S-long swung-out dataset, EMD-46710/PDB 9DB1 for the local refinement of the proximal D0 from the FCoV-23 S-long with mixed D0 conformations, EMD-46709 PDB 9DB0 for FCoV-23 S-short, and EMD-46708/PDB 9DAZ for the F.cat APN-bound FCoV-23 RBD.

## Research involving human participants, their data, or biological material

Policy information about studies with human participants or human data. See also policy information about sex, gender (identity/presentation), and sexual orientation and race, ethnicity and racism.

| | |
|---|---|
| Reporting on sex and gender | N/A |
| Reporting on race, ethnicity, or other socially relevant groupings | N/A |
| Population characteristics | N/A |
| Recruitment | N/A |
| Ethics oversight | N/A |

Note that full information on the approval of the study protocol must also be provided in the manuscript.

# Field-specific reporting

Please select the one below that is the best fit for your research. If you are not sure, read the appropriate sections before making your selection.

☒ Life sciences  ☐ Behavioural & social sciences  ☐ Ecological, evolutionary & environmental sciences

For a reference copy of the document with all sections, see nature.com/documents/nr-reporting-summary-flat.pdf

# Life sciences study design

All studies must disclose on these points even when the disclosure is negative.

| | |
|---|---|
| Sample size | We use 10 mice to generate sera against HCoV-229E S (a typical group size) |
| Data exclusions | We did not exclude any data of the analysis |
| Replication | All binding and pseudovirus entry/neutralization experiments were replicated with at least two independent biological replicates, as indicated. |
| Randomization | We did not need to randomize the mice since we used this study only to generate polyclonal antibodies. |
| Blinding | We did not need to randomize the mice since we used this study only to generate polyclonal antibodies and all animals were analyzed. |

# Reporting for specific materials, systems and methods

We require information from authors about some types of materials, experimental systems and methods used in many studies. Here, indicate whether each material, system or method listed is relevant to your study. If you are not sure if a list item applies to your research, read the appropriate section before selecting a response.

## Materials & experimental systems

| n/a | Involved in the study |
|-----|-----------------------|
| ☐ | ☒ Antibodies |
| ☐ | ☒ Eukaryotic cell lines |
| ☒ | ☐ Palaeontology and archaeology |
| ☐ | ☒ Animals and other organisms |
| ☒ | ☐ Clinical data |
| ☒ | ☐ Dual use research of concern |
| ☒ | ☐ Plants |

## Methods

| n/a | Involved in the study |
|-----|-----------------------|
| ☒ | ☐ ChIP-seq |
| ☒ | ☐ Flow cytometry |
| ☒ | ☐ MRI-based neuroimaging |

# Antibodies

| | |
|---|---|
| Antibodies used | 76El monoclonal antibody (in house generated), HA tag monoclonal antibody (Thermofisher, cat# 26183, lot: YL384846), HA rabbit polyclonal antibody (Proteintech, cat# 51064-2, lot:00154837), anti-MLV p30 antibody (Abcam, cat# ab130757, lot: 1047279-23), , VSV-M antibody (Kerafast, cat # EB0011, lot: 200826), 488-conjugated goat antimouse secondary antibody (Thermofisher, cat# A-11029, lot: 2821059), Alexa Fluor 680-conjugated donkey anti-human secondary antibody (Jackson Immuno Research, code: 709-625-149, lot:161754), Alexa Fluor 680-conjugated goat antimouse secondary antibody (Jackson Immuno Research, code: 115-625-174, lot:161031), Alexa Fluor 680-conjugated goat anti-rabbit secondary antibody (Jackson Immuno Research, code: 111-625-144, lot:145147) |
| Validation | 76El monoclonal antibody. Sun, X., Yi, C., Zhu, Y. et al. Neutralization mechanism of a human antibody with pan-coronavirus reactivity including SARS-CoV-2. Nat Microbial 7, 1063-1074 (2022). https://doi.org/10.1038/s41564-022-01155-3.

HA monoclonal antibody (Thermo Fisher, https://www.thermofisher.com/antibody/product/HA-Tag-Antibody-clone-2-2-2-14-Monoclonal/26183)

HA-tag polyclonal antibody (Proteintech,, https://www.ptglab.com/products/HA-tag-Antibody-51064-2-AP.htm?srsltid=AfmBOoord6gacBa-N5uOxqfAsVXqOhkiKQvY7YjHkboTCIKutsnQE3KN and publications https://www.ptglab.com/products/HA-tag-Antibody-51064-2-AP.htm?srsltid=AfmBOoord6gacBa-N5uOxqfAsVXqOhkiKQvY7YjHkboTCIKutsnQE3KN#publications)

VSV M (23H12) monoclonal antibody (Kerafast, file:///Users/tortoric/Desktop/EB0011.pdf)

MLV p30 antibody (Abcam, https://www.abcam.com/en-us/products/primary-antibodies/mlv-p30-antibody-4b2-ab130757)

488-conjugated goat anti-mouse secondary antibody (Thermo Fisher, https://www.thermofisher.com/antibody/product/Goat-anti-Mouse-IgG-H-L-Highly-Cross-Adsorbed-Secondary-Antibody-Polyclonal/A-11029)

Alexa Fluor 680-conjugated donkey anti-human secondary antibody (Jackson Immuno Research, https://www.jacksonimmuno.com/catalog/products/709-625-149)

Alexa Fluor 680-conjugated goat anti-rabbit (Jackson Immuno Research, https://www.jacksonimmuno.com/catalog/products/111-625-144)

Alexa Fluor 680-conjugated goat anti-mouse (Jackson Immuno Research, https://www.jacksonimmuno.com/catalog/products/115-625-174) |

# Eukaryotic cell lines

Policy information about cell lines and Sex and Gender in Research

| | |
|---|---|
| Cell line source(s) | HEK293T (CRL-3216), A549 (CRM-CCL-185), Calu3 (HTB-55) LLC-MK2 (CCL-7), BHK- 21 (CCL-10), L2 (CCL-149),DF-1 (CRL-3586), CRFK (CCL-94), AK-0 (CCL-150), A-72 (CRL-1542), MOCK (CCL-34), LLC-PKl (CL-101), MDBK (CCL-22), av KID, Fcwf-CU, VeroE6-TMPRSS2 (JCRB1819). Cell lines ExpiCHO cells and Expi293F cells were obtained from ThermoFisher Scientific. Human bronchial epithelial cells (HBEC cells) were provided by Dr. Richard Cerione lab at Cornell University. Ovine kidney cells (OV KID) and Ovine fetal turbinate (OFT) cells were provided by Dr. Diego Diel at Cornell University Animal Health Diagnostic Center. Feline macrophage-like cells, Fcwf-4, were initially obtained from ATCC (CRL-2787) and their progeny (Fcwf-CU) was selected by Dr. Edward Dubovi and Dr. Gary Whittaker at Cornell University to be significantly better at propagating feline coronavirus. |
| Authentication | None of the cells were authenticated |
| Mycoplasma contamination | Cell lines were not routinely tested for mycoplasma contamination. |
| Commonly misidentified lines (See ICLAC register) | No commonly misidentified lines were used in this study |

## Animals and other research organisms

Policy information about studies involving animals; ARRIVE guidelines recommended for reporting animal research, and Sex and Gender in Research

| | |
|---|---|
| Laboratory animals | Female BALB/cAnNHsd mice were purchased from Envigo (order code 047) at 7 weeks of age and were maintained in a pathogen-free facility within the Department of Comparative Medicine at the University of Washington, Seattle, accredited by the Association for Assessment and Accreditation of Laboratory Animal Care (AAALAC). |
| Wild animals | No wild animals were used in this study |
| Reporting on sex | We did not analyze sex of the mice as a variable as it was not relevant to our study given that the mice were used only to generate polyclonal antibodies. |
| Field-collected samples | No field-collected samples were used in this study |
| Ethics oversight | Animal experiments were conducted in accordance with the University of Washington's Institutional Animal Care and Use Committee under protocol 4470-01 |

Note that full information on the approval of the study protocol must also be provided in the manuscript.

## Plants

| | |
|---|---|
| Seed stocks | N/A |
| Novel plant genotypes | N/A |
| Authentication | N/A |

