## [Peer Review file · Nature]

Loss of FCoV-23 spike domain 0 enhances fusogenicity and entry kinetics

Corresponding Author: Professor David Veesler

Version 0:

Reviewer comments:

Referee #1

(Remarks to the Author)

In this manuscript, Tortorici et al. and colleagues presented interesting findings related to the molecular mechanisms of cell entry and pathogenicity of a highly pathogenic Alphacoronavirus infecting cats (FCoV-23) through a combination of structural, biochemical, cellular biology, and immunology approaches. They determined high-quality (2.3-2.7 Å) structures of prefusion-stabilized S ectodomain trimer constructs of both long and short versions with particular focuses on domain 0 and its impact on the overall Spike architecture and function. The two domain 0 conformations were defined as "swung out" and "proximal". The authors suggested that the "proximal" conformation could sterically limit access of host proteases to cleave the S2' cleavage site, which is consistent with their observation that in-host loss of domain 0 in FCoV-23 S leads to enhanced fusogenicity in feline cells. The authors also comprehensively characterized the sialosides and APN receptor utilization with clear conclusions. Lastly, the authors tested the cross-neutralization efficiencies of several neutralizing antibodies and human plasma samples with a prior 229E infection.

These findings can greatly expand our knowledge of Alphacoronavirus-1. This is not only important for companion animals but also for humans that are susceptible to potential spillovers of these viruses. The manuscript is well-written and rich in information, with appropriate credit given to previous publications. Overall, this study is novel and would be of high interest to the field. I have several comments that may further improve the quality of this manuscript.

Major:

1. While the authors clearly showed enhanced syncytia formation induced by FCoV-23 S Short in Fig.3b (AK-D, A-72, and Fcwf-CU) and Extended Data Figure 8 (CRFK), is it possible that the phenotype is affected by the higher expression level or better cell surface distribution of FCoV-23 S Short compared to FCoV-23 S Long? The data quality of this cell-cell fusion assay data (especially Fig.3b) can be further improved by updating data with scale bars, more time points, zoomed-in views, and probably quantitative information. Alternatively, data with low background (no signal without syncytia formation) can also be produced using cell-cell fusion assays utilizing the split-reporter genes (sfGFP or Luciferase).

2. Since most PSV entry assay and neutralization assays used VSV pseudotypes, is there a specific reason for utilizing MLV-pseudotyped viruses in testing different cell lines in Fig.3a?

Minor:

Line 51: Typo: "feilds" should be "field".

Line 200: "CcoV-HuPn-2018" should be "CCoV-HuPn-2018".

Line 113: Consider updating "α-coronavirus" to "Alphacoronavirus" for consistency.

Lines 338-339: The authors mentioned that "Our structural data reveal that the proximal domain 0 conformation would sterically limit access of activating host proteases to the fusion peptide and S2' cleavage site". I suggest showing clear structure details in Figures 1a,b, or Supplementary Figures (S2' cleavage sites upstream of the fusion peptide should also be indicated).

Fig.1e: APN-interacting residues are not shown in FCoV1683. Additionally, there might be a mislabel of "794".

Fig.1g: Please precisely indicate the cleavage sites.

Line135: Considering that the S sequences derived peptides rather than the S trimmer are tested in this assay, which may not fully recapitulate the true protease sensitivity, I suggest updating the title to "Sensitivity of FCoV-23 S2' cleavage site sequences to host proteases" to be accurate. This can also be mentioned in the discussion section as a limitation.

Fig.3b: The strong fluorescence signal at only 4 hours post-transfection in FcW-CU cells is a bit surprising to me, please check if this time point is correct.

Fig.4a, b Please define the error bars.

Referee #2

(Remarks to the Author)

The incentives for this structural, biochemical and functional analysis of a feline coronavirus spike protein were prompted by a recent outbreak of severe feline infectious peritonitis (FIP) in Cyprus. The questions are whether feline coronaviruses might transmit zoonotically, and also, importantly, whether the viruses might acquire virulence. The evolution of a mild virus causing feline enteritis (FECV) to one causing virulent FIP is correlated with deletion of a spike domain (D0), and therefore, the authors here aimed to understand feline spike structure and function with and without the D0 domains. They generated several high-quality results showing structures of proline-stabilized feline spike ectodomains, with and without D0. They also showed that D0 is probably not a lectin (does not hemagglutinate), that the Domain B (RBD) binds APN, that canine APN orthologs can also bind domain B, that feline spike pseudoviruses (pps) can transduce feline and canine APN+ cells, and that a spike nAb blocks the pp transductions. Impressive data make for a good story about an important epidemic FCoV. Several findings allow for significant claims that APN is the FCoV receptor, that D0 occupies alternative conformation states on spike trimers, that D0 is dispensable, that loss of D0 changes spike character in cell fusion. Along with valuable findings, there are also claims that appear to require additional experimental support, as itemized below.

1. The “Molecular basis of pathogenicity.....” title gives the impression that results of virus pathogenicity are presented, for example, one might expect to see direct comparisons of in vivo infections by FCoVs with or without D0. Pathogenicity tests and results are not present and there are no experiments in which authentic FCoVs are evaluated.

2. In the abstract, it is stated that “in-host loss of domain 0.....enhances entry into cells....by facilitating protease access”. While there are results showing which proteases can cleave a peptide representing the S2prime cleavage sites, these findings do not say anything about whether domain 0 interferes with protease cleavages on virus spikes. Viruses (or pseudoviruses) with or without spike D0 domains might be evaluated for differential sensitivities to various cell entry-blocking protease inhibitors. These and other experiments seem necessary to support claims about D0 sterically blocking protease access to viral spike proteins.

3. From text in the abstract and discussion, one gets the impression that spike fusogenicity, as measured by syncytial spread in immortalized cell lines, correlates strongly with virus pathogenicity. This claim may need more support. On a related note about the measurements of cell-cell fusion that present in main figures, it may be important to also measure cell-surface levels of spike proteins with and without D0. Might D0 suppress or delay spike protein folding or transport to cell surfaces, with subsequent reduced cell-cell fusion?

4. The striking similarities between FCoV and TGEV evolution might be brought forward in more than the one statement... “reminiscent of the evolutionary process that led from TGEV to PRCV”. In the FCoV evolution, the loss of spike D0 is claimed to increase pathogenicity, through increased spike proteolysis and heightened fusion. In the TGEV evolution, the opposite pattern is observed, with loss of spike D0 decreasing pathogenicity. This seems worth considering in discussion.

Minor points:

1. While it eventually becomes evident that S “long” and S “short” signifies presence and absence of D0, the text could start out by precisely defining this “long” “short” nomenclature at the outset, particularly to help readers outside the field.

2. Fig S3; label the A, B, C, D domains

3. Fig S4; why is the 4c panel (WB of S proteins in MLV pps) present in this figure. This seems unrelated to in vitro protease activity assays and results, and the 4c panel is not brought up in the main text.

4. Fig S8, S protein-induced cell-cell fusion is not a measure of “permissiveness”, consider rewording title; also pp transductions can measure cell susceptibility to coronavirus (susceptible to entry) but not cell “permissivity” (i.e., cell support of post entry coronavirus replication and assembly); may want to reconsider terms in text.

Referee #3

(Remarks to the Author)

The manuscript by Tortorici et al. presents cryo-EM structures of the FCoV-23 spike protein in both its long and short forms. Key findings include: (1) In the long form, Domain 0 adopts two distinct conformations, with one potentially blocking protease access to the S2 cleavage site. (2) In the short form, Domain 0 is absent, leading to increased fusogenicity, as demonstrated by a cell-cell fusion assay. (3) The structure of the RBD/APN complex reveals specific virus-receptor interactions, showing that FCoV-23 S can utilize APN orthologs from several species, but not human APN. (4) The study also identifies an antibody that neutralizes FCoV-23 pseudoviruses.

Here are my comments:

The FCoV-23 S protein does not bind human APN, indicating that the virus cannot infect human cells, thereby posing no threat to humans. This contrasts with CCoV-HuPN-2018, which can infect humans, reducing the broader implications of this study.

Previous work by this group described the structure of the CCoV-HuPN-2018 spike protein complexed with an APN dimer. The FCoV-23 S protein shares 82.1% sequence similarity with the CCoV-HuPN-2018 spike glycoprotein, and the FCoV-23 RBD shares 93.5% similarity with the CCoV-HuPN-2018 RBD. Given these high similarities, the overall structures of the FCoV-23 S protein and RBD closely resemble those previously reported, reducing the novelty of the observed Domain 0 structure and RBD classification.

The proposed mechanisms involving Domain 0 in this study rely heavily on a recent preprint (<https://doi.org/10.1101/2023.11.08.566182>) suggesting that Domain 0 deletions are common in FIP-infected cats. However, it remains unclear whether Domain 0 is directly linked to disease progression. The authors propose that Domain 0 deletions may increase FIP infections in cats, but several key gaps remain between their findings and conclusions:

- i. Previous studies by this group showed that Domain 0 can bind sialosides, a critical feature for coronaviruses that facilitates spread. However, in this study, the FCoV-23 S Domain 0 did not hemagglutinate human erythrocytes, despite structural alignment. What are the structural differences causing these functional discrepancies? The authors should thoroughly investigate these variations to understand if they drive the observed differences, as the study suggests this domain has two opposing roles in viral infection.
- ii. All Domain 0 deletion experiments rely solely on in vitro assays. Neither this study nor the preprint provides enough evidence to conclusively establish a link between Domain 0 deletion and disease. The preprint notes that only two fecal samples (less than 6% of total samples) contained sequences with Δ D0, suggesting potential sampling bias.
- iii. Feline samples have shown considerable variability in the size of FCoV-23 spike deletions, ranging from 152 to 245 amino acids. Why did the authors focus on this specific short S variant? What do these variations imply for FCoV-23 infections?
- iv. In Figure 3B, the authors show a notable effect on cell-cell fusion (Δ Domain 0 vs. Domain 0), while Figure 3A shows minor differences in pseudovirus entry relative to cell-cell fusion. How can this discrepancy be explained? Are additional factors involved? Furthermore, the differences in pseudovirus entry lack statistical validation, with no visibly significant differences shown.
- v. Binding assays were conducted only between APN and RBD. However, given the study's emphasis on Domain 0's role in FCoV-23 entry, all BLI assays should include comparisons of APN binding with the full spike protein (with and without Domain 0) to assess whether Domain 0 influences receptor binding.
- vi. The hypothesis that Domain 0 blocks protease access to the S2 cleavage site requires biochemical validation, as structural data alone do not adequately demonstrate how a flexible domain could affect accessibility to this site.

Overall, the study's novelty is limited due to structural similarities with previously reported structures. Additionally, its scientific impact is reduced by a lack of comprehensive functional analysis of Domain 0. Robust evidence demonstrating whether Domain 0 deletion affects feline infection would be particularly valuable.

Referee #4

(Remarks to the Author)

The study by Tortorici et al. focuses on the newly identified and highly pathogenic recombinant coronavirus, FCoV-23, which is an important veterinary pathogen linked to a widespread outbreak of feline infectious peritonitis.

The authors use cryo-EM to elucidate the structures of the long and short spike isoforms of FCoV-23. By combining these data with infection and cell-cell fusion experiments, the authors propose that the in-host loss of domain 0 enhances viral entry and fusogenicity by improving protease access. This is an intriguing hypothesis that could explain FCoV-23 biotype switching and associated lethality.

The researchers further demonstrate that FCoV-23 can use various aminopeptidase N orthologs as receptors, providing insights into the molecular determinants of species tropism, including a glycan that influences receptor usage in humans. Additionally, the authors explore antigenic relationships among alphacoronaviruses affecting both humans and other mammals, identifying a cross-reactive monoclonal antibody that inhibits FCoV-23 pseudovirus entry. This finding could be of interest for the development of future vaccine and therapeutic strategies against this pathogen.

The strength of this manuscript is that it is timely and provides comprehensive structural and functional data relating to an important veterinary pathogen, offering insights into its tropism and its antigenic relationship with other alphacoronaviruses. The results will mainly be of interest to researchers in the coronavirus and veterinary medicine fields. The structural data is of very high quality and is completely in line with the complementary biolayer interferometry data. The most interesting aspect of the paper is the potential explanation for the in-host loss of domain 0, which the authors speculate increases the

accessibility of host cell proteases to the S2' site, leading to enhanced cell-cell fusion. However, it is with this aspect of the manuscript that I perceive flaws which currently prohibit its publication.

A weakness of this study is that, for the most part, it does not offer many surprising conclusions. The authors recently published a similarly comprehensive set of data on a closely related spike protein, CCoV-HuPn-2018 [1]. The degree of expected similarity with respect to spike structure, receptor binding, and antigenicity is highlighted by the authors multiple times throughout the manuscript. A notable difference is the lack of hemagglutination observed for the FCoV-23 domain 0. The large conformational changes observed in the domain 0, implicated by the authors in protease accessibility, were also observed in another study on the spike protein of the alphacoronavirus PEDV [2].

Major Concerns

The most interesting claim of the paper, that the loss of domain 0 in the short form of the spike permits protease access and increases fusogenicity, is not convincingly substantiated by the data presented. There is no mention of how many images were acquired for the cell-cell fusion experiments, and no statistical analysis is presented. Moreover, while the peptide cleavage experiments are a nice addition to the paper, it would be important to show differences in protease sensitivity in the context of the spike trimer. Finally, alternative explanations for the in-host evolution, such as increased sampling of the open RBD conformation upon loss of domain 0, are not explored. To warrant publication in *Nature*, the authors will need to strengthen these aspects of the paper with additional data.

Additional Experiments Could Look as Follows:

1. The authors should calculate the mean syncytia size for the long and short spike from multiple fields of view at different timepoints and carry out statistical analysis. There are several examples of such analysis being performed in the literature (e.g., [3-4]).
2. The authors should perform protease cleavage experiments in the context of the long and short form FCoV-23 spike and assess whether there are differences in cleavage, as suggested. The setup could use ectodomains, similar to the authors' previous SARS-CoV-2 experiments, or possibly pseudoviruses [5].
3. The authors should explore alternative explanations for the reason for domain 0 loss. One obvious example that springs to mind is that loss of domain 0 could lead to increased sampling of the open RBD conformation. Could the authors explore this by incubating the long and short spike ectodomains with soluble APN and carrying out negative stain analysis to quantify the number of receptor-bound complexes? This is assuming that APN binding can be detected with the spike ectodomain.

Minor Issues

- The manuscript appears to have been hastily written. This is evident from multiple mistakes with the figure callouts throughout the text.
- "CryoEM" and "Cryo-EM" are not used consistently between the main and supplementary text.
- There appears to be an error in reference 28 – "no title."
- In figure 1e, "PDCV" should be "PDCoV."
- Error in extended data figure 1a – "think layer of carbon."
- In the methods, "Thermofisher" should be "Thermo Fisher."
- Line number would be useful for future review.

References

1. Tortorici MA, Walls AC, Joshi A, Park YJ, Eguia RT, Miranda MC, Kepl E, Dosey A, Stevens-Ayers T, Boeckh MJ, Telenti A, Lanzavecchia A, King NP, Corti D, Bloom JD, Velesler D. Structure, receptor recognition, and antigenicity of the human coronavirus CCoV-HuPn-2018 spike glycoprotein. *Cell*. 2022 Jun 23;185(13):2279-2291.e17. doi: 10.1016/j.cell.2022.05.019. Epub 2022 May 27. PMID: 35700730; PMCID: PMC9135795.
2. Huang CY, Draczkowski P, Wang YS, Chang CY, Chien YC, Cheng YH, Wu YM, Wang CH, Chang YC, Yang TJ, Tsai YX, Khoo KH, Chang HW, Hsu SD. In situ structure and dynamics of an alphacoronavirus spike protein by cryo-ET and cryo-EM. *Nat Commun*. 2022 Aug 19;13(1):4877. doi: 10.1038/s41467-022-32588-3. PMID: 35986008; PMCID: PMC9388967.
3. Buchrieser J, Dufloo J, Hubert M, Monel B, Planas D, Rajah MM, Planchais C, Porrot F, Guivel-Benhassine F, Van der Werf S, Casartelli N, Mouquet H, Bruel T, Schwartz O. Syncytia formation by SARS-CoV-2-infected cells. *EMBO J*. 2020 Dec 1;39(23):e106267. doi: 10.15252/embj.2020106267. Epub 2020 Nov 4. Erratum in: *EMBO J*. 2021 Feb 1;40(3):e107405. doi: 10.15252/embj.2020107405. PMID: 33051876; PMCID: PMC7646020.
4. Ren W, Ju X, Gong M, Lan J, Yu Y, Long Q, Kenney DJ, O'Connell AK, Zhang Y, Zhong J, Zhong G, Douam F, Wang X, Huang A, Zhang R, Ding Q. Characterization of SARS-CoV-2 Variants B.1.617.1 (Kappa), B.1.617.2 (Delta), and B.1.618 by Cell Entry and Immune Evasion. *mBio*. 2022 Apr 26;13(2):e0009922. doi: 10.1128/mbio.00099-22. Epub 2022 Mar 10. PMID: 35266815; PMCID: PMC9040861.
5. Walls AC, Xiong X, Park YJ, Tortorici MA, Snijder J, Quispe J, Cameron E, Gopal R, Dai M, Lanzavecchia A, Zambon M, Rey FA, Corti D, Velesler D. Unexpected Receptor Functional Mimicry Elucidates Activation of Coronavirus Fusion. *Cell*. 2019 Feb 21;176(5):1026-1039.e15. doi: 10.1016/j.cell.2018.12.028. Epub 2019 Jan 31. Erratum in: *Cell*. 2020 Dec 10;183(6):1732. doi: 10.1016/j.cell.2020.11.031. PMID: 30712865; PMCID: PMC6751136.

Version 1:

Reviewer comments:

Referee #1

(Remarks to the Author)

The authors have conducted extensive experiments and gathered sufficient data to address the reviewers' concerns, particularly regarding the cell-cell fusion data essential for reinforcing the key conclusions. The quality of the revised manuscript has been improved significantly, which meets the high-quality standards of Nature.

I have several minor suggestions that could further strengthen the manuscript:

- (1) I recommend using "S.scr" instead of "S.scro" for "Sus scrofa" to maintain consistency with the nomenclature of other abbreviations unless there are specific reasons to differ.
- (2) Please define the abbreviation for Gallus gallus (G.gal) at its first mention in line 182.
- (3) Line 57-58: the statement "Alphacoronaviruses were initially thought to all use aminopeptidase N (APN) as a primary receptor" may not reflect a consensus in the field.
- (4) line 204, the phrase "concentration-dependent inhibition of FCoV-23 S-long, S-short and CCoV-HuPn-2018 S VSV pseudovirus entry into cells mediated by F.cat and C.fam APN" is somewhat misleading. I suggest revising it as "concentration-dependent inhibition of FCoV-23 S-long, S-short and CCoV-HuPn-2018 S VSV pseudovirus entry into cells mediated by dimeric F.cat and C.fam APN-Fc ectodomains".
- (5) Line 246, "C,fa" should be corrected to "C.fam."
- (6) The definitions and n numbers for plotting Fig. 3 a,b and ED Fig. 7g-i are unclear to me. Are these technical duplicates or means from different experiments?
- (7) Line 317-318, I believe the "proteolytic activation of the fusion machinery" also plays a role in the enhanced fusion by trypsin in addition to the "proteolytic removal of domain 0".

Referee #2

(Remarks to the Author)

The manuscript is significantly improved. Most concerns were addressed. The question of whether D0 deletion increases in vivo pathogenicity remains, but this reviewer appreciates the authors' statement that the answer to this important question requires construction of infectious FCoV clones, characterization of recombinant viruses and in vivo testing. As noted by authors, this extensive work can be considered beyond the scope of the paper.

There are some comments in this second round of review. Some concern Fig 4 data. The figure layout and text gives a view that cell-cell fusion and virus entry kinetics are related but this seems at odds with the Fig 4h findings suggesting that virus entry takes place after endocytosis. Cell-cell fusion (syncytia) is extracellular and neutral pH and may be different than endosomal virus-cell entry. Related to this – the results in Fig 4f and 4g are stated as measuring "entry kinetics", but they are actually measuring viral reporter gene expressions that come significantly after virus entry. Particularly in the case of retrovirus transduction, one can be skeptical whether reporter gene expressions coming after reverse and then forward transcription can measure changes in entry kinetics --- other events are probably more rate limiting than virus entry. Also, the Fig 4fg results may not be measuring entry "efficiency" as stated on lines 325-326. Unless statistics show convincingly less final levels of reporter gene expression with the S-long pseudovirus, the results show that reporter gene expressions come a bit earlier with D0 deletion but efficiency of the expressions may ultimately be similar with or without D0.

Another comment concerns the first paragraph of the discussion section. This paragraph highlights the D0 in controlling membrane fusion and makes a sensible speculation that increased fusion after D0 deletion correlates with increased pathogenesis. However, can a role for D0 in tissue tropism (as is the case for TGEV/PRCV) be ruled out?

Referee #3

(Remarks to the Author)

The authors have addressed my previous comments, but I still believe this work is more appropriate for a specialized journal.

Referee #4

(Remarks to the Author)

The authors have gone to great lengths to address my comments, many of which align with those of the other reviewers. As a result, the key claim of the paper regarding the loss of domain 0 and increased fusogenicity is better substantiated. I have no further objections to the publication of this manuscript, which is a welcome addition to the coronavirus literature.

We would like to thank the reviewers for taking the time to evaluate our manuscript and for providing insightful comments which helped make the revised version even stronger than the original submission (as described below).

Referee #1 (Remarks to the Author):

In this manuscript, Tortorici et al. and colleagues presented interesting findings related to the molecular mechanisms of cell entry and pathogenicity of a highly pathogenic Alphacoronavirus infecting cats (FCoV-23) through a combination of structural, biochemical, cellular biology, and immunology approaches. They determined high-quality (2.3-2.7 Å) structures of prefusion-stabilized S ectodomain trimer constructs of both long and short versions with particular focuses on domain 0 and its impact on the overall Spike architecture and function. The two domain 0 conformations were defined as "swung out" and "proximal". The authors suggested that the "proximal" conformation could sterically limit access of host proteases to cleave the S2' cleavage site, which is consistent with their observation that in-host loss of domain 0 in FCoV-23 S leads to enhanced fusogenicity in feline cells. The authors also comprehensively characterized the sialosides and APN receptor utilization with clear conclusions. Lastly, the authors tested the cross-neutralization efficiencies of several neutralizing antibodies and human plasma samples with a prior 229E infection.

These findings can greatly expand our knowledge of Alphacoronavirus-1. This is not only important for companion animals but also for humans that are susceptible to potential spillovers of these viruses. The manuscript is well-written and rich in information, with appropriate credit given to previous publications. Overall, this study is novel and would be of high interest to the field. I have several comments that may further improve the quality of this manuscript.

Major:

1. While the authors clearly showed enhanced syncytia formation induced by FCoV-23 S Short in Fig.3b (AK-D, A-72, and Fcwf-CU) and Extended Data Figure 8 (CRFK), is it possible that the phenotype is affected by the higher expression level or better cell surface distribution of FCoV-23 S Short compared to FCoV-23 S Long? The data quality of this cell-cell fusion assay data (especially Fig.3b) can be further improved by updating data with scale bars, more time points, zoomed-in views, and probably quantitative information. Alternatively, data with low background (no signal without syncytia formation) can also be produced using cell-cell fusion assays utilizing the split-reporter genes (sfGFP or Luciferase).

As suggested, we carried out all the fusion assays using a split GFP system via live-cell imaging to follow the kinetics of fusion after carefully normalizing S expression levels by adjusting the amounts of DNA transfected (as we found that S-short is better expressed than S-long). These new data support and extend our previous conclusions by showing that loss of domain 0 enhances FCoV-23 S fusogenicity in all cell types tested relative to the long form of the S trimer. In the

revised manuscript, we show that these findings also apply to CCoV-HuPn-2018 for which a domain 0-deletion mutant mediates markedly higher membrane fusion relative to the long form. Finally, we show that loss of domain 0 markedly enhances FCoV-23 S-mediated entry kinetics in multiple relevant feline and canine cell lines.

All these data are presented in Figure 4 and Extended Data Figures 9-10 of the revised manuscript and in the section entitled *Loss of domain 0 enhances S fusogenicity and entry kinetics*.

Figure 4. Loss of S domain 0 increases fusogenicity and entry kinetics. **a-e**, Kinetics of cell-cell fusion (GFP⁺ area) between BHK-21-GFP₁₋₁₀ cells transiently transfected with FCoV-23 S-long or S-short (**a-c**), CCoV-HuPn-2018 S-long or S-short (**d-e**) and BHK-21-GFP₁₁ target cells transiently transfected with membrane-anchored F.cat APN (**a**) or C.fam APN (**d**), cat macrophage-like Fcwf-CU cells (**b**), cat epithelial kidney CRFK cells (**c**) or canine fibroblast A-72 cells (**e**) cells endogenously expressing F.cat or C.fam APN and transiently transfected with GFP₁₁. S-transfected cells added

to BHK-21-GFP₁₁ target cells not expressing APN (no APN) or untransfected BHK-21-GFP₁₋₁₀ cells (no S) added to target cells expressing GFP₁₁ were used as negative controls. Representative fields of view obtained after S-expressing cells (pretreated or not with trypsin) were added to target cells are shown for the time points indicated by arrows, corresponding to t=18h (**a, c, d, e**) or t=5h (**b**) after which FcwF-CU cells were detaching or clumping hindering fluorescence measurements. Scale bars: 1000 μm. Statistical analysis was performed using area under the curve (AUC) of GFP⁺ area between the biological replicates with Two-Way ANOVA (**a,d**) or One-Way ANOVA (**b,c and e**) with Tukey's multiple comparison test (Graphpad Prism 10) using fusion data at t=18h (**a,c,d,e**) and t= 5h (**b**). Each dot in the bar graphs represents a biological experiment, each done with two to ten technical replicates. Means and SEM are shown as bars and lines. ns, not significant; *P ≤ 0.05, **P ≤ 0.01, ***P ≤ 0.001, ****P ≤ 0.0001. Western blot quantification of membrane-anchored S expression in BHK-21-GFP₁₋₁₀ cells detected using the fusion-peptide-specific 76E1 monoclonal antibody⁶² as primary antibody and Alexa Fluor 680-conjugated goat anti-human as secondary antibody. Data shown are one representative of two to four biological experiments. **f**, Entry kinetics of VSV particles pseudotyped with FCoV-23 S-long or S-short in FcwF-CU cells (**top**) and CRFK cells (**bottom**). **g**, Entry kinetics of MLV particles pseudotyped with FCoV-23 S-long or S-short in FcwF-CU cells (**top**), AK-D cells (**middle**) and A72 cells (**bottom**). Comparison of entry values between S-long and S-short for each time point was performed using multiple ratio paired t tests (**f**) and ratio paired t test (**g**) (Graphpad Prism 10). Each dot represents a biological experiment comprising ten to twelve technical replicates for VSV-pseudotypes and four replicates for MLV-pseudotypes. ns, not significant; *P ≤ 0.05, and **P ≤ 0.01. **h**, Relative entry of VSV pseudotyped with the indicated FCoV-23 S in FcwF-CU cells (**top**) or CRFK cells (**bottom**) cells treated with 25 μM nafamostat, camostat, E-64d or 1 μM bafilomycin A1. Normalization was based on the entry values obtained from cells treated with DMSO alone. Each dot represents the average of ten to twelve technical replicates from one representative experiment. Mean between the two biological replicates and the SEM are shown with bars and lines, respectively.

2. Since most PSV entry assay and neutralization assays used VSV pseudotypes, is there a specific reason for utilizing MLV-pseudotyped viruses in testing different cell lines in Fig.3a?

VSV entry and neutralization assays were run in the Veessler lab whereas MLV entry assays were run in the Whittaker lab. The agreement between the two assays strengthens our conclusions even further.

Minor:

Line 51: Typo: "feilds" should be "field".

Fixed.

Line 200: "CcoV-HuPn-2018" should be "CCoV-HuPn-2018".

Fixed.

Line 113: Consider updating "α-coronavirus" to "Alphacoronavirus" for consistency.

Done

Lines 338-339: The authors mentioned that "Our structural data reveal that the proximal domain 0 conformation would sterically limit access of activating host proteases to the fusion peptide and S2' cleavage site". I suggest showing clear structure details in Figures

1a,b, or Supplementary Figures (S₂' cleavage sites upstream of the fusion peptide should also be indicated).

Further investigations did not support the proposed increased S₂' sensitivity to proteases for FCoV-23 S-short vs S-long and we therefore removed this statement from the manuscript and introduced the new data as follows:

“Furthermore, we did not observe major differences of proteolytic sensitivity at S₁/S₂ or S₂' of FCoV-23 S-long and S-short anchored in the membrane of VSV pseudoviruses upon incubation with receptor, trypsin and proteinase K, suggesting that D0 does not impact S processing extensively (**Extended Data Figure 5g-i).**”

“Although exogenous trypsin addition to S-expressing cells enhanced S-long-mediated membrane fusion at early time points (**Fig 4a**), possibly as a result of proteolytic removal of domain 0 (**Extended Data Figure 5h**), S-long remained less fusogenic than S-short.”

Extended Data Figure 5. Sensitivity of FCoV-23 S to host proteases. g-i, Western blot evaluation of proteolytic sensitivity of FCoV-23 S-short and S-long anchored in the membrane of VSV pseudoviruses upon incubation with trypsin alone (**g**), with the recombinant C.fam APN ectodomain dimer (**h**) or anchored in the membrane of VSV pseudoparticles (**i**) followed by trypsin and proteinase K digestion. Anti-HA tag polyclonal antibody and Alexa Fluor 680-conjugated goat anti-rabbit were used as primary and secondary antibodies, respectively. Each western blot corresponds to one representative experiment out of two biological replicates.

As explained in answer to major point #1, loss of domain 0 greatly increases fusogenicity and entry kinetics.

Fig.1e: APN-interacting residues are not shown in FCoV1683. Additionally, there might be a mislabel of “794”.

Although it has been shown that FCoV serotype II viruses (e.g. 1683) use APN as a receptor, there is no structure available of the complex with the RBD. We therefore opted not to show any residues for this RBD in Fig1e as they would be speculative. We rephrased the legend of Fig 1e as follows for clarity:

“Structure-based phylogenetic classification of alpha- and delta-coronavirus RBDs calculated with FoldTree⁵². Conserved APN-interacting residues are shown in red (except for the FCoV1683 RBD which is an AlphaFold3-predicted structure⁵³). ”

Fig.1g: Please precisely indicate the cleavage sites.

We labeled the S₂' cleavage site in Fig 1g, as requested.

Line135: Considering that the S sequences derived peptides rather than the S trimer are tested in this assay, which may not fully recapitulate the true protease sensitivity, I suggest updating the title to "Sensitivity of FCoV-23 S₂' cleavage site sequences to host proteases" to be accurate. This can also be mentioned in the discussion section as a limitation.

Per the reviewer suggestion above, we have included an analysis of trypsin sensitivity in the context of membrane-anchored FCoV-23 S-short and S-long trimers in the revised manuscript (cf. answer to minor point #4 and Extended Data Fig5g-i).

We therefore opted to keep the heading to "Sensitivity of FCoV-23 S to host proteases".

Fig.3b: The strong fluorescence signal at only 4 hours post-transfection in FcwF-CU cells is a bit surprising to me, please check if this time point is correct.

This is correct. We further confirmed and extended these data (cf. answer to major point #1).

Cell-cell fusion is so effective in FcwF-CU cells that they are detaching by 5h in the new assay as well.

Fig.4a, b Please define the error bars.

Done.

Referee #2 (Remarks to the Author):

The incentives for this structural, biochemical and functional analysis of a feline coronavirus spike protein were prompted by a recent outbreak of severe feline infectious peritonitis (FIP) in Cyprus. The questions are whether feline coronaviruses might transmit zoonotically, and also, importantly, whether the viruses might acquire virulence. The evolution of a mild virus causing feline enteritis (FECV) to one causing virulent FIP is correlated with deletion of a spike domain (D0), and therefore, the authors here aimed to understand feline spike structure and function with and without the D0 domains. They generated several high-quality results showing structures of proline-stabilized feline spike ectodomains, with and without D0. They also showed that D0 is probably not a lectin (does not hemagglutinate), that the Domain B (RBD) binds APN, that canine APN orthologs can also bind domain B, that feline spike pseudoviruses (pps) can transduce feline and canine APN+ cells, and that a spike nAb blocks the pp transductions. Impressive data make for a good story about an important epidemic FCoV. Several findings allow for significant claims

that APN is the FCoV receptor, that D0 occupies alternative conformation states on spike trimers, that D0 is dispensable, that loss of D0 changes spike character in cell fusion. Along with valuable findings, there are also claims that appear to require additional experimental support, as itemized below.

1. The “Molecular basis of pathogenicity.....” title gives the impression that results of virus pathogenicity are presented, for example, one might expect to see direct comparisons of in vivo infections by FCoVs with or without D0. Pathogenicity tests and results are not present and there are no experiments in which authentic FCoVs are evaluated.

Thank you for this comment. We have unfortunately not yet been able to rescue a replicating FCoV-23 isolate from the samples obtained in Cyprus (likely due to suboptimal storage conditions). We are currently working on generating an infectious clone, which will take time and is beyond the scope of this manuscript. We therefore modified the title of the manuscript accordingly to:

“Loss of FCoV-23 spike domain 0 enhances fusogenicity and entry kinetics”

2. In the abstract, it is stated that “in-host loss of domain 0.....enhances entry into cells....by facilitating protease access”. While there are results showing which proteases can cleave a peptide representing the S2prime cleavage sites, these findings do not say anything about whether domain 0 interferes with protease cleavages on virus spikes. Viruses (or pseudoviruses) with or without spike D0 domains might be evaluated for differential sensitivities to various cell entry-blocking protease inhibitors. These and other experiments seem necessary to support claims about D0 sterically blocking protease access to viral spike proteins.

Thank you for these comments. We agree and carried out the requested experiments which are now reported in the revised Fig4 and described as follows:

“Pseudovirus entry into Fcwf-CU cells was reduced by bafilomycin A1 (an inhibitor of endosomal acidification⁸⁰) (**Fig 4h, top panel**) and entry into CRFK cells was dampened by bafilomycin A1 and E-64d (a cysteine protease inhibitor) for both FCoV-23 S-short and S-long (**Fig 4h, bottom panel**). These results and the lack of effect of camostat and nafamostat (serine protease inhibitors) suggest that FCoV-23 favors an endosomal entry route in the cell lines tested. Given the comparable impact of host protease inhibitors on entry of FCoV-23 S-short and S-long pseudoviruses along with the enhanced entry kinetics (VSV and MLV pseudoviruses) and fusogenicity (cell-cell fusion) mediated by FCoV-23 S-short, relative to S-long, these properties likely reflect intrinsic functional differences of these two S isoforms instead of distinct entry routes.”

Figure 4. Loss of S domain 0 increases fusogenicity and entry kinetics. **h**, Relative entry of VSV pseudotyped with the indicated FCoV-23 S in Fcwf-CU cells (**top**) or CRFK cells (**bottom**) cells treated with 25 μ M nafamostat, camostat, E-64d or 1 μ M bafilomycin A1. Normalization was based on the entry values obtained from cells treated with DMSO alone. Each dot represents the average of ten to twelve technical replicates from one representative experiment. Mean between the two biological replicates and the SEM are shown with bars and lines, respectively.

Please also see answer to minor point #4 from reviewer 1.

3. From text in the abstract and discussion, one gets the impression that spike fusogenicity, as measured by syncytial spread in immortalized cell lines, correlates strongly with virus pathogenicity. This claim may need more support. On a related note about the measurements of cell-cell fusion that present in main figures, it may be important to also measure cell-surface levels of spike proteins with and without D0. Might D0 suppress or delay spike protein folding or transport to cell surfaces, with subsequent reduced cell-cell fusion?

Please see answer to major point #1 from reviewer 1. The new data are described in the revised Fig.4 and Extended Data Figures 9-10.

The marked differences in fusion and entry kinetics we observed for FCoV-23 S-short vs S-long (and extended to the human pathogen CCoV-HuPn-2018) are reminiscent of findings on the SARS-CoV-2 Delta variant for which enhanced fusogenicity and pseudovirus entry kinetics, relative to previous variants, led to enhanced viral replication kinetics and pathogenicity in vivo (PMID: 34823256, PMID: 35550680, PMID: 34698504). Although no FCoV-23 authentic viral isolate are available, we propose that FCoV-23 found a different molecular solution leading to a similar phenotype and carefully worded this in the discussion as follows:

“Furthermore, the markedly greater fusogenicity and entry kinetics of FCoV-23 S-short observed in various cell lines, relative to S-long, suggest that deletion of D0 may promote increased syncytia formation, replication kinetics and pathology, as described for the SARS-CoV-2 Delta variant as a result of the P681R substitution⁶⁶. These findings offer a possible explanation for the repeated loss of D0 among FCoV-23-infected cats, which can occur genomically (as observed *in vivo*) and putatively proteolytically (as observed with FCoV-23 S-long VSV pseudovirus).”

4. The striking similarities between FCoV and TGEV evolution might be brought forward in more than the one statement... “reminiscent of the evolutionary process that led from TGEV to PRCV”. In the FCoV evolution, the loss of spike D0 is claimed to increase pathogenicity, through increased spike proteolysis and heightened fusion. In the TGEV evolution, the opposite pattern is observed, with loss of spike D0 decreasing pathogenicity. This seems worth considering in discussion.

Thank you for this suggestion. We edited this section as follows:

“These findings are reminiscent of the evolutionary process that led from TGEV to PRCV upon deletion of D0 in the TGEV S gene, resulting in distinct tissue tropism between the two viruses^{81,90}. TGEV replicates mainly in small intestine epithelial cells and causes severe and often fatal diarrhea in young piglets whereas PRCV replicates mostly in respiratory epithelial cells and causes mild respiratory symptoms without diarrhea. Therefore, the loss of D0 possibly yields opposite outcomes in terms of severity for TGEV/PRCV and for FCoV-23.”

Minor points:

1. While it eventually becomes evident that S “long” and S “short” signifies presence and absence of D0, the text could start out by precisely defining this “long” “short” nomenclature at the outset, particularly to help readers outside the field.

Thank you for this suggestion. We clarified this nomenclature in the ‘Architecture of the FCoV-23 S glycoprotein’ section of the Results.

2. Fig S3; label the A, B, C, D domains

Done

3. Fig S4; why is the 4c panel (WB of S proteins in MLV pps) present in this figure. This seems unrelated to in vitro protease activity assays and results, and the 4c panel is not brought up in the main text.

Thank you for this suggestion. We reorganized the figure panels for more clarity. The panel referred to by the reviewer is now part of Extended Data Fig9.

Extended Data Fig5g-i now shows evaluation of proteolytic sensitivity of FCoV-23 S-long and S-short anchored in the membrane of VSV pseudoviruses upon incubation with receptor, trypsin and proteinase K.

4. Fig S8, S protein-induced cell-cell fusion is not a measure of “permissiveness”, consider rewording title; also pp transductions can measure cell susceptibility to coronavirus (susceptible to entry) but not cell “permissivity” (i.e., cell support of post entry coronavirus replication and assembly); may want to reconsider terms in text.

We agree and changed the title of this figure to ‘Entry of MLV particles pseudotyped with FCoV-23 S-long and S-short in a panel of cell lines spanning various species and tissues’ in the revised Extended Data Fig9.

Referee #3 (Remarks to the Author):

The manuscript by Tortorici et al. presents cryo-EM structures of the FCoV-23 spike protein in both its long and short forms. Key findings include: (1) In the long form, Domain 0 adopts two distinct conformations, with one potentially blocking protease access to the S2 cleavage site. (2) In the short form, Domain 0 is absent, leading to increased fusogenicity, as demonstrated by a cell-cell fusion assay. (3) The structure of the RBD/APN complex reveals specific virus-receptor interactions, showing that FCoV-23 S can utilize APN orthologs from several species, but not human APN. (4) The study also identifies an antibody that neutralizes FCoV-23 pseudoviruses.

Here are my comments:

The FCoV-23 S protein does not bind human APN, indicating that the virus cannot infect human cells, thereby posing no threat to humans. This contrasts with CCoV-HuPn-2018, which can infect humans, reducing the broader implications of this study.

We apologize for the confusion. The APN species tropism of FCoV-23 and CCoV-HuPn-2018 are identical (due to closely related RBDs) and although the latter virus is not well adapted for human infection, it is clearly capable of doing so as it has been isolated in humans, as was also the case for HuCCoV_Z19Haiti (PMID: 34013321, PMID: 34718467). These findings indicate that FCoV-23 may be able to cross the species barrier but that further adaptive mutations would be required for efficient infection (as would be the case for CCoV-HuPn-2018 and HuCCoV_Z19Haiti).

These data are reminiscent of recent results we described for HKU5, which appears unable to use the human ACE2 receptor with single-round (non-replicating) pseudoviruses but is actually able to do so weakly when using authentic virus in the presence of trypsin, most likely due to amplification of signal allowing detection upon replication (PMID: 39253417).

Previous work by this group described the structure of the CCoV-HuPn-2018 spike protein complexed with an APN dimer. The FCoV-23 S protein shares 82.1% sequence similarity with the CCoV-HuPn-2018 spike glycoprotein, and the FCoV-23 RBD shares 93.5% similarity with the CCoV-HuPn-2018 RBD. Given these high similarities, the overall

structures of the FCoV-23 S protein and RBD closely resemble those previously reported, reducing the novelty of the observed Domain 0 structure and RBD classification.

We decided to characterize the structure, function and antigenicity of FCoV-23 due to the fact that it mediates severe disease in cats and its resemblance to two human viruses (CCoV-HuPn-2018 and HuCCoV_Z19Haiti).

Similarly, when SARS-CoV-2 emerged, it was key to characterize the spike of the Wuhan-Hu-1 isolate although it shared ~80% sequence identity with that of SARS-CoV-1 (which had been characterized previously) or of emerging variants that differed by even fewer mutations.

The proposed mechanisms involving Domain 0 in this study rely heavily on a recent preprint (<https://doi.org/10.1101/2023.11.08.566182>) suggesting that Domain 0 deletions are common in FIP-infected cats. However, it remains unclear whether Domain 0 is directly linked to disease progression. The authors propose that Domain 0 deletions may increase FIP infections in cats, but several key gaps remain between their findings and conclusions:

i. Previous studies by this group showed that Domain 0 can bind sialosides, a critical feature for coronaviruses that facilitates spread. However, in this study, the FCoV-23 S Domain 0 did not hemagglutinate human erythrocytes, despite structural alignment. What are the structural differences causing these functional discrepancies? The authors should thoroughly investigate these variations to understand if they drive the observed differences, as the study suggests this domain has two opposing roles in viral infection.

As shown in Extended Data Fig 6g, the FCoV-23 and CCoV-HuPn-2018 domain 0 sequences are highly divergent. We have not yet been able to identify the nature of the sialoside receptor used by CCoV-HuPn-2018 (a glycan array failed to yield any hits), thereby hindering our understanding of the differences around the yet unknown binding site.

However, the data do not suggest opposing roles for domain 0 but instead that it may participate in tissue tropism (as is the case for TGEV vs PRCV) and pathogenicity and we observed similar phenotypes for loss of D0 in FCoV-23 and CCoV-HuPn-2018.

ii. All Domain 0 deletion experiments rely solely on in vitro assays. Neither this study nor the preprint provides enough evidence to conclusively establish a link between Domain 0 deletion and disease. The preprint notes that only two fecal samples (less than 6% of total samples) contained sequences with $\Delta D0$, suggesting potential sampling bias.

We apologize for any confusion. The preprint from the Tait-Burkard lab reports that the majority of samples (>90%) show a deletion in domain 0 and reports sequences from 57 cats with a deletion of domain 0 (<https://www.biorxiv.org/content/10.1101/2023.11.08.566182v3.full>). Furthermore, the only two fecal samples where sequences could be obtained also show a deletion in domain 0.

The marked differences in fusion and entry kinetics we observed for FCoV-23 S-short vs S-long (and extended to the human pathogen CCoV-HuPn-2018) are reminiscent of findings on the SARS-CoV-2 Delta variant for which enhanced fusogenicity and pseudovirus entry kinetics, relative to previous variants, led to enhanced viral replication kinetics and pathogenicity (PMID: 34823256, PMID: 35550680, PMID: 34698504). Although no FCoV-23 authentic viral isolate are available, we propose that FCoV-23 found a different molecular solution leading to a similar phenotype and carefully worded this in the discussion.

iii. Feline samples have shown considerable variability in the size of FCoV-23 spike deletions, ranging from 152 to 245 amino acids. Why did the authors focus on this specific short S variant? What do these variations imply for FCoV-23 infections?

We selected this variant with a deletion of 211 residues (residues 53-264) as this is the consensus sequence described in the preprint from the Tait-Burkard lab (<https://www.biorxiv.org/content/10.1101/2023.11.08.566182v3.full>) and is very close to the average deletion size (665.88 nt) and the median (657 nt) observed among infected cats. We note that the fact that our findings on the role of domain 0 hold true for the human pathogen CCoV-HuPn-2018 underscores the robustness of our conclusions.

For the sake of clarity for the readers, we added a detailed description of the deletion to the “Architecture of the FCoV-23 S glycoprotein” section along with a dedicated Extended Data Figure 1.

Extended Data Figure 1. Sequence alignment of the FCoV-23 S-long and S-short constructs used in this study. The S-short construct corresponds to the C10-DA isolate.

iv. In Figure 3B, the authors show a notable effect on cell-cell fusion (Δ Domain 0 vs. Domain 0), while Figure 3A shows minor differences in pseudovirus entry relative to cell-cell fusion. How can this discrepancy be explained? Are additional factors involved? Furthermore, the differences in pseudovirus entry lack statistical validation, with no visibly significant differences shown.

Please see answer to major point #1 of reviewer 1 for our updated and much more extensive analysis of the phenotypic consequences of the loss of domain 0. Our conclusions hold true.

We now show that the markedly enhanced entry kinetics of FCoV-23 S-short vs S-long are very obvious at early time points but reduced at much later time points (e.g. 24h for VSV and 72h for MLV), likely due to the dynamic range of the assay and consistent with previous data on the SARS-CoV-2 Delta variant (PMID: 34698504). These new data are described in the revised Fig4 and Extended Data Fig9-10 and in the text as follows:

“We next investigated if the increase in fusogenicity promoted greater kinetics of S-mediated entry into cells by comparing VSV pseudoviruses harboring FCoV-23 S-short and S-long at various time points post transduction. Compared to S-long VSV, S-short VSV entered 1.9-fold, 1.3-fold and 1.1-fold more efficiently in Fcwf-CU cells at 3, 4 and 6 h post infection, respectively (**Fig 4f top panel and Extended Data Figure 10f**) and 1.5-fold, 3-fold and 3.6-fold more efficiently in CRFK cells at 4, 6 and 8 h post infection, respectively (**Fig 4f bottom panel and Extended Data Figure 10f**). These differences were reduced at the end of the experiment (24h time point), likely explaining the small differences in MLV pseudovirus entry detected for the two isoforms 72 h post transduction (**Extended Data Figure 9a**). Accordingly, we observed greater entry of MLV pseudotyped with FCoV-23 S-short, relative to S-long, at 8 h post-transduction in Fcwf-CU (3.6-fold), AK-D (4-fold) and A-72 (50-fold) cells (**Fig 4g and Extended Data Figure 9b-c**). These results show that loss of D0 promotes more rapid entry into target cells endogenously expressing F.cat or C.fam APN, concurring with the increased syncytia formation for FCoV-23 S-short relative to S-long.”

v. Binding assays were conducted only between APN and RBD. However, given the study’s emphasis on Domain 0’s role in FCoV-23 entry, all BLI assays should include comparisons of APN binding with the full spike protein (with and without Domain 0) to assess whether Domain 0 influences receptor binding.

To answer this comment, we opted to carry out canine APN-Fc pull-down experiments using the FCoV-23 S-long and S-short prefusion ectodomain trimers. Our data show that both S constructs engage the receptor with comparable efficiency, suggesting that domain 0 does not modulate receptor binding. These new data are described in Extended Data Fig6 and in the revised text as follows:

“To evaluate the impact of the loss of D0 on receptor engagement, we carried out APN pull-down assays using prefusion FCoV-23 S-long and S-short ectodomain trimers immobilized on NTA cobalt-based beads. We found that both S trimers interacted with comparable efficiency with the C.fam APN-Fc ectodomain (**Extended Data Figure 7c-d**), ruling out a role of D0 on imparting major conformational differences of the RBD, concurring with the structural data (except for the exclusive detection of closed trimers by cryo-EM).”

Extended Data Figure 7. Characterization of receptor binding and entry of FCoV-23, CCoV-HuPn-2018 and HCoV-229E. c-d, Coomassie-stained SDS-PAGE analysis of C.fam APN-Fc pull-down experiments using either FCoV-23 S-short (c) or S-long (d) immobilized on NTA-cobalt based magnetic beads. Asterisks indicate the positions of the pulled down APN/S complexes. UB₁: unbound fraction after incubating with S or buffer, UB₂: unbound fraction after incubation with C.fam APN-Fc or buffer; W₁: final wash after incubating with S or buffer, W₂: final wash after incubating with C.fam APN-Fc; B, bound. Purified S coupled to beads and not incubated with C.fam APN-Fc and C.fam APN-Fc incubated with uncoupled beads are shown on the gels at the left hand side of panels c and d. One pull down experiment corresponding to one preparation of proteins is shown out of two biological replicates.

vi. The hypothesis that Domain 0 blocks protease access to the S2 cleavage site requires biochemical validation, as structural data alone do not adequately demonstrate how a flexible domain could affect accessibility to this site.

Please see answer to minor point #4 from reviewer 1.

Overall, the study's novelty is limited due to structural similarities with previously reported structures. Additionally, its scientific impact is reduced by a lack of comprehensive functional analysis of Domain 0. Robust evidence demonstrating whether Domain 0 deletion affects feline infection would be particularly valuable.

We respectfully disagree. We decided to characterize this newly emerged virus due to the fact that it claimed ~10,000 cat lives and its resemblance to two recently discovered human viruses (CCoV-HuPn-2018 and HuCCoV_Z19Haiti) with which it shares identical receptor species utilization. The importance of this work is clearly recognized by reviewers 1, 2 and 4 and the

comprehensive characterization of the biochemical and functional role of domain 0 included in the revised manuscript strengthens our study even further. Finally, we show that our findings extend to the human-infecting CCoV-HuPn-2018 and are more generalizable than previously anticipated.

Referee #4 (Remarks to the Author):

The study by Tortorici et al. focuses on the newly identified and highly pathogenic recombinant coronavirus, FCoV-23, which is an important veterinary pathogen linked to a widespread outbreak of feline infectious peritonitis.

The authors use cryo-EM to elucidate the structures of the long and short spike isoforms of FCoV-23. By combining these data with infection and cell-cell fusion experiments, the authors propose that the in-host loss of domain 0 enhances viral entry and fusogenicity by improving protease access. This is an intriguing hypothesis that could explain FCoV-23 biotype switching and associated lethality.

The researchers further demonstrate that FCoV-23 can use various aminopeptidase N orthologs as receptors, providing insights into the molecular determinants of species tropism, including a glycan that influences receptor usage in humans. Additionally, the authors explore antigenic relationships among alphacoronaviruses affecting both humans and other mammals, identifying a cross-reactive monoclonal antibody that inhibits FCoV-23 pseudovirus entry. This finding could be of interest for the development of future vaccine and therapeutic strategies against this pathogen.

The strength of this manuscript is that it is timely and provides comprehensive structural and functional data relating to an important veterinary pathogen, offering insights into its tropism and its antigenic relationship with other alphacoronaviruses. The results will mainly be of interest to researchers in the coronavirus and veterinary medicine fields. The structural data is of very high quality and is completely in line with the complementary biolayer interferometry data. The most interesting aspect of the paper is the potential explanation for the in-host loss of domain 0, which the authors speculate increases the accessibility of host cell proteases to the S2' site, leading to enhanced cell-cell fusion. However, it is with this aspect of the manuscript that I perceive flaws which currently prohibit its publication.

A weakness of this study is that, for the most part, it does not offer many surprising conclusions. The authors recently published a similarly comprehensive set of data on a closely related spike protein, CCoV-HuPn-2018 [1]. The degree of expected similarity with respect to spike structure, receptor binding, and antigenicity is highlighted by the authors multiple times throughout the manuscript. A notable difference is the lack of hemagglutination observed for the FCoV-23 domain 0. The large conformational changes observed in the domain 0, implicated by the authors in protease accessibility, were also observed in another study on the spike protein of the alphacoronavirus PEDV [2].

Major Concerns

The most interesting claim of the paper, that the loss of domain 0 in the short form of the spike permits protease access and increases fusogenicity, is not convincingly substantiated by the data presented. There is no mention of how many images were acquired for the cell-cell fusion experiments, and no statistical analysis is presented. Moreover, while the peptide cleavage experiments are a nice addition to the paper, it would be important to show differences in protease sensitivity in the context of the spike trimer. Finally, alternative explanations for the in-host evolution, such as increased sampling of the open RBD conformation upon loss of domain 0, are not explored. To warrant publication in Nature, the authors will need to strengthen these aspects of the paper with additional data.

Additional Experiments Could Look as Follows:

1. The authors should calculate the mean syncytia size for the long and short spike from multiple fields of view at different timepoints and carry out statistical analysis. There are several examples of such analysis being performed in the literature (e.g., [3-4]).

We agree with the reviewer. Please see answer to major point #1 from reviewer 1.

2. The authors should perform protease cleavage experiments in the context of the long and short form FCoV-23 spike and assess whether there are differences in cleavage, as suggested. The setup could use ectodomains, similar to the authors' previous SARS-CoV-2 experiments, or possibly pseudoviruses [5].

Thank you for this suggestion. Please see answer to minor point #4 from reviewer 1.

3. The authors should explore alternative explanations for the reason for domain 0 loss. One obvious example that springs to mind is that loss of domain 0 could lead to increased sampling of the open RBD conformation. Could the authors explore this by incubating the long and short spike ectodomains with soluble APN and carrying out negative stain analysis to quantify the number of receptor-bound complexes? This is assuming that APN binding can be detected with the spike ectodomain.

We performed the requested experiments using pull-down assays. Please see answer to point v from reviewer 3.

As suggested, we carried out all the fusion assays using a split GFP system via live-cell imaging to follow the kinetics of fusion after carefully normalizing S expression levels by adjusting the amounts of DNA transfected (as we found that S-short is better expressed than S-long). These new data support and extend our previous conclusions by showing that loss of domain 0 enhances FCoV-23 S fusogenicity in all cell types tested relative to the long form of the S trimer. In the

revised manuscript, we show that these findings also apply to CCoV-HuPn-2018 for which a domain 0-deletion mutant mediates markedly higher membrane fusion relative to the long form. Finally, we show that loss of domain 0 markedly enhances FCoV-23 S-mediated entry kinetics in multiple relevant feline and canine cell lines.

All these data are presented in Figure 4 and Extended Data Figures 9-10 of the revised manuscript and in the section entitled *Domain 0 deletion enhances S fusogenicity and entry kinetics*.

Minor Issues

- **The manuscript appears to have been hastily written. This is evident from multiple mistakes with the figure callouts throughout the text.**

We apologize about this and we fixed these issues.

- **“CryoEM” and “Cryo-EM” are not used consistently between the main and supplementary text.**

Fixed.

- **There appears to be an error in reference 28 – “no title.”**

This is not an error, we are citing a textbook.

- **In figure 1e, “PDCV” should be “PDCoV.”**

Fixed

- **Error in extended data figure 1a – “think layer of carbon.”**

Fixed

- **In the methods, “Thermofisher” should be “Thermo Fisher.”**

Fixed

- **Line number would be useful for future review.**

Added

References

1. Tortorici MA, Walls AC, Joshi A, Park YJ, Eguia RT, Miranda MC, Kepl E, Dosey A, Stevens-Ayers T, Boeckh MJ, Telenti A, Lanzavecchia A, King NP, Corti D, Bloom JD, Veessler D. Structure, receptor recognition, and antigenicity of the human coronavirus

- CCoV-HuPn-2018 spike glycoprotein. *Cell*. 2022 Jun 23;185(13):2279-2291.e17. doi: 10.1016/j.cell.2022.05.019. Epub 2022 May 27. PMID: 35700730; PMCID: PMC9135795.
2. Huang CY, Draczkowski P, Wang YS, Chang CY, Chien YC, Cheng YH, Wu YM, Wang CH, Chang YC, Yang TJ, Tsai YX, Khoo KH, Chang HW, Hsu SD. In situ structure and dynamics of an alphacoronavirus spike protein by cryo-ET and cryo-EM. *Nat Commun*. 2022 Aug 19;13(1):4877. doi: 10.1038/s41467-022-32588-3. PMID: 35986008; PMCID: PMC9388967.
3. Buchrieser J, Dufloo J, Hubert M, Monel B, Planas D, Rajah MM, Planchais C, Porrot F, Guivel-Benhassine F, Van der Werf S, Casartelli N, Mouquet H, Bruel T, Schwartz O. Syncytia formation by SARS-CoV-2-infected cells. *EMBO J*. 2020 Dec 1;39(23):e106267. doi: 10.15252/embj.2020106267. Epub 2020 Nov 4. Erratum in: *EMBO J*. 2021 Feb 1;40(3):e107405. doi: 10.15252/embj.2020107405. PMID: 33051876; PMCID: PMC7646020.
4. Ren W, Ju X, Gong M, Lan J, Yu Y, Long Q, Kenney DJ, O'Connell AK, Zhang Y, Zhong J, Zhong G, Douam F, Wang X, Huang A, Zhang R, Ding Q. Characterization of SARS-CoV-2 Variants B.1.617.1 (Kappa), B.1.617.2 (Delta), and B.1.618 by Cell Entry and Immune Evasion. *mBio*. 2022 Apr 26;13(2):e0009922. doi: 10.1128/mbio.00099-22. Epub 2022 Mar 10. PMID: 35266815; PMCID: PMC9040861.
5. Walls AC, Xiong X, Park YJ, Tortorici MA, Snijder J, Quispe J, Cameroni E, Gopal R, Dai M, Lanzavecchia A, Zambon M, Rey FA, Corti D, Veasler D. Unexpected Receptor Functional Mimicry Elucidates Activation of Coronavirus Fusion. *Cell*. 2019 Feb 21;176(5):1026-1039.e15. doi: 10.1016/j.cell.2018.12.028. Epub 2019 Jan 31. Erratum in: *Cell*. 2020 Dec 10;183(6):1732. doi: 10.1016/j.cell.2020.11.031. PMID: 30712865; PMCID: PMC6751136.

Referee #1 (Remarks to the Author):

The authors have conducted extensive experiments and gathered sufficient data to address the reviewers' concerns, particularly regarding the cell-cell fusion data essential for reinforcing the key conclusions. The quality of the revised manuscript has been improved significantly, which meets the high-quality standards of Nature.

Thank you very much.

I have several minor suggestions that could further strengthen the manuscript:
(1) I recommend using "S.scr" instead of "S.scro" for "Sus scrofa" to maintain consistency with the nomenclature of other abbreviations unless there are specific reasons to differ.

Done. We updated throughout the manuscript.

(2) Please define the abbreviation for Gallus gallus (G.gal) at its first mention in line 182.

Done.

(3) Line 57-58: the statement "Alphacoronaviruses were initially thought to all use aminopeptidase N (APN) as a primary receptor" may not reflect a consensus in the field.

To address this comment, we rephrased as follows:

"Although many alphacoronaviruses use aminopeptidase N (APN) as a primary receptor³⁸, the identification of ACE2 as the HCoV-NL63 receptor³⁹ revealed a diversity of receptor usage in this genus."

(4) line 204, the phrase "concentration-dependent inhibition of FCoV-23 S-long, S-short and CCoV-HuPn-2018 S VSV pseudovirus entry into cells mediated by F.cat and C.fam APN" is somewhat misleading. I suggest revising it as "concentration-dependent inhibition of FCoV-23 S-long, S-short and CCoV-HuPn-2018 S VSV pseudovirus entry into cells mediated by dimeric F.cat and C.fam APN-Fc ectodomains".

Done.

(5) Line 246, "C,fa" should be corrected to "C.fam."

Done.

(6) The definitions and n numbers for plotting Fig. 3 a,b and ED Fig. 7g-i are unclear to me. Are these technical duplicates or means from different experiments?

We rephrased for clarity as follows:

Fig.3 a,b : “Each curve represents a biological replicate (n=2, using distinct pseudovirus batches) performed with two technical duplicates each. Error bars represent the standard error of the mean (SEM) of the technical replicates..”

Extended Data Fig.7g-i: ” Each curve corresponds to a single biological replicate performed with technical duplicates. The error bars represent the standard error of the mean (SEM) of the technical duplicates.”

(7) Line 317-318, I believe the “proteolytic activation of the fusion machinery” also plays a role in the enhanced fusion by trypsin in addition to the "proteolytic removal of domain 0".

Although this is a possibility, trypsin-mediated proteolytic activation of the fusion machinery would not be expected to be different between S-long and S-short. The fact that trypsin addition boosts fusion mediated by S-long but not S-short points to proteolytic removal of D0 as a possible explanation.

In any case, per the reviewer request, we modified the text as follows:

“Although exogenous trypsin addition to S-expressing cells enhanced S-long-mediated membrane fusion at early time points (Fig 4a), possibly as a result of proteolytic removal of domain 0 and direct activation of the fusion machinery (Extended Data Figure 5h), S-long remained less fusogenic than S-short.”

Referee #2 (Remarks to the Author):

The manuscript is significantly improved. Most concerns were addressed. The question of whether D0 deletion increases in vivo pathogenicity remains, but this reviewer appreciates the authors’ statement that the answer to this important question requires construction of infectious FCoV clones, characterization of recombinant viruses and in vivo testing. As noted by authors, this extensive work can be considered beyond the scope of the paper.

Thank you very much.

There are some comments in this second round of review. Some concern Fig 4 data. The figure layout and text gives a view that cell-cell fusion and virus entry kinetics are related but this seems at odds with the Fig 4h findings suggesting that virus entry takes place after endocytosis. Cell-cell fusion (syncytia) is extracellular and neutral pH and may be different than endosomal virus-cell entry. Related to this – the results in Fig 4f and 4g are stated as measuring “entry kinetics”, but they are actually measuring viral reporter gene expressions that come significantly after virus entry. Particularly in the case of retrovirus transduction, one can be skeptical whether reporter gene expressions coming after reverse and then forward transcription can measure changes in entry kinetics --- other

events are probably more rate limiting than virus entry. Also, the Fig 4fg results may not be measuring entry “efficiency” as stated on lines 325-326. Unless statistics show convincingly less final levels of reporter gene expression with the S-long pseudovirus, the results show that reporter gene expressions come a bit earlier with D0 deletion but efficiency of the expressions may ultimately be similar with or without D0.

Our rationale for using the term “entry kinetics” in the pseudoparticle infection assay is that reporter gene expression correlates with pseudovirus entry efficiency, which can be detected 2h post VSV transduction (Fig4f). Nevertheless, to address the reviewer’s point for MLV pseudovirus, we added the following statement to the corresponding methods section:

“We note that for MLV pseudoviruses, expression of the luciferase reporter gene entails reverse transcription, genomic integration and forward transcription and is therefore more complex than for VSV pseudoviruses.”

We agree that our membrane fusion assay takes place at the plasma membrane whereas the inhibitor assays (Fig4h) imply that pseudoviruses mainly enter via the endosomal route in the cell types used. This points to the robustness of our results indicating that the functional differences observed between S-short and S-long are independent of the location of membrane fusion and of the assay type. Based on these findings, we propose that cell-cell fusion efficiency and viral entry kinetics are related.

Finally, as indicated in the text (as follows) and shown in Fig4, we see clear (statistically significant) differences between S-long and S-short-mediated entry across two pseudovirus systems and two distinct labs.

“Compared to S-long VSV, S-short VSV entered 1.9-fold, 1.3-fold and 1.1-fold more efficiently in Fcwf-CU cells at 3, 4 and 6 h post infection, respectively (**Fig 4f top panel and Extended Data Figure 10f**) and 1.5-fold, 3-fold and 3.6-fold more efficiently in CRFK cells at 4, 6 and 8 h post infection, respectively (**Fig 4f bottom panel and Extended Data Figure 10f**). [...] Accordingly, we observed greater entry of MLV pseudotyped with FCoV-23 S-short, relative to S-long, at 8 h post-transduction in Fcwf-CU (3.6-fold), AK-D (4-fold) and A-72 (50-fold) cells (**Fig 4g and Extended Data Figure 9b-c**).”

Another comment concerns the first paragraph of the discussion section. This paragraph highlights the D0 in controlling membrane fusion and makes a sensible speculation that increased fusion after D0 deletion correlates with increased pathogenesis. However, can a role for D0 in tissue tropism (as is the case for TGEV/PRCV) be ruled out?

We agree that the FCoV-23 S D0 may play a role in tissue tropism and rephrased this section as follows to reflect this:

“TGEV replicates mainly in small intestine epithelial cells and causes severe and often fatal diarrhea in young piglets whereas PRCV replicates mostly in respiratory epithelial cells and

causes mild respiratory symptoms without diarrhea. Therefore, the loss of D0 possibly yields opposite outcomes in terms of severity for TGEV/PRCV and for FCoV-23 and might alter FCoV-23 tissue tropism (as is the case for TGEV/PRCV).”